# The whole is greater than the sum of its parts: a holistic graph-based assessment approach for natural hazard risk of complex systems

Marcello Arosio[1], Mario L.V. Martina[1], and Rui Figueiredo[1,2]

[1]Scuola Universitaria Superiore IUSS Pavia, Palazzo del Broletto - Piazza della Vittoria 15, 27100 Pavia (Italy)
[2] CONSTRUCT-LESE, Faculty of Engineering, University of Porto, Portugal

*Correspondence to:* Marcello Arosio (marcello.arosio@iusspavia.it)

**Abstract:** Assessing the risk of complex systems to natural hazards is an important but challenging problem. In today's intricate socio-technological world, characterized by strong urbanization and technological trends, the connections and interdependencies between exposed elements are crucial. These complex relationships call for a paradigm shift in collective risk assessments, from a reductionist approach to a holistic one. Most commonly, the risk of a system is estimated through a reductionist approach, based on the sum of the risk evaluated individually at each of its elements. In contrast, a holistic approach considers the whole system as a unique entity of interconnected elements, where those connections are taken into account in order to assess risk more thoroughly. To support this paradigm shift, this paper proposes a holistic approach to analyse risk in complex systems based on the construction and study of a graph, the mathematical structure to model connections between elements. We demonstrate that representing a complex system such as an urban settlement by means of a graph, and using the techniques made available by the branch of mathematics called Graph Theory, will have at least two advantages. First, it is possible to establish analogies between certain graph metrics (e.g. authority, degree, hub values) and the risk variables (exposure, vulnerability and resilience) and leveraging these analogies to obtain a deeper knowledge of the exposed system to a hazard (structure, weaknesses, etc.). Second, it is possible to use the graph as a tool to propagate the damage in the system, not only direct but also indirect and cascading effects and, ultimately, to better understand the risk mechanisms of natural hazards in complex systems. The feasibility of the proposed approach is illustrated by an application to a pilot study in Mexico City.

## 1. Introduction

We live in a complex world: today's societies are interconnected in complex and dynamic social-technological networks, and have become more dependent on the services provided by critical facilities. Population and assets in natural hazard-

prone areas are increasing, which translates into higher economic losses (Bouwer et al., 2009). In coming years, climate change is expected to exacerbate these trends (Alfieri et al., 2017). In this context, natural hazard risk is a worldwide challenge that institutions and private individuals must face at both global and local scales. Today, there is growing

attention to the management and reduction of natural hazard risk, as illustrated for example by the wide adoption of the Sendai Framework for Disaster Risk Reduction (SFDRR, 2015).

### 1.1. Collective disaster risk assessment: traditional approaches

The effective implementation of strategies to manage and reduce collective risk, i.e. the risk assembled by a collection of elements at risk, requires support from Risk Assessment (RA) studies that quantify the impacts that hazardous events may

have on the built environment, economy and society (Grossi and Kunreuther, 2005). The research community concerned with Disaster Risk Reduction (DRR), particularly in the fields of physical risk, has generally agreed on a common approach for the calculation of risk ($R$) as a function of hazard ($H$), exposure ($E$), and vulnerability ($V$): $R = f ( H, E, V )$ (e.g.  Balbi et al., 2010; David, 1999; IPCC, 2012; Schneiderbauer and Ehrlich, 2004). Hazard defines the potentially damaging events and their probabilities of occurrence, exposure represents the population or assets located in hazard

zones that are therefore subject to potential loss, and vulnerability links the intensity of a hazard to potential losses to exposed elements. This framework has been in use by researchers and practitioners in the field of seismic risk assessment for some time (Bazzurro and Luco, 2005; Crowley and Bommer, 2006), and has more recently also become standard practice for other types of hazards, such as floods (Arrighi et al., 2013; Falter et al., 2015).

Despite consensus on the conceptual definition of risk, different stakeholders tend to have their own specific perspectives.

For example, while insurance and reinsurance companies may focus on physical vulnerability and potential economic losses, international institutions and national governments may be more interested in the social behaviour of society or individuals in coping with or adapting to hazardous events (Balbi et al., 2010). As such, even though this risk formulation can be a powerful tool for RA, it has its limits. For instance, it does not consider social conditions, community adaptation or resilience (i.e. a system's capacity to cope with stress and failures and to return to its previous state). In fact, resilience

is still being debated and there is not a common and consolidated approach to assess it (Bosetti et al., 2016; Bruneau et al., 2004; Cutter et al., 2008, 2010).

To overcome some of these limits, different approaches have been put forward in recent research. For example, Carreño et al. (2007b, 2007a, 2012) have proposed to include an aggravating coefficient in the risk equation in order to reflect socio-economic and resilience features. Another example can be found in the Global Earthquake Model, which  aims to

assess so-called integrated risk by combining hazard (seismic), exposure and vulnerability of structures with metrics of

socio-economic vulnerability and resilience to seismic risk (Burton and Silva, 2015). Multi-risk assessment studies resulting from a combination of multiple hazards and vulnerabilities are also receiving growing scientific attention (Eakin et al., 2017; Gallina et al., 2016; Karagiorgos et al., 2016; Liu et al., 2016; Markolf et al., 2018; Wahl et al., 2015; Zscheischler et al., 2018). These new approaches are seen with increasing international interest, particularly with regard to climate change adaptation (Balbi et al., 2010; Terzi et al., 2019).

While some research has explored the potential of an integrated approach to risk and multi-risk assessment of natural hazards, quantitative collective RA still requires further development to consider the connections and interactions between exposed elements. Although holistic approaches are in strong demand (Cardona, 2003; Carreño et al., 2007b; IPCC, 2012), the majority of methods and especially models developed so far are based on a reductionist paradigm, which estimates the collective risk of an area as the sum of the risk of its exposed elements individually, neglecting the links between them. In fact, the reductionist approaches are neglecting one of the famous conjecture attributed to Aristotle "*a whole is greater than the sum of its parts*" (384BC - 322BC).

### 1.2. Modelling natural hazard risk in complex systems: state of the art and limitations

Modern society increasingly relies on interconnections. The links between elements are now crucial, especially considering current urbanization and technological trends. Complex socio-technological networks, which increase the impact of local events on broader crises, characterize the modern technology of present-day urban society (Pescaroli and Alexander, 2016).  Such aspects support the perception that collective risk assessment requires a more comprehensive approach than the traditional reductionist one, as it needs to involve "whole systems" and "whole life" thinking (Albano et al., 2014). The reductionist approach, in which the "*risks are an additive product of their constituent parts*" (Clark-Ginsberg et al., 2018), contrasts with the complex nature of disasters. In fact, these tend to be strongly non-linear i.e. the ultimate outcomes (losses) are not proportional to the initial event (hazard intensity and extensions) and are expressed by emergent behaviour (i.e. macroscopic properties of the complex system) that appear when the number of single entities (agents) operate in an environment, giving rise to more complex behaviours as a collective (Bergström, Uhr and Frykmer, 2016). In the last decade, many disasters have shown high levels of complexity and the presence of nonlinear paths and emergent behaviour that have led towards secondary events. Examples of such large-scale extreme events are the eruption of the Eyjafjallajokull volcano in Iceland in 2010, which affected Europe's entire aviation system, the flooding in Thailand in 2011, which caused a worldwide shortage of computer components, and the energy distribution crisis triggered by hurricane Sandy in New York in 2012.

Secondary events (or indirect losses) due to dependency and interdependency have been thoroughly analysed in the field of critical infrastructures such as telecommunications, electric power systems, natural gas and oil, banking and finance, transportation, water supply systems, government services and emergency services (Buldyrev et al., 2010). Rinaldi et al. (2001), in one of the most quoted papers on this topic, proposed a comprehensive framework to identify, understand and analyse the challenges and complexities of interdependency. Since then, numerous works have focused on the issue of systemic vulnerability, due to the increase in interdependencies in modern society (e.g. Lewis, 2014; Menoni et al., 2002; Setola et al., 2016). Menoni (2001) defines systemic risk as *"the risk of having not just statistically independent failures, but interdependent, so-called 'cascading' failures in a network of N interconnected system components."* The article also highlights that *"In such cases, a localized initial failure ('perturbation') could have disastrous effects and cause, in principle, unbounded damage as N goes to infinity.* Ouyang (2014) reviews existing modelling approaches of interdependent critical infrastructure systems and categorizes them into six groups: empirical, agent-based, system dynamics-based, economic theory-based, network-based, and others. This wide range of models reflects the different levels of analysis of critical infrastructures (physical, functional or socio-economic). Trucco et al. (2012) propose a functional model aimed at i) propagating impacts, within and between infrastructures, in terms of disservice due to a wide set of threats and ii) applying it to a pilot study in the metropolitan area of Milan. Pant et al. (2018) proposed a spatial network model to quantify flood impacts on infrastructures in terms of disrupted customer services both directly and indirectly linked to flooded assets. These analyses could inform flood risk management practitioners to identify and compare critical infrastructure risks on flooded and non-flooded land, for prioritising flood protection investments and improve resilience of cities.

However, this well-developed branch of research is mostly focused on the analysis of a single infrastructure typology, and the aim is usually to assess the efficiency of the infrastructure itself rather than the impact that its failure may have on society. In particular, *"representations of infrastructure network interdependencies in existing flood risk assessment frameworks are mostly non-existent"* (Pant et al., 2018). These interdependencies are crucial for understanding how the impacts of natural hazards propagate across infrastructures and towards society.

A full research branch analyses the complex socio-physical-technological relationships of society considering a System-of-System (SoS) perspective, whereby systems are merged into one interdependent system of systems. In a SoS, people belong to and interact within many groups, such as households, schools, workplaces, transport, health care systems, corporations and governments. In a SoS, the dependencies are therefore distinguished between links within the same system or between different systems (Alexoudi et al., 2011). The relation between different systems are modelled in the literature using qualitative graphs or flow diagrams (Kakderi et al., 2011) and by matrices (Abele and Dunn, 2006).

Tsuruta and Kataoka (2008) use matrices to determine damage propagation within infrastructure networks (e.g. electric power, waterworks, telecommunication, road) due to interdependency, based on past earthquake data and expert judgment. Menoni (2001) proposes a framework showing major systems interacting in a metropolitan environment based on observations on the Kobe earthquake. Lane and Valerdi (2010) provide a comparison of various SoS definitions and concepts, while Kakderi et al. (2011) have delivered a comprehensive literature review of methodologies to assess the vulnerability of a SoS.

### 1.3. Positioning and aims

The aspects of complexity and interdependency have been investigated by various models of critical infrastructure as a single system, or as systems of systems, which are networks by construction (e.g. drainage system or electric power network, Holmgren, 2006; Navin, 2016). However, the current practice related to both single system and SoS needs further research, in particular when it comes to modelling the complexity of interconnections between individual elements that do not explicitly constitute a network, which tends to be neglected by traditional reductionist risk assessments. In fact, although several authors have shown how to model risk in systems which are already networks by construction (Havlin et al., 2010; Reed et al., 2009; Rinaldi, 2004; Zio, 2016), fewer have addressed the topic of risk modelling in systems where that is not the case, i.e. systems are not immediately and manifestly depicted as a network (Hammond et al., 2013; Zimmerman et al., 2019). These include cities, regions or countries, which are complex systems made of different elements (e.g. people, services, factories) connected in different ways among each other in order to carry out their own activities. Therefore, in this manuscript we would like to promote an approach, which has previously deserved the attention of other authors, to model the interconnections between the elements that constitute those systems and assess collective risk in a holistic manner. The approach involves the translation of the complex system into a graph, i.e. a mathematical structure used to model relations between elements. This allows modelling and assessing interconnected risk (due to the complex interaction between human, environment and technological systems) and cascading risk (which results from escalation processes). The interactions between elements at risk and their influence on indirect impacts are assessed within the framework of Graph Theory, the branch of mathematics concerned with graphs. The results can be used to support more informed DRR decision making (Pescaroli and Alexander, 2018).

The aims of this paper can be summarized as follows:

- to call for a paradigm shift from a reductionist to a holistic approach to assess natural hazard risk, supported by the construction of a graph;

- to show the potential advantages of the use of a graph: (1) understanding fundamental aspects of complex systems which may have relevant implications to natural hazard risk, leveraging well known graph properties, (2) using the graph as a tool to model the propagation of impacts of a natural hazard and, eventually, assess risk in complex systems;
- to present the feasibility of implementing the approach through a pilot study in Mexico City;
- to discuss the limitations, potentialities and future developments of this approach compared to other more traditional approaches.

## 2. Methodology

In this section, which presents the methodology, we aim to answer the three following questions:

1) How to "translate" a complex system, that does not explicitly constitute a network, into a graph?
2) Which properties of the graph could give us insights on the risk related properties of the system?
3) How to propagate the impacts of a natural hazard by means of the graph?

The answers to these questions are formulated proposing the workflow of the graph-based approach, which is divided in three main steps described in the following sub-sections: 2.1 Construction of the graph; 2.2 Analogy between graph properties and risk variables; and 2.3 Hazard impact propagation within the graph.

The workflow is presented in Figure 1.

### 2.1. Construction of the graph

The construction of a graph for systems already in a form of a network is well developed and consolidated in the literature (e.g. Rinaldi, 2004; Setola et al., 2016). Instead, the use of the graph-theory – and the exploitation of its diagnosis tools - for systems non already structurally in the form of a network is relatively new. At this regard, in this section we propose a procedure to build a graph for a complex system such as a city by linking the individual elements constituting it.

According to the objects of each specific context, the graph construction phase starts by defining the hypothesis of the analysis and the system boundaries according to the objects of each specific context. In particular, it establishes the two main objects of the graph: vertices (nodes) and edges (links) and their characteristics.

The nodes can theoretically represent all the entities that the analysis wants to consider: physical elements like a single building, bridge and electric tower, suppliers of services such as schools, hospitals and fire brigades, or beneficiaries such as population, students or specific vulnerable groups such as elderly people. Due to the very wide variety of elements that can be chosen, it is necessary to select the category of nodes most relevant to the specific context of analysis. It is also

necessary to define, for each node, the operational state that can be characterized, from the simplistic Boolean

(functional/non-functional), to discreet states (30/60/100% of service/functionality), or even a complete continuous

function (similarly to vulnerability functions). In a graph, the states of each node depend both on the states of the adjacent

nodes and on the hazard. In this paper, we use the term *node* to refer to its graph characteristics and term *element* to refer

to the entity that it represents in the real world. The links between the nodes that create the graph can range from physical,

geographical, cyber or logical connections (Rinaldi et al., 2001). According to the different typologies of connections and

nodes selected, it is necessary to define direction and weight of the links. The graph will be directed when the direction

of the connection between elements is relevant and it will be weighted if the links have different importance, intensity or

difference capacity.

In defining the topology, it is crucial to define the level of analysis details coherently with the scope and scale, both for

the selection of elements and for the relationship between elements that need to be considered. In the case of a very high

detail for example, a node of the graph could represent a single person within a population, and in the case of lower

resolution, it could represent a large group of people with a specific common characteristic, such as living in the same

block or having the same hobby. In the case of analyses at a coarser level, an entire network (e.g. electric power system)

can be modelled as a single node of another larger network (e.g. national power system). The definition of the topology

structure of the graph also identifies immediately the system boundaries (e.g. which hospitals to be considered in the

analysis: only the potential flood area, the ones in the district or in the region?). Up to which extent it is necessary to

consider elements as nodes of the graph? The topology definition is a necessary step to perform the computational analysis

and introduces approximations of the open systems that need to be acknowledged.

Once the graph is conceptually defined, in order to actually build the graph, it is then necessary to establish the connection

between all the selected elements. The relations described above determine the existence of connections between

categories of elements, but it does not define how a single node of one category is linked to a node of another category.

Therefore, it is necessary to define rules that establish the connections between each single node. For the sake of clarity,

an example could be the following: the conceptual relationship is defined between students and school ("students go to

school"); subsequently, it is necessary to make the link between each student and a school in the area, applying a rule

such as "students go to the closest school". This is an example of geographical connection with nodes that are linked by

their spatial proximity.

The connections between the single elements can be represented either by a list of pairs of nodes or, more frequently, by

the adjacency matrix. Any graph $G$ with $N$ nodes can be represented in fact by its adjacency matrix $A(G)$ with $N$ x $N$

elements $A_{ij}$, whose value is $A_{ij} = A_{ij} = 1$ if nodes $i$ and $j$ are connected, and 0 otherwise. If the graph is weighted, $A_{ij}=A_{ji}$

can have a value between 0 and *1* expressing the weight of the connection between the nodes. The properties of the nodes are represented in both cases by another matrix, with a column for each property associated with the node (e.g. name, category, type). In practical terms, the list of all connections or the adjacency matrix can be automatically obtained via GIS analysis, in the case of geographical connections, or by database analysis, in the case of other categories of connections. The list of nodes, together with either the list of links or the adjacency matrix, are the inputs to build the mathematical graph.

Once a graph has been setup and a constructed, it is then possible to compute and analyse its properties by means of Graph Theory and propagate the hazard impact into the graph as illustrated in the following sub-sections.

## 2.2. Analysis of the graph properties

### 2.2.1. Summary of relevant graph properties

The mathematical properties of a graph can be studied using Graph Theory (Biggs et al., 1976), which is the branch of mathematics that studies the properties of graphs (Barabasi, 2016). Graphs can represent networks of physical elements in the Euclidean space (e.g. electric power grids and highways) or of entities defined in an intangible space (e.g. collaborations between individuals) (Wilson, 1996). Since its inception in the eighth century (Euler, 1736), Graph Theory has provided answers to questions in different sectors, such as pipe networks, roads, and the spread of epidemics. Over recent decades, studies of graph concepts, connections and relationships have strongly accelerated in every area of knowledge and research (from physics to information technology, from genetics to mathematics, to building and urban design), showing the image of a strongly interconnected world in which relationships between individual objects are often more important than the objects themselves (Mingers and White, 2009).

Formally, a complex network can be represented by a graph *G* which consists of a finite set of elements *V(G)* called vertices (or nodes, in network terminology), and a set *E(G)* of pairs of elements of *V(G)* called edges (or links, in network terminology) (Boccaletti et al., 2006). The graph can be undirected or directed (Figure 2a and b). In an undirected graph, each of the links is defined by a pair of nodes *i* and *j*, and is denoted as $l_{ij}$. The link is said to be incident in nodes *i* and *j*, or to join the two nodes; the two nodes *i* and *j* are referred to as the end-nodes of link $l_{ij}$. In a directed graph, the order of the two nodes is important: $l_{ij}$ stands for a link from *i* to *j*, node *i* points to node *j*, and $l_{ij} \neq l_{ji}$. Two nodes joined by a link are referred to as adjacent (Börner et al., 2007; Luce and Perry, 1949). In addition, a graph could have edges of different weights representing their relative importance, capacity or intensity. In this case, a real number representing the weight of the link is associated to it, and the graph is said to be weighted (Figure 2c) (Börner et al., 2007).

A short list of the most common set of node, edge and graph measures used in Graph Theory is presented here and summarized in Table 1 (Nepusz and Csard, 2018; Newman, 2010). There are measures that analyse the properties of nodes or edges, local measures that describe the neighbourhood of a node (single part of the system), and global measures that analyse the entire graph (whole system). From a holistic point of view, it is important to note that since some node/edge measures require the examination of the complete graph, this allows looking the studied area as a unique entity that results from the connections and interactions between its parts and characterizing the whole system.

The degree (or connectivity, $k$) of a node is the number of edges incident with the node. If the graph is directed, the degree of the node has two components: the number of outgoing links (referred to as the degree-out of the node), and the number of ingoing links (referred to as the degree-in of the node). The distribution of the degree of a graph is its most basic topological characterization, while the node degree is a local measure that does not take into account the global properties of the graph. On the contrary, path lengths, closeness and betweenness centrality are properties that consider the complete graph. The path length is the geodesic length from node $i$ to node $j$: in a given graph, the maximum value of all path lengths is called diameter and the average shortest path length is named characteristic path length. Closeness is the shortest path length from a node to every other nodes in the network, and betweenness is defined as the number of shortest paths between pairs of nodes that pass through a given node.

Other relevant characteristics that are commonly analysed in directed graphs to assess the relative importance of a node, in terms of the global structure of the graph, are the hub and authority properties. A node with high hub value points to many other nodes, while a node with high authority value is linked by many different hubs. Mathematically, the authority value of a node is proportional to the sum of the node hubs pointing to it and the hub value of a node is proportional to the sum of authority of nodes pointing to it (Nepusz and Csard, 2018; Newman, 2010). In the World Wide Web, for example, websites (nodes) with higher authorities contain the relevant information on a given topic (e.g. wikipedia.com) while websites with higher hubs point to such information (e.g. google.com).

The mathematical properties presented above are useful metrics to analyse the structural (i.e. network topology, arrangement of a network) and functional (i.e. network dynamics, how the network status changes after perturbation) properties of complex networks. Depending on the statistical properties of the degree distributions, there are two broad classes of networks: homogeneous, and heterogeneous (Boccaletti et al., 2006). Homogeneous networks show a distribution of the degree with a typically exponential and fast decaying tail, such as Poissonian distribution, while heterogeneous networks have a heavy-tailed distribution of the degree, well-approximated by a power-law distribution. Many real-world complex networks show power-law distribution of the degree and these are also known as scale-free networks, because power-laws have the same functional form on all scales (Boccaletti et al., 2006). Networks with highly

heterogeneous degree distribution have few nodes linked to many other nodes (i.e. few hubs), and a large number of poorly connected elements.

The properties of the static network structure are not always appropriate to fully characterize real-world networks that also display dynamic aspects. There are examples of networks that evolve with time or according to external environment perturbations (e.g. removal of nodes/links). Two important properties to explore the dynamic response to a perturbation are percolation thresholds and fragmentation modes.

Percolation was born as the model of a porous medium, but soon became a paradigm model of statistical physics. Water can percolate in a medium if a large number of links exists (i.e. the presence of links means the possibility of water flowing through the medium), and this depends largely on the fraction of links that are maintained. When the graph has is characterized by many links, there is a higher probability that connection between two nodes may exist and, in this case, the system percolates. Vice versa, if most links are removed, the network becomes fragmented (Van Der Hofstad, 2009). The percolation threshold is an important network feature resulting from the percolation concept which is obtained by removing vertices or edges from a graph. When a perturbation is simulated as a removal of nodes/links, the fraction of nodes removed is defined as $f = \frac{Nodes_{removed}}{Nodes_{Total}}$, and the probability of nodes/links present in a percolation problem is $p = 1 - f = \frac{Nodes_{remainig}}{Nodes_{Total}}$. Consequently, it is possible to define the percolation threshold ($p_c$) as the minimum value of $p$ that leads to the connectivity phase of the graph (Gao et al., 2015). In practical terms, the percolation threshold discriminates between the connected and fragmented phases of the network. In a random network (i.e. network with N nodes where each node pair is connected with probability p), for example, $p_c=1/\bar{k}$, where $\bar{k}$ is the mean of degree $k$ (Bunde and Havlin, 1991).

The second property that investigates dynamic evolution is the fragmentation (i.e. number and size of the portions of the network that become disconnected). The number and the size of the sub-networks obtained after removing the vertices/edges provide useful information. In the case of a so-called giant component fragmentation, the network retains a high level of global connectivity even after a large amount of nodes have been removed, while in the case of total fragmentation, the network collapses into small isolated portions. For this reason, "*keeping track of the fragmentation evolution permits the determination of critical fractions of removed components (i.e., fraction of component deletion at which the network becomes disconnected), as well as the determination of the effect that each removed component has on network response*" (Dueñas-Osorio et al., 2004).

### 2.2.2. Analogy between graph properties and risk variables

The proposed graph properties can be used to more thoroughly characterize systems of exposed elements. In fact, the traditional conceptual skeleton to describe risk can still be adopted within the framework of the proposed graph-based approach. The properties calculated from a graph consist in a new layer of information for some of those risk variables that go beyond their traditional interpretations within the reductionist paradigm. In particular, they provide a more comprehensive characterization of the single nodes (deriving from their relationships with other nodes), as well as of the

system as a whole. As such, from the risk variables presented in Section 1, the hazard preserves its traditional definition as an event that can impact such systems, or part(s) of it, with certain intensities and associated probabilities of occurrence. For the three other variables, namely, exposure, vulnerability, and resilience, below we propose and provide an innovative and original discussion on their analogies with the graph properties presented in previous Sub-section. The analogies are summarized in Table 2.

**Exposure**

In analogy with the traditional approach but at the same time extending its concept, the value of each exposed elements can be estimated as the relative importance that is given to it by the graph, which is measured by the network itself by means of the connections that point to each node. In Graph Theory, this relative importance among elements, based on standardized values, can be investigated through the authority analysis. A high authority value of a node indicates that

there are many other nodes (or otherwise some hubs) that provide services (i.e. providers or suppliers) to that node. In other words, the system privileges it compared with others according to their connections with the provider nodes. For example, a factory settled in an industrial district may receive more services (e.g. electric power, roads for heavy vehicles, logistic systems) than a factory located in the old quarter of a city; in this case, the former is structurally privileged by the system compared with the latter.

**Vulnerability**

In the reductionist approach, vulnerability is the propensity of an asset to be damaged because of a hazardous event. By adopting a graph perspective, the vulnerability can be estimated both for the single node as well as for the system as a whole.

In the first case, the vulnerability depends on the relationship that the node has with the others. In particular, the closeness

represents the likelihood of a node to be affected indirectly by a hazard event due to the lack of services provided by other nodes. A lower value of closeness, i.e. a shortest path length from a node to every other nodes in the network, means a

higher probability of a node of being impacted by a hazard event. On the other hand, high value of closeness, i.e. a longer path length from a node to every other nodes in the network, means a low probability of being impacted.

In the second case, the vulnerability can be defined as the propensity of the network to be split into isolated parts due to a hazardous event. In that condition, an isolated part is unable to provide and receive services, which can translate into indirect losses. The system vulnerability, therefore, can be evaluated by means of the following graph properties: hubs, betweenness and degree out distribution. The presence of nodes with high hub values indicates a propensity of the network to be indirectly affected more extensively by a hazard event, since a large number of nodes are connected with the hubs. A network that has nodes with high betweenness values has a higher tendency to be fragmented, because it has a strong aptitude to generate isolated sub-networks. Finally, the degree distribution, which expresses network connectivity of the whole system (i.e. the existence of paths leading to pairs of vertices), has a strong influence on network vulnerability after a perturbation. The shape of the degree distribution determines the class of a network: heterogeneous graphs (power-law distribution and scale-free network) are more resistant to random failure, but they are also more vulnerable to intentional attack (Schwarte et al., 2002). As emphasised above, scale-free networks have few nodes linked to many nodes (i.e. few hubs), and a large number of poorly connected elements. In the case of random failure, there is a low probability of removing a hub, but if an intentional attack hits the hub, the consequences for the network could be catastrophic.

**Resilience**

Resilience differentiates from vulnerability in terms of dynamic features of the system as a whole. The properties and functions used to model vulnerability are static characteristics that do not consider any time evolution, or using the words of Sapountzaki (2007), "*vulnerability is a state, while resilience is a process*"; in fact the definition of resilience implies a time evolution of the characteristics of the whole system. In addition, Lhomme et al. (2013) underline "*the need to move beyond reductionist approaches, trying, instead, to understand the behaviour of a system as a whole*". These two features, dynamic aspect and whole system, make vulnerability different from resilience and further clarify the need to develop an approach that it is able to consider the dynamic of the system as a whole.

In this context, the study of the percolation threshold ($p_c$) can be used to explain the resilience of the network after a perturbation. The $p_c$ value distinguishes between the connectivity phase (above $p_c$) and the fragmented phase (below $p_c$). In the connectivity phase, the network can lose nodes without losing the capacity to cope with the perturbation as a network, while in the fragmented phase, the network does not actually exist anymore and the remaining nodes are unable to cope with the disruption alone.

This critical behaviour is a common feature also observed in disasters induced by natural hazard. In some cases, the exposed elements withstand some damage and loss, but the overall system maintains its structure. However, there are

events in which the amount of loss (affected nodes) is so relevant that the system loses the overall network structure. In the first case, the system has the capacity to cope independently and tackle the event, while in the second case, the system is unable to cope.

The dynamic responses are characterized by the network fragmentation property, which describes the performance of a network when its components are removed (Dueñas-Osorio and Vemuru, 2009). For instance, the so-called giant component fragmentation (the largest connected sub-network) and the total fragmentation describe network connectivity and determine the failure mechanism (Dueñas-Osorio et al., 2004). Keeping track of fragmentation evolution makes it possible to determine both the critical fraction of components removed (i.e. the smallest component deletion that disconnects the network), and the effect that each component removed has on the network response.


For these reasons, we consider percolation threshold and network fragmentation good indicators of resilience, also because they are able to show the emergent behaviour of the whole system beyond just considering the single parts of the network (e.g. node).

### 2.3. Hazard impact propagation within the graph

While the literature of the impact propagation or cascading effects for critical infrastructures is large (e.g. Pant et al., 2018; Trucco et al., 2012), applications on the risk quantification of natural hazard including the cascading effects are scarce. Besides the considerable amount of information that can be obtained by analysing graph properties from the viewpoint of natural hazard risk, the graph itself also provides an optimal structure to propagate the impacts of a hazard throughout an affected system. Indeed, the use of a graph allows estimating, besides direct losses to elements directly

affected (such as elements within a flooded area), also indirect losses to elements outside the affected area that rely on services provided by directly hit elements, which may have lost some capacity to provide those services as a result. The propagation and quantification of impacts through a graph allows understanding the risk mechanisms of the system, and identifying weaknesses that can translate into larger indirect consequences. It also enables the possibility of quantitatively estimating risk considering those indirect consequences.

Figure 3 depicts this process through a conceptual flowchart. In order to propagate the impacts by means of the graph and quantify indirect losses resulting from second-order and cascading effects, the modelled graph must first be integrated with hazard data. This data must include hazard footprints that allow establishing the hazard intensity (e.g. water depth) at the location of each element. The direct and indirect impacts can then be computed according to the proposed methodology, based on three levels of vulnerability:

- Level I is the physical vulnerability of a directly affected element in its traditional definition. The hazard intensity is the input variable to compute the direct damage of the element;

- Level II is the vulnerability associated to the link between an affected element and its receivers. The direct damage as obtained by vulnerability level 1 is the input to compute the loss of service provided by the directly-damaged element to the elements that receive it;

- Level III is the vulnerability of the service-receiving element. The loss of service as obtained by vulnerability level 2 is the input to estimate the indirect loss of the element that receives the service.

These vulnerabilities can be represented by vulnerability functions analogous to the ones adopted within the traditional risk assessment approach, and can be different for each category of element and service.

By computing impacts for hazard scenarios with different probabilities of occurrence, and adopting the three level of
vulnerability functions, a quantitative estimate of risk can be obtained. An illustrative example of propagation of impacts is presented in Section 2.4, and more detailed information on the propagation of impacts through the graph and the estimation of impact are presented in the pilot study in section 3.3.

### 2.4. Illustrative example

In order to illustrate the application of the graph-based approach in the characterization of a system exposed to natural
hazards, in Figure 4 we present an example of a hypothetical city comprising various elements of different types which provide services among them. In specific, our example includes 20 elements: 9 Blocks of residential buildings, 1 Hospital, 2 Fire Stations, 3 Schools, 3 Fuel Stations and 2 Bridges. Blocks are intended to represent the population, which receives services from the other nodes. Bridges provide a transportation service, Fire Stations provide a recovery service, Hospitals provide a healthcare service, Schools provide an education service, and Fuel Stations provide a power service. Figure 4(a)
shows how the elements are connected into a graph. The authority and hub values have been computed using the R igraph package (http://igraph.org/r/). The full library of functions adopted are available in Nepusz and Csard (2018).

In Figure 4(b), the size of the elements is proportional to their authority values. Blocks 6, 18, 19 and 20 have higher authority values than the other elements of this typology because they receive a service from the Hospital (node 16), which is an important hub. Fire Station 5 and School 9 have high values of authority because they are serviced by Bridge
3, which is also an important hub. The importance of a node in Graph Theory is closely connected with the concept of topological centrality. Referring to the illustrative example, Block 6 has the highest authority value; if a flood hit it, it would therefore affect the most central node of the network, or in other words, the node which is implicitly more privileged by the system.

In Figure 4(c), the major hubs are the elements with largest diameters: Hospital 16, Bridge 3, School 7 and Fuel Station 15. Bridge 3 is an important hub since it provides its service to Block 6, which has the highest authority value, and to Fire Station 5 and School 9. Fuel Station 15 and School 7 are also important hubs because they provide services to Block 6. The elements in the south-east part of the network inherited a relative importance (i.e. authority) from the most important hub in that area (i.e. Hospital 16). Bridge 3 is an exception to this aspect; in fact, this Bridge connects the south part (i.e. Block 6) with the north part of the city (i.e. Fire Station 5 and School 9). A flood event in the south-east part of the network would likely generate a major indirect impact on the whole system compared to other parts of the network.

We assume that these elements are located in a flood-prone area and Bridge 3 and Block 6 are directly flooded (Figure 4d). Since those elements are directly damaged, it is possible to follow the cascading effects following the direction of the service within the graph from providers to receivers. In this artificial example, the transportation service provided from by Bridge is lost and this has an indirect consequence to the Hospital 16, which is not directly damaged but cannot provide healthcare services since people cannot reach the hospital any more. The graph allows to extend the impact not only to the elements directly hit by the hazard, but also to all elements that receive service from element directly or indirectly affected by the hazard.

Note that similar analyses could be carried out for other properties of the graph (e.g. betweenness) in order to obtain additional insight into the properties of the system, which could be useful for the purpose of a risk assessment. For the sake of brevity, such analyses have not been included here. A complete study of all relevant graph properties discussed above and a more realistic hazard scenario are presented in the following section.

## 3. Pilot study: Mexico City

Floods, landslides, subsidence, volcanism and earthquakes make Mexico City one of the most hazard-prone cities in the world. Mexico is one of the most seismological active regions on earth (Santos-Reyes et al., 2014), floods and storms are recorded in indigenous documents, and the Popocatépetl volcano has erupted intermittently for at least 500,000 years. At present, people settle in hazardous areas such as scarps, steep slopes, ravines and next to stream channels.

The Mexico City Metropolitan Area (MCMA) is one of the largest urban agglomerations in the world (Campillo et al., 2011). This pilot study focuses on Mexico City (also called the Federal District - MCFD), where approximately 8.8 million people live. The choice of MCFD as a pilot case allows showing the importance of modelling connections and interdependencies in a complex urban environment.

Tellman et al. (2018) show how the risk in Mexico City's history has become interconnected and reinforced. In fact, as cities expand spatially and become more interconnected, the risk becomes endogenous. Urbanization increases the demand for water and land. The urbanized area inhibit aquifer recharge, and the increase in water demand exacerbate subsidence due to an increase of pumping activity out of the aquifer. Subsidence alters the slope of drainage pipes, decreasing the efficiency of built infrastructure and the capacity of the system to both remove water from the basin in floods as well as deliver drinking water to consumers. This exacerbates both water scarcity and flood risk.

## 3.1. Construction of the graph

Given the very large scale of the city, certain simplifications and hypotheses had to be assumed for conceptualizing the network. Furthermore, the choice of element typologies, the connections between them and the definition of rules were also done considering the availability of data provided by the UNAM Institute of Engineering for this study case. While this data is only partially representative of the entirety of the exposed assets in MCFD (with the exclusion of 3 districts for which the data were not available: Álvaro Obregón, Milpa Alta and Xochimilco), we consider it suitable for the specific purpose of this work, which is to illustrate the proposed approach and highlight its potential. Note that the boundaries of the system are defined by the selection of typologies, connections, and the studied geographical area. These simplifications and hypotheses of real open-ended systems, while necessary to enable the computational analysis, should be recognized and taken into account when evaluating the results of the analysis (Clark-Ginsberg et al., 2018).

Among the possible exposed elements, we selected 6 typologies that are representative of both emergency management phase (e.g. Fire Stations) and long term impacts (e.g. Schools). The typologies of elements considered in this pilot case, which provide and/or receive services reciprocally, are: Fire Stations, Fuel Stations, Hospitals, Schools, Blocks, and Crossroads. The Fire Station represent the node type from which the Recovery service is provided to all the other elements present in the area (except Crossroads). The Fuel Station represents the node type that provides the Power service, the Hospital provides the Healthcare service, and the School provides the Education service; the elements with these three typologies deliver their respective services to all the Blocks. The Block is the node type defined as the proxy for population, which receives services from all the other considered elements. The simulation uses Blocks instead of population, as this enables a reduction in computational demand by lowering the number of nodes from 8 million to few tens of thousands. Finally, the analysis considers 17 Crossroads, which provide the Transportation service to all the other elements. The Crossroads were identified by selecting the major intersections between the main highways present in the road network of MCFD. All the typologies, numbers of elements and the connections between them are presented in the

conceptual graph in Figure 5 and Figure 6 presents the GIS representation of the providers, and the services that are

provided between them.

The link between two elements of two different typologies was set up based on the geographical proximity rule: each specific service is received by the nearest provider (e.g., a Block receives the Education service from the closest School, and the School receives the Recovery service from the closest Fire Station). This simple assumption is due to the lack of data available at this stage; in case of more data, it will be possible to define this relation more accurately (e.g. School

offers education service to its zoning) but without changing the general validity of the method. Note that this hypothesis does not consider the redundancy that might exist between some services, which would necessarily influence the propagation of cascade effects. The service provided by the road network was modelled considering that each element in the area receives a transportation service from the closest Crossroad among the 17 that were identified. This approach does not aim to be representative of the complete behaviour of the road network system, particularly the paths between

nodes or possible alternative paths, but it does allow considering transportation network in the analysis in a simplified manner.

The list of nodes, which contain all the elements of all typologies, together with the list of links between them, both obtained according to the hypothesis presented above, are the inputs to build the mathematical graph. As for the illustrative example, the graph was obtained using the open source igraph package for network analysis of the R environment .

**3.2. Analysis of the graph properties**

The following paragraphs present the results from the graph analysis and show how the properties of the single elements and the whole system are assessed, from both provider (or supplier) and receiver (or consumer) perspectives.

### 3.2.1. Vulnerability of the single elements

As described in section 2.2.2, the systemic vulnerability of a node is the aptitude to remain isolated from the whole system

when the graph is perturbed. The tendency to observe isolated parts is here analysed by the closeness property, which measures the mean distance from a vertex to other vertices Figure 7 shows the geographical distribution values of closeness-in values of the Blocks.

In accordance to the model conceptualization, the Blocks increase their distance to the network if their providers are not connected between each other. As an example, if a School and a Hospital provide services to a Block, the closeness-in of

this Block will be higher if the School and the Hospital receive the transportation service from same Crossroad and this

one is also serving the Block. In this specific case, where the nodes are more interconnected, the distance between the Block node and the whole network is lower, and by definition its closeness-in is higher.

Figure 7 shows that the region with the majority of Blocks with the highest values of closeness-in is in the southeast part of MCFD. This area is the part of the city that is surrounded by few providers, which are the major hubs as illustrated in the next paragraph in Figure 10. The presence of few providers forces them to exchange services between themselves and to serve all the receivers of the area, meaning that the Blocks have a lower distance to the providers and can therefore be more vulnerable.

### 3.2.2. Vulnerability of the whole system

The analysis in this section shows the structural properties of the whole network (i.e. network topology, arrangement of a network) and investigates how the network, as a unique entity, is vulnerable to a potential external perturbation (e.g. hazardous event).

As mentioned in Section 2.2, there are two types of networks, heterogeneous or homogeneous, depending if the degree distribution is respectively heavy tailed or not. Heterogeneous networks have few hubs that appear as outliers in the degree distribution, this feature can represent a potential weakness of a system, because if one of the hubs is affected by an event, it will propagate the impacts more extensively than other nodes. Note that this is not per se an indication of risk, which is a function of not only the exposed system but also the hazard. However, it may be used to evaluate the vulnerability of the system as a whole, similarly to how single-site vulnerability analyses assess the potential impact of an event regardless of its actual likelihood.

There is an objective way to estimate if the degree distribution is heavy tailed by means of its statistical properties: a distribution is defined heavy tailed if its tail is not bounded by the exponential distribution. In order to verify if the degree distribution of a network is heavy tailed, one can infer the Generalized Pareto Distribution (GPD) on the observation and analyse the shape parameter (Beirlant et al., 1999; Scarrott and Macdonald, 2012). If the shape parameter of the GPD is equal to zero, the tail of distribution is exponential. Instead, if the shape parameter is greater than zero, the tail of the distribution if fatter than the exponential, and therefore the distribution is heavy tailed. However, in order to fit the GPD to the data, it is first necessary to select a threshold value and consider only the exceeding values. There are different techniques to select the right threshold value (Coles, 2001). Figure 8 shows the values of shape parameter (sp) for the degree-out distribution of the Mexico City network for different values of threshold in terms of data percentile. The shape parameter $\varepsilon$ is positive for any value below 0.8; over that value, the degree distribution is meaningless and does not represent the whole network anymore, but only the extreme values that are the only still above the threshold. For this

reason, we can assert that the degree-out distribution is heavy tailed. This confirms that the network built for Mexico City is strongly non-homogeneous, with few hubs (providers) that are linked to many elements. According to these results, if an hazard event hit one of these few nodes with high value of hub, the consequences for the network could be catastrophic due to the central role of the hub.

### 3.2.3. Cascade effects

The analysis of the topological structure of the providers in the network show their relative relevance into the system, according to their connections with the receivers. In particular, we propose a comparison between providers through the analysis of two properties: hub analysis of all nodes that provide service to the population, and betweenness analysis of the Crossroads.

**Providers: role of hubs**

The importance of a node in directed graphs, within the purpose of providers that deliver a service, is closely connected with the concept of topological centrality: the capacity of a node to influence, or be influenced by, other nodes by virtue of its connectivity. In Graph Theory, the influence of a node in a network can be provided by the eigenvector centrality, of which the hub and authority measures are a natural generalization (Koenig and Battiston, 2009). A node with high hub value points to many nodes, while a node with high authority value is linked by many different hubs.

The hub analysis considers all the elements in the graph that provide services; for this reason, Blocks are excluded from this analysis. Figure 9 reveal outliers that are useful to identify the elements in the graph that, in case of potential failure, could have a large impact on the network due to, for instance, their role as major hubs. In particular, one Hospital has the hub value equal to 1, which by definition is the highest, immediately followed by a Crossroad with value around 0.85, while some Schools, Fuel Stations and Fire Stations have hub values around 0.5. The ranking of elements according to
their hub values can be a very useful to prioritize intervention actions and maximize the mitigation effects for the whole network. If an external perturbation hit an element with very high hub value, the cascading effects on the network would be more relevant due to its central role in the system. On the other hand, a mitigation measure applied to the elements with higher hub values would produce higher benefit on the whole network.

    The hub outliers in Figure 9 are associated to the elements of the network that are geographically located mainly in the
southeast part of Mexico City; as shown in Figure 10, the biggest icons are in this part of the city. Based on the available data, the density of elements that provide services in southeast part is much lower compared to the other areas of the city; as such, the few providers existing in this part become important hubs for the whole system.

This part of the city has few providers that are central hubs of the city and Blocks with very high closeness. Together, these two aspects underline the need for additional providers in this area. This would reduce the respective number of receivers, decreasing the hub values of providers and reducing the number of Blocks depending on each of them.

**Crossroads: betweenness analysis**

As described in section 2.2.2, a network that has nodes with high betweenness values has higher tendency to be fragmented because it has a strong aptitude to generate isolated sub-networks. In this case study, the transportation is the only service that allows the analysis of the betweenness values of the nodes. In fact, vehicles (e.g. fire trucks, family cars) need to pass through Crossroads to go from point A to point B (e.g. fire trucks going from a Fire Station to an affected location; a family car going from a Block to a School). The betweenness analysis presented here shows the number of shortest paths between pairs of nodes that pass through the selected Crossroads. As mentioned previously, the few Crossroads considered in this pilot study are not intended to reproduce the very complex road network of Mexico City, but to present some highlights of the betweenness property.

Figure 11 shows the Crossroads adopted in the analysis, where the dimension of the icons is proportional to the value of betweenness. It can be observed that the Crossroads in the ring road around the city centre have higher values of betweenness, which is due to the fact that they connect the very large suburb areas and the city centre. In particular, the Crossroads in the south have the highest values, because the number of nodes in the south is greater than that in the north of the city. Instead, the Crossroads in the city centre connect mostly the nodes that are inside the ring road, and for this reason they have lower values of betweenness.

The betweenness value shows which Crossroad is more central, or more important and influent in the network, based on shortest paths between the nodes. As an example, in case a Crossroad is flooded, it will reduce or completely interrupt its transportation service. A Crossroad with higher betweenness will influence a higher number of nodes, and as such, if its functionality is affected, this will have a higher impact on the network compared to a Crossroad with lower betweenness.

### 3.2.4. Exposure: which elements have higher centrality in the system?

Regarding the analysis from the receivers' point of view, we explore how the system privileges some receivers compared with others according to their connections with the providers. In particular, we propose a comparison between receivers through the authority analysis.

Figure 12 show that the authority of the nodes tend to be clustered around certain values, presenting discontinuities between them. This results from the fact that all Blocks receive exactly five services from five providers (i.e. degree-in=5), and as such have the same values of authority when they receive services from the same provider nodes. Nodes

with similar authority values should therefore be geographically located close to one another. This is confirmed in Figure 13, where the Blocks are represented in space and coloured according to their authority values.

Figure 13 shows a clear pattern from low values in the northwest to higher values in the southeast part of MCFD. The Blocks with higher authority values are located in the part of the city that is surrounded by the providers with highest hub values, as illustrated in Figure 10. In contrast, the Blocks in the city centre and in the northwest have the lowest values of authority. In fact, this part of the city has the highest density of providers, which decreases the number of receivers for each provider, and consequently their hub values. Note that this aspect likely results from the assumption of not considering redundancy, meaning that each node can only receive a certain service from its nearest provider. Otherwise, if redundancy was considered, the Blocks in the city centre would receive the same service from many different providers due to the higher density of such nodes.

According to these results, if a hazardous event hits the Blocks in the southeast part of the city, this will impact the whole system more heavily, because there will be more requests to the same few hubs. Such hubs, which are potentially more overburdened in an ordinary situation due to the high number of services they provide, can put in crisis a considerable part of the network after an external perturbation. The strong correlation between hubs and authority explain the results described above. However, it is necessary to underline that these outputs also reflect the assumption of the rules of proximity adopted in this model, where the network has no redundancy by construction. The redundancy can change the values of hub and authority of the nodes, and therefore influence the magnitude of cascade impacts that are presented in the next section.

### 3.3. Flood impact propagation within the graph

In this section, we present a preliminary analysis of a flood scenario in the case of Mexico City according to the proposed graph-based approach. The aim is to show the potential of the approach to highlight the impacts of a hazard over the whole system, including indirect consequences to elements outside the flooded area, based on a graph built for this specific purpose.

The adopted hazard scenario is based on the development of a simplified model that explicitly integrates the drainage system and the surface runoff for the estimation of flood area extension for different return periods, under the condition of possible failure of the pumping system in the drainage system (Arosio et al., 2018). Note that a detailed hazard analysis is not the main goal of this article; therefore, the adopted flood modelling approach does not intend to be as detailed as possible, but instead to represent an adequate compromise between accuracy and simplicity. The hydrological and hydraulic simulations are based on EPASWMM (Rossman, 2015) and implemented on the primary deep drainage system

(almost 200 km of network, 14 main channels and 108 manholes). As for the rainfall, patterns associated with different return periods were obtained through the uniform Intensity Duration Frequency curve (IDF) for the entire MCFD (Amaro, 2005). In particular, Chicago hyetographs with a duration of 6 hours (Artina et al., 1997) and an intensity peak at 2.1 hours were constructed starting from the IDF curve. For each return period, the flooded areas are computed based on the volume spilled out of each of the main manholes of the drainage system. For each drainage catchment, assumed hydraulically independent from the others, a water depth-area relationship extracted from the DTM is used to compute the flood extension and depth. Figure 14a shows the flooded areas for a return period of 100 years. The majority of water depth values are between 0 to 1 m (lighter blues), and only a few raster cells (darker blue) have higher values that reach up to 9.83 m in some low-lying areas.

Some provider elements are located within the flood area, as seen in Figure 14a. These elements provide services to other elements located both inside and outside flooded areas, as shown in Figure 14b. Even if some of these receiver elements are not directly damaged, they can potentially experience indirect consequences due to the reduction or interruption of services from the providers that are directly affected. Using the hub analysis between the providers that are flooded, it is possible to identify the nodes that have more central role and can generate a potentially larger cascade effect for this flood scenario.

Figure 15 shows the values of hub values between the 17 providers inside the flood area. By integrating the information of the hazard scenario (i.e. flood area for specific a return period) with the hub and authority analysis of the network, it is possible to qualitatively assess that the red zone of the city has a relatively higher risk compared with the rest of the city. This zone is characterized by few providers with high hub values, which serve many blocks that have high values of authority as a result. This result shows the need for new additional providers in the red zone around the flooded area, in order to reduce the flood impact. As a matter of fact, this would reduce the number of receivers per provider, reducing the hub values of flooded providers. Consequently, the number of affected Blocks outside the flood footprint would be reduced.

For this pilot study, the estimation of the direct impact of the nodes is obtained adopting simplified binary vulnerability functions. According to this assumption, it then occurs zero damage in case of no flood, full damage in case of flood regardless of its intensity (vulnerability of level I), impacted nodes fully lose their capacity to provide services (vulnerability of level II), receiver elements are fully affected when even a single service is dismissed (vulnerability of level III). Despite of the availability of many vulnerability functions, for the purpose of this study we prefer to adopt such as simplified assumption since it does not affect conceptually the correctness of the process. As a matter of fact, the focus here is on the mechanism of the propagation of the impacts through the graph rather than the correct quantification of

them. Thus, the cascading effects are propagated through the graph by accounting for the nodes indirectly impacted i.e. those that have lost at least one service from their providers. By using the graph properties this task is straightforward. A new graph ($G_1$) is generated by removing the nodes directly impacted by the flood from the original graph ($G_0$). After that, the degree-in of each node in $G_1$, representing the number of incoming services, is compared with the corresponding degree-in in $G_0$. All the nodes with a reduction of degree-in are removed and a new graph $G_2$ is generated. This process is repeated until there are not more affected nodes in the graph and we obtain the final graph with the maximum impact extension that can be compared with the original graph.

Figure 16 shows the number of directly (blue) and indirectly (red) impacted elements due to the 100 year return period flood. The total number of elements affected is about 31,000, respectively more than 4,000 directly and almost 27,000 indirectly affected. These results, even acknowledging the relevance of the hypothesis adopted (i.e. no services redundancy and binomial vulnerability function), shows that indirect damages represent a significant part of the total damages. Furthermore, in Figure 16 the hypothetical direct and indirect impact curves are also plotted for illustrative purpose, as they could result by computing the results for other return periods.

The adoption of the graph adds to the traditional reductionist risk assessment the opportunity to explore the loss not only also in term of services and not only in term of elements. In fact, comparing the original graph ($G_0$) with the final graph obtained after the impact propagation, the approach allows to compute also the services lost. Figure 17 shows the number of services lost after the impact propagation separated within the categories of the elements and between the services lost due to the dismissal of providers (brown) and receivers (green) nodes. In terms of nodes there is no difference between those affected because of loss of received service and those affected because of loss of demanded service. Instead, in terms of services (i.e. links) there is a difference between those dismissed because of loss of a provider and those dismissed because of loss of a receiver. This difference can be important to evaluate the relative importance of these two different cases.

We acknowledge that these results are affected by the two important assumptions highlighted above and also due to the fact that the services are provided only by the elements inside the MCFD (as elements outside this area are not considered). Changing these assumptions could result in different cascading impacts. Regardless, the framework illustrated here shows the potentiality to quantitatively assess indirect impacts, which can subsequently be integrated into collective risk assessments.

## 4. Discussion and final considerations

In this paper we have tried to look at the problem of risk assessment of natural hazards in a holistic perspective, focussing on the "system" as a whole. We used "system" as a general term to identify the set of the different entities, assets, parts of a mechanism connected among each other in order to operate as, for instance, an organism, an organization, a city. Most of such systems are complex because of the high number of elements and the large variety of connections linking them. Nevertheless, our society is structured in these complex systems which are widespread everywhere. How can the risk of such complex systems be assessed? We believe that a reductionist approach that separates the parts of a system, computes the risk (losses, impacts, etc.) for each of them, and then sums them up to come out with a total estimate of risk is not adequate. Most of the research on natural hazards and their risk adopted implicitly the reductionist approach (i.e. "split the problem in small parts and solve it"). However, we mentioned also an emerging literature which adopts a different approach ("keep the system as a whole"), a holistic approach.

How can the system be represented as a single, intact, entire entity? And how can all the connections of its parts be represented? We believe, as other authors do, that the best approximation to represent a complex system is the graph. Many authors have already used the graph to model systems already organized as networks by construction (e.g. electric power network) and assess the risk of natural hazards in such a manner. Fewer authors have used the graph to model systems not immediately and manifestly depicted as physical networks and proceed in this manner to model the risk. Once the effort to "translate" a system with all its components into a graph is made, there are several advantages and benefits.

First of all, there is a mature theory of mathematics, the Graph Theory, that already studies the properties of a graph. Are these graph properties telling us something useful to assess the risk of natural hazards affecting these complex systems? We showed that some of the graph properties can disclose some relevant characteristics of the system related to the risk assessment. What is the vulnerability and exposure of the system? We proposed new analogies between some graph properties such as authority, hub, betweenness and degree-out values and the "systemic" exposure and vulnerability. The adoption of these analogies is supported by the recent work published by Clark-Ginsberg et al. (2018): despite having a different scope, they also use certain graph properties to assess the hazards of the companies operating in the case study, and promote a network representation of the risk. In section 2.2 and 3.2 we highlight the importance, before to quantify the risk, to look the single risk components from the systemic lens provided by the graph properties. This information could support more informed DRR decision making by strategically suggest how to prioritize intervention in order to minimize exposure and vulnerability from a system point of view.

A second advantage is that the graph can be used as a tool to propagate the impacts throughout the system from wherever the hazard hit it, including indirect or cascading effects. The links between nodes allow passing from the direct physical damage to broader economic and social indirect impacts. The indirect impact suffered by a certain node may be defined
as a function of two factors: 1) the direct damage sustained by one or more of its parent nodes (i.e. traditional impact); 2) the loss of service the latter provide to the former (i.e. vulnerability function). The integration of indirect impact quantification within the graph-based framework has been addressed in the pilot study using a simplified binary vulnerability function.

Despite the two advantages in adopting this systems perspective in risk assessment, Clark-Ginsberg et al. (2018)
highlights that there are "questions about the validity of such assessment" regarding the ontological foundations of networked risk, the non-linearity and emergent phenomena that characterize system phenomena. In fact, the emergence of the risk system demonstrates that the risk will never be completely knowable, and for this reason the "unknown unknowns are an inseparable part of a risk networks"; in fact, the boundary definition of open systems is by nature artificial.

The application to the case of urban flooding in Mexico City it is a first attempt to demonstrate the feasibility of the proposed approach and it is also the first example in literature that try to quantitatively analyse the propagation of impact into a network of individual elements that do not explicitly constitute a network. In this study, the complexity of Mexico City is depicted by modelling certain selected typologies of elements of the urban system and by assuming simplified rules of connection between them. Furthermore, the system complexity acknowledged in this study is restricted to the
elements inside the MCFD and neglects any potential contributions from outside elements. The definition of a geographical boundary condition, which is a straightforward assumption in the traditional reduction approaches, can be controversial in the holistic approaches that aims to model the emergent characteristics of open-ended systems. However, the flexibility of our approach allows for a graph to be designed with any intended level of detail, depending on the purpose of each specific application and the availability of data. For instance, if a more comprehensive characterization
of the road network was required, the graph could be expanded to include additional elements other than the major Crossroads. Another example regards the rules of connections adopted in this study, which do not allow for redundancy, as each node is considered to receive its services from the nearest provider only. A more detailed graph could include, for example, influencing areas for each service, which would allow considering multiple providers for some of them, provided that the required data were available. Adopting different rules (e.g. a provider could deliver its service to as many elements
as inside a defined distance) would allow a degree of redundancy of the network, which could significantly change the impact of a hazardous event. We have adopted a simple flood scenario to illustrate how some of the measures of a graph

can be used in the context of natural hazard risk assessment. However, within our framework, additional potentially relevant information can be obtained. For example, here we have presented the results of the structural analysis of the graph without looking into functional properties such as the percolation threshold, which characterizes the resilience of a network and can therefore provide valuable information for practical applications. Another possible extension consists in studying how the network evolves with time following external perturbations at different return periods.

Furthermore, the proposed approach could introduce a common base for future research on both multi-hazard and integrated risk assessment. Being the graph properties hazard independent, it is possible to integrate these properties with the characteristics of the single node, such as the physical vulnerability of a building with respect to earthquake or flooding (adopted by reductionist approaches) and analyse multi-hazard using the same graph. Besides, the use of this approach can be applied to physical as well as social or integrated risk. In the former case, the graph has only physical elements (e.g. buildings), the latter case the graph has nodes that reflect also social aspects (e.g. population, age, education, etc.).

Further research will aim to fully implement and integrate the graph-based approach in quantitative risk assessments, both at scenario and probabilistic level. One of the challenges that will need to be addressed is related with data requirements and availability. Currently, most exposure and vulnerability databases focus on the properties of single elements, and tend to contain little to no information on the connections between them. As we have discussed, this information is key for more thoroughly understanding and assessing the risk of a system. For this reason, developing and collecting data with information related to the connections between the elements is paramount. To promote this perspective, it is necessary consider shifting the RA from using traditional relational databases to so-called graph databases. In such databases, each node contains, further to the traditional characteristics, also a list of relationship records which represent its connections with other nodes. The information on these links is organized by type and direction, and may hold additional attributes.

Finally, the introduction of the graph-based approach into the RA for collective disaster risk aims, in the long term, to be a first step for future developments of Agent Based Models and Complex Adaptive Systems in collective risk assessment. In this perspective, the nodes of the network are agents, with defined state (e.g. level of damage), and the interaction between the other agents is controlled by specific rules (e.g. vulnerability and functional functions) inside the environment within they live (e.g. natural hazard phenomena).

**Acknowledge**

This research was partly funded by Fondazione Cariplo under the project "NEWFRAME: NEtWork-based Flood Risk Assessment and Management of Emergencies" and it has been developed within the framework of the project "Dipartimenti di Eccellenza", funded by the Italian Ministry of Education, University and Research at IUSS Pavia.

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

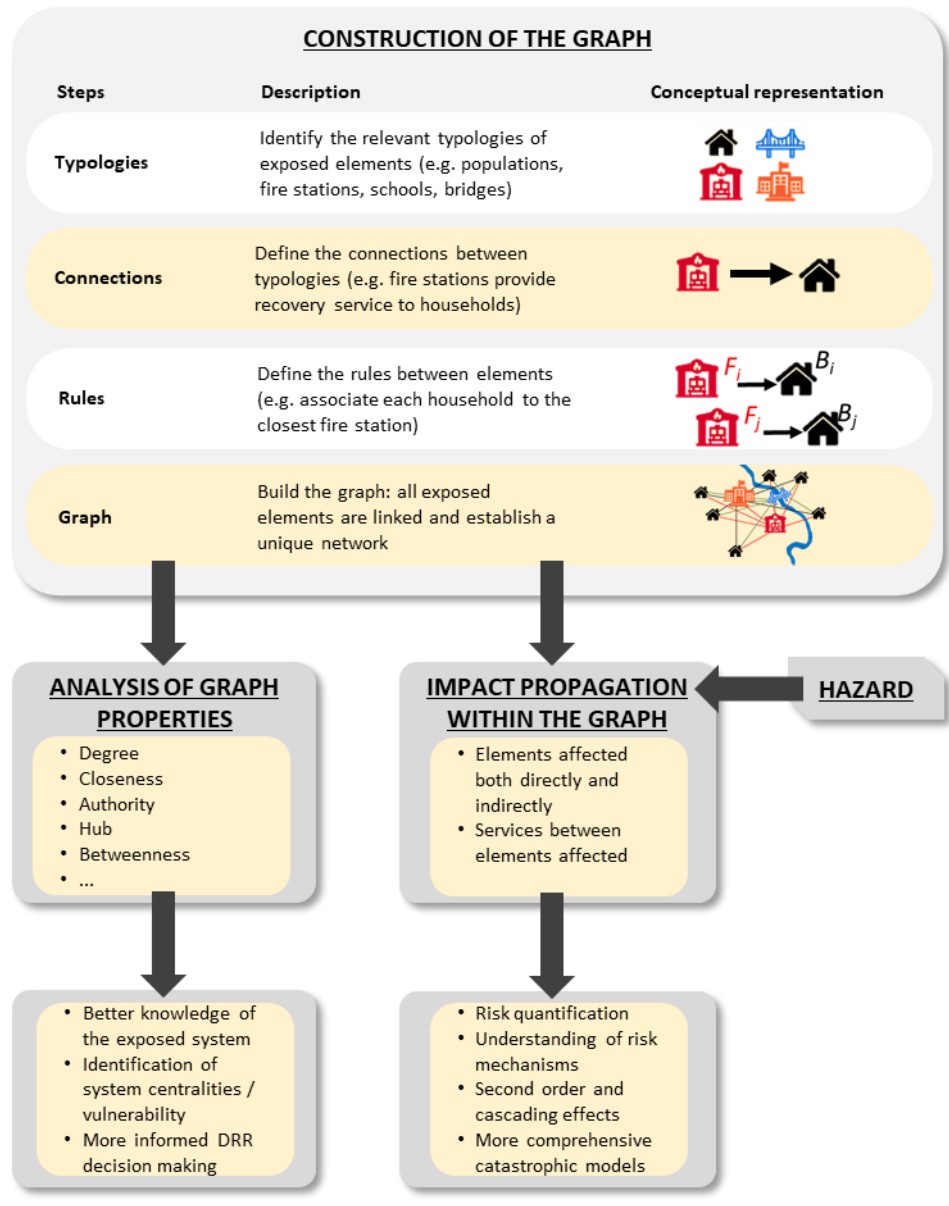


**Figure 1: Workflow.**

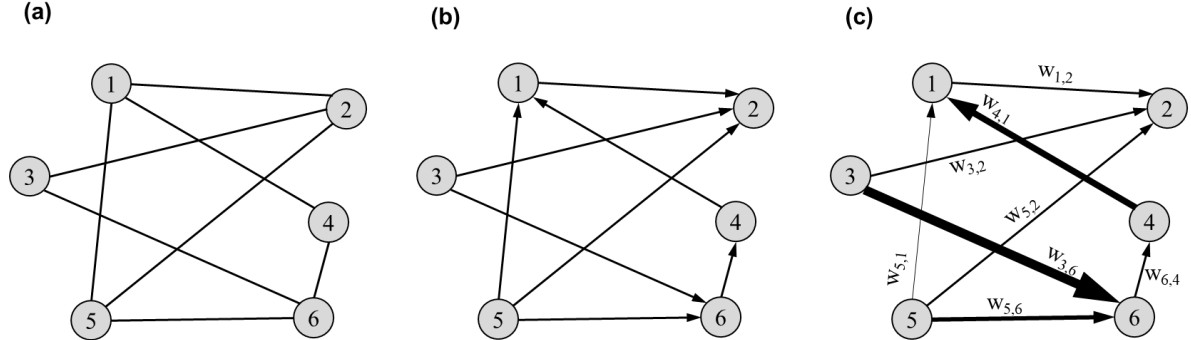

**Figure 2: Graph representation of a network. (a) Undirected. (b) Directed. (c) Weighted directed.**

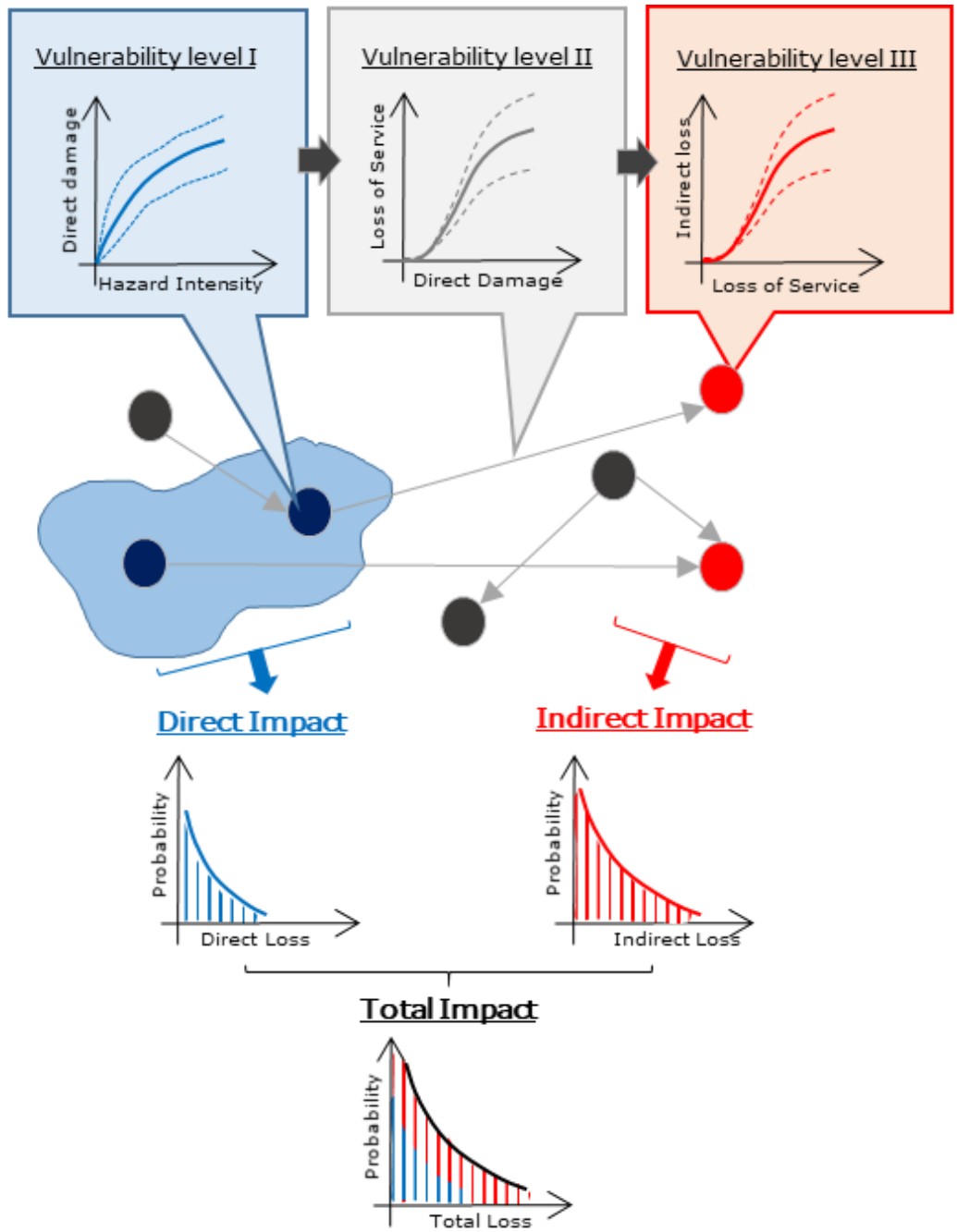

**Figure 3: Risk Framework**

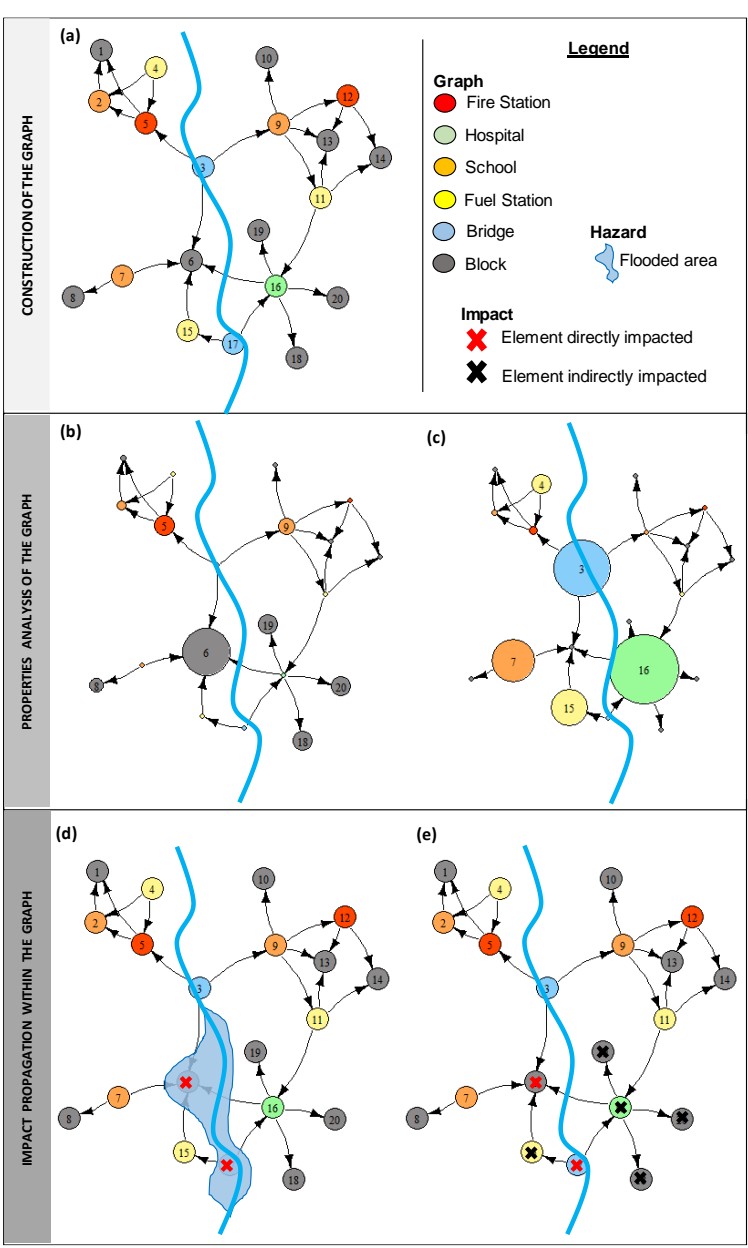

**Figure 4: (a) Map of the various elements of a hypothetical municipality in a flood-prone area; (b) Same, with node sizes proportional to authority values; (c) Same, with node sizes proportional to hub values; d) Same, with flood area and nodes directly impacted highlighted with in red cross; (e) Same, with also the nodes indirectly impacted highlighted with black cross.**


| Network conceptualization | Type of nodes | Number of elements | Service Provided | Destination of the service |
|---|---|---|---|---|
|  | Crossroads | 17 | Transportation | Fire Stations, Schools, Hospitals, Fuel Stations , Blocks |
| | Fire Stations | 11 | Recovery | Schools, Hospitals, Fuel Stations, Blocks |
| | Fuel Stations | 103 | Power | Blocks |
| | Hospitals | 39 | Healthcare | Blocks |
| | Schools | 130 | Education | Blocks |
| | Blocks | 64,282 | - | - |

**Figure 5: List of nodes adopted in the network conceptualization.**

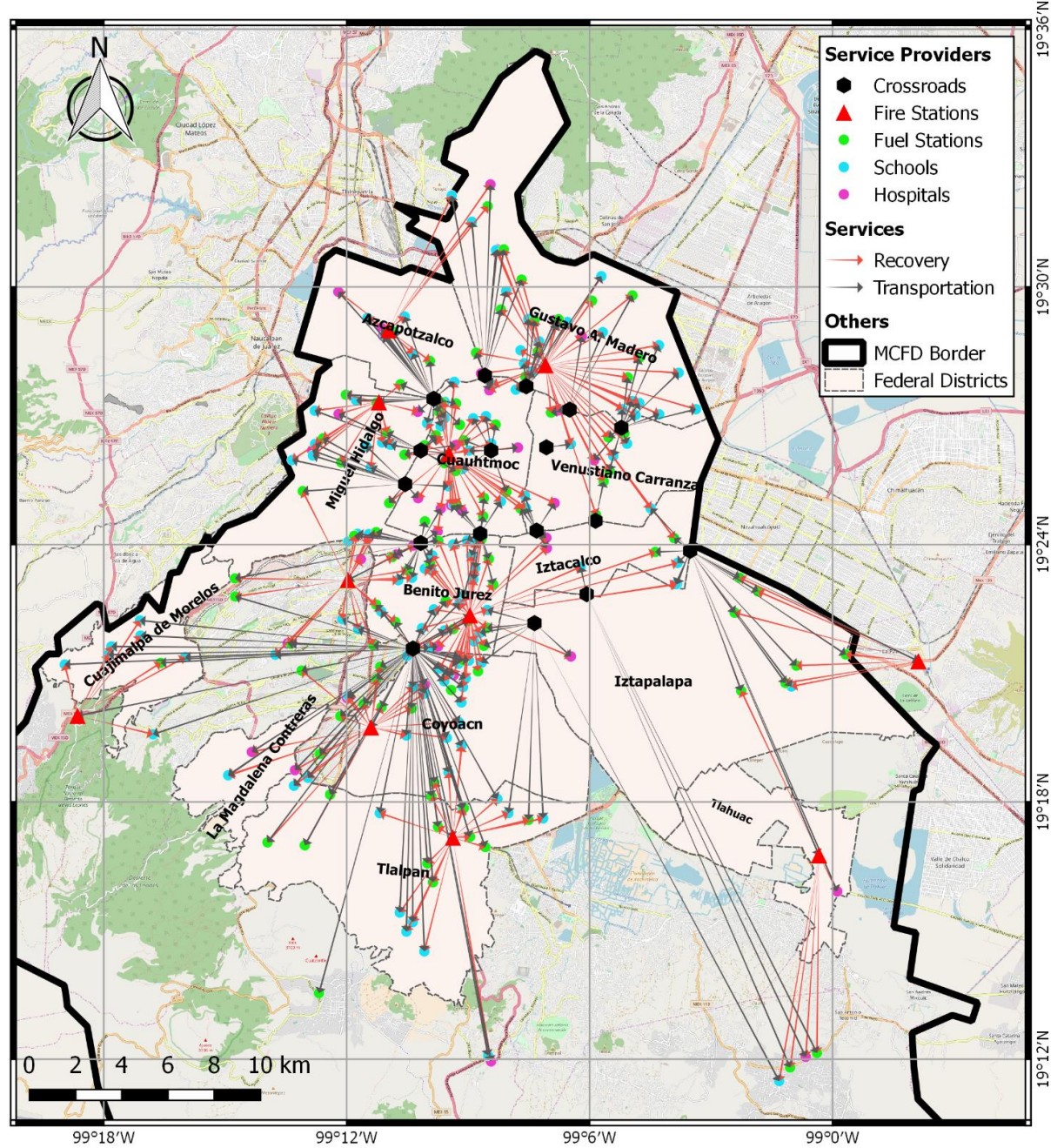

**Figure 6: Map of nodes and services provided among them. For readability, Blocks are not included (© *OpenStreetMap contributors 2019. Distributed under a Creative Commons BY-SA License.*).**

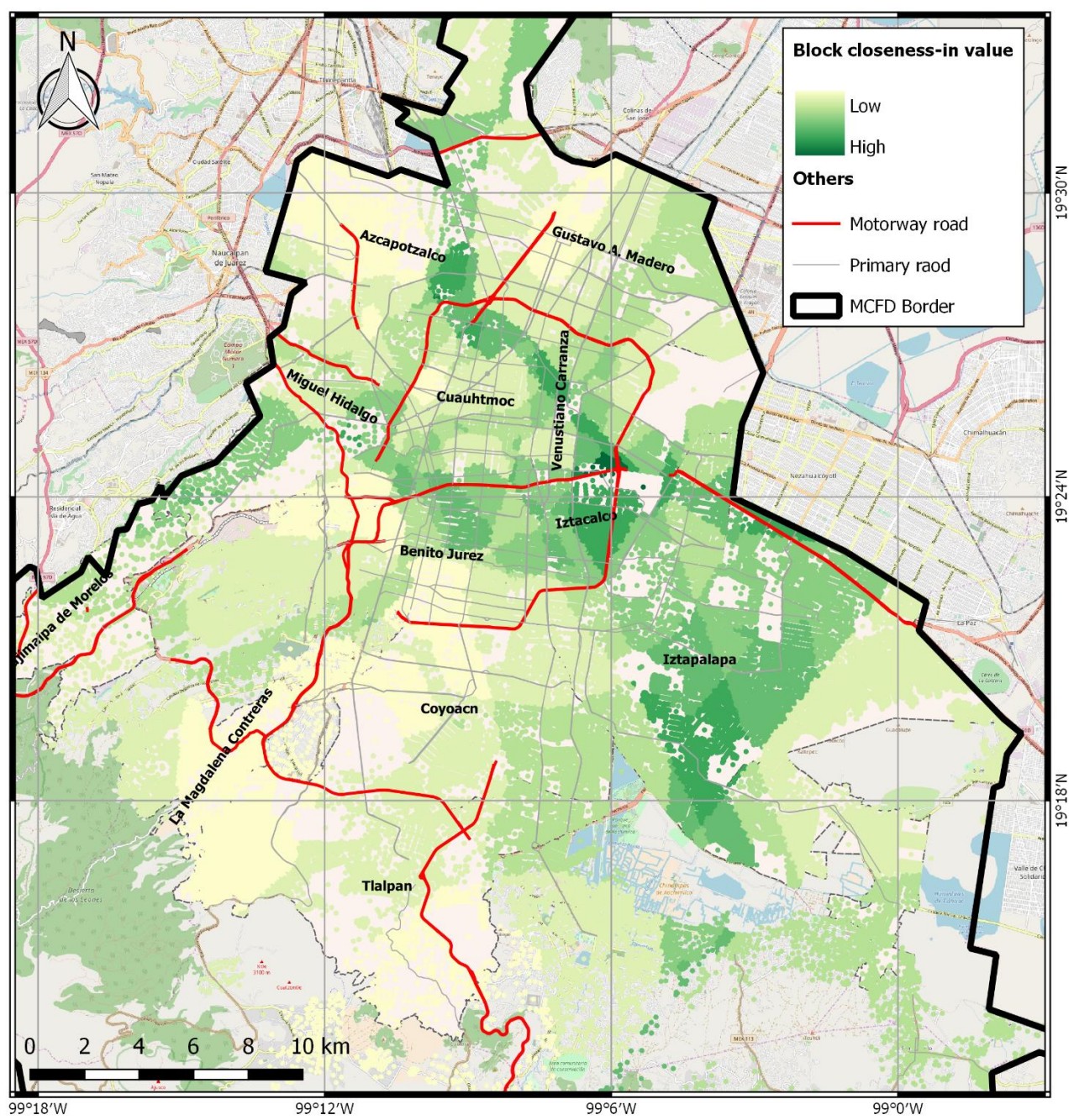

**Figure 7: Geographical distribution of the Block closeness-in value** (*© OpenStreetMap contributors 2019. Distributed under a Creative Commons BY-SA License.*).


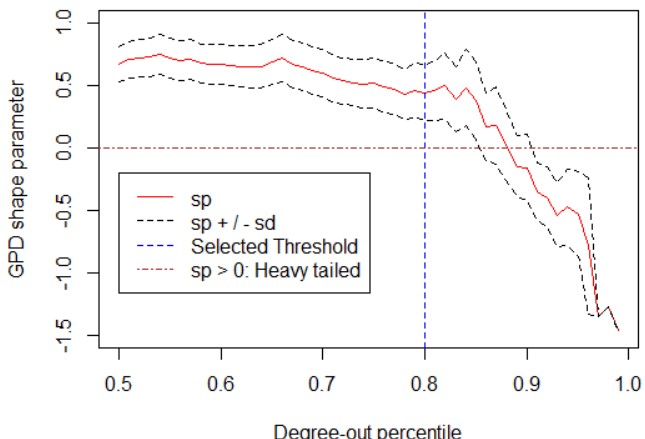

**Figure 8: Parameter estimation (sp) against thresholds for degree-out data (sd: standard deviation).**

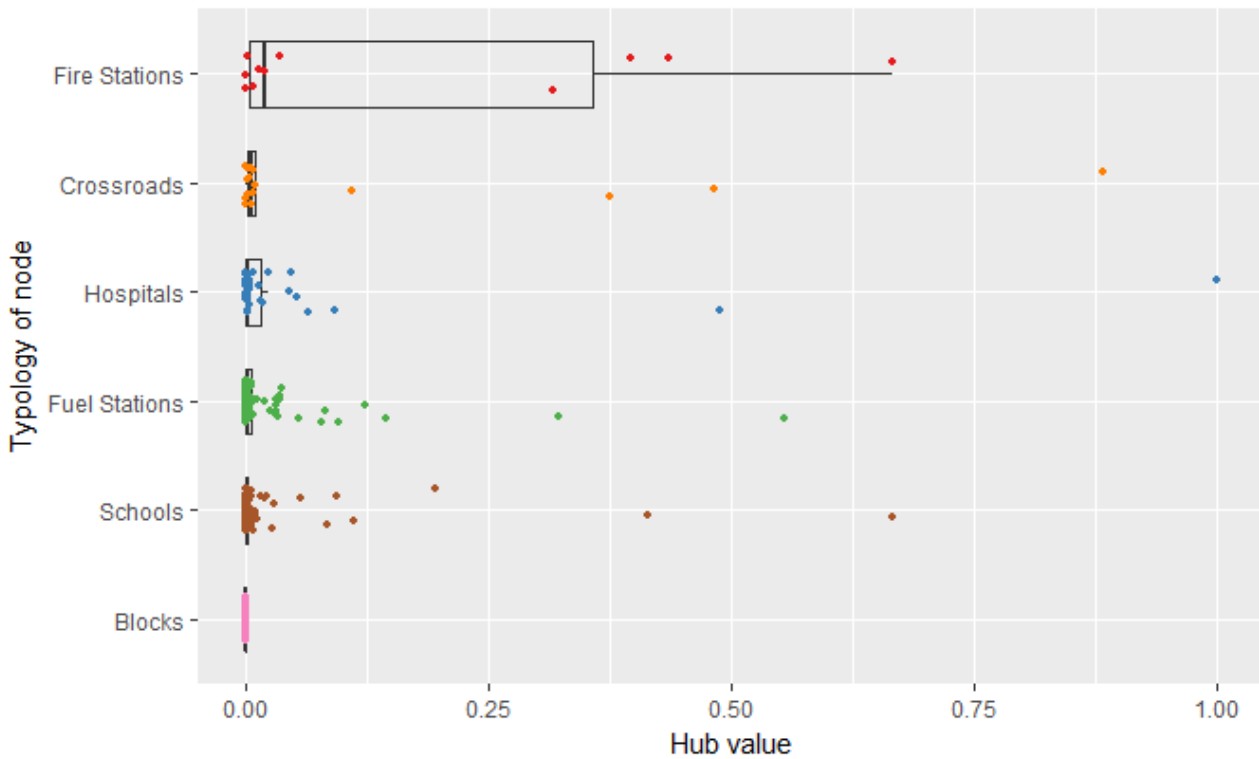

**Figure 9: Boxplots of hub value for different typologies of service providers.**

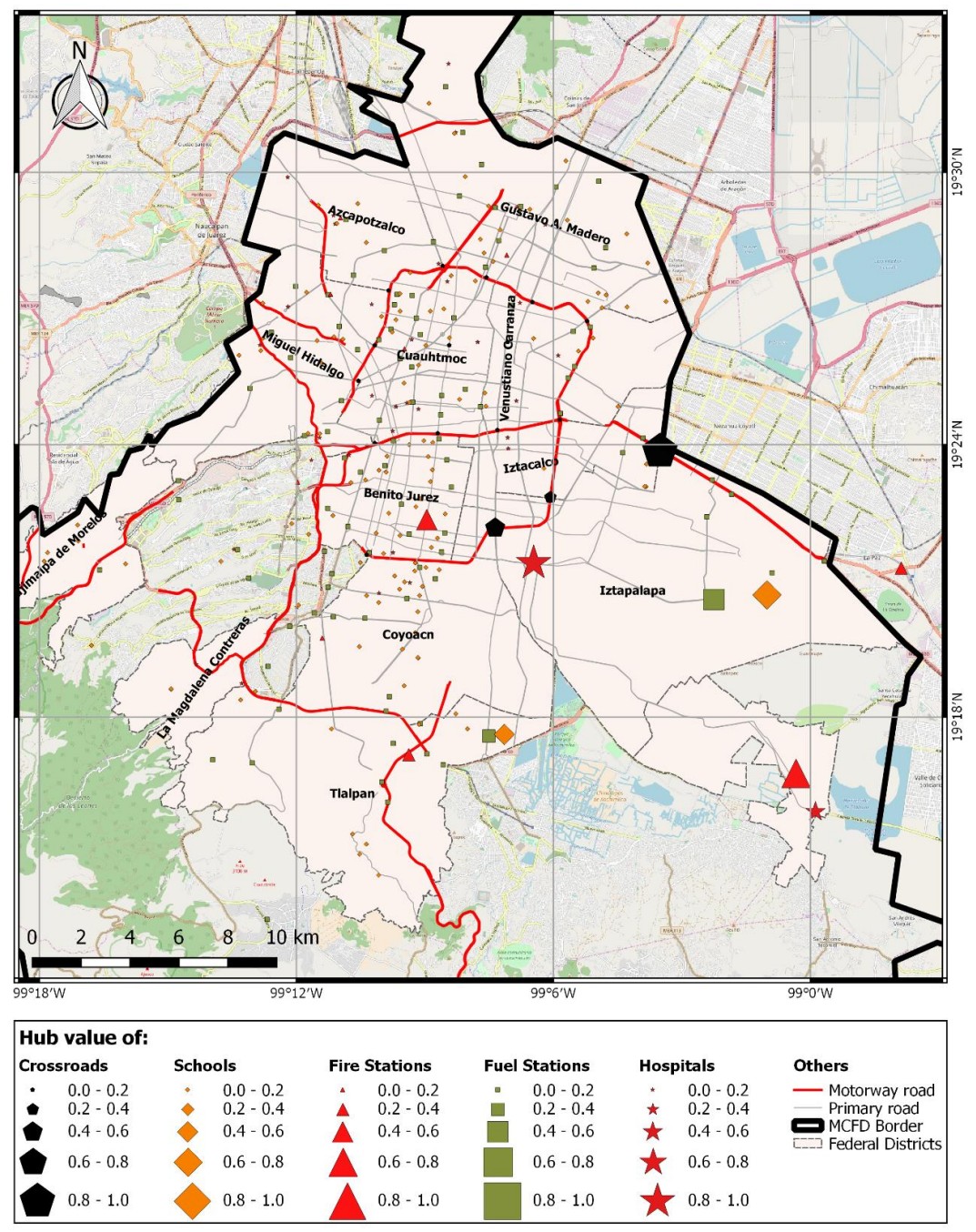

**Figure 10: Map of providers. Icon dimensions are proportional to the hub values (©** *OpenStreetMap contributors 2019.*
*Distributed under a Creative Commons BY-SA License.***).**

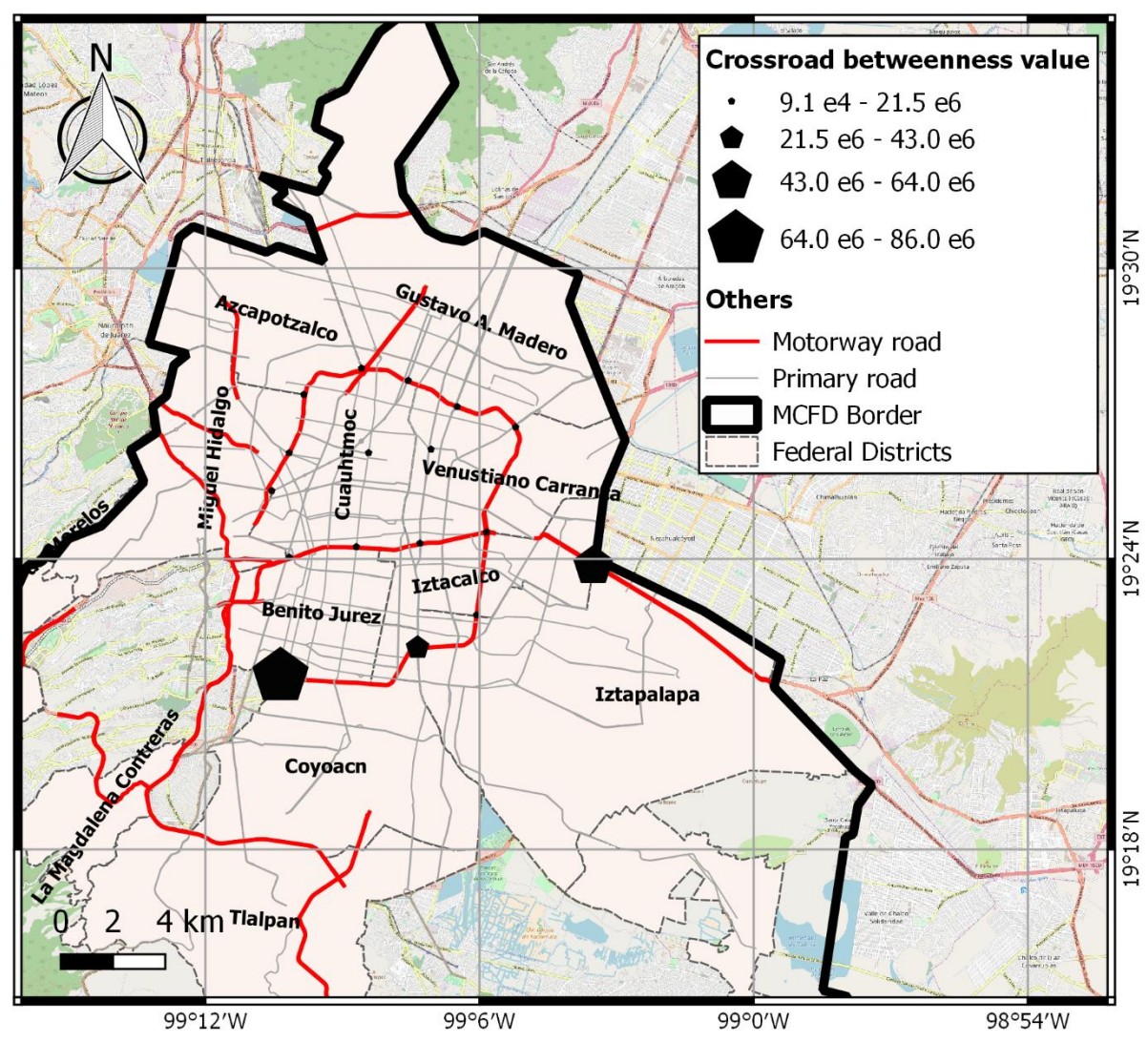


**Figure 11: Map of Crossroads. Icon dimensions are proportional to the betweenness values** (*© OpenStreetMap contributors 2019. Distributed under a Creative Commons BY-SA License.*).


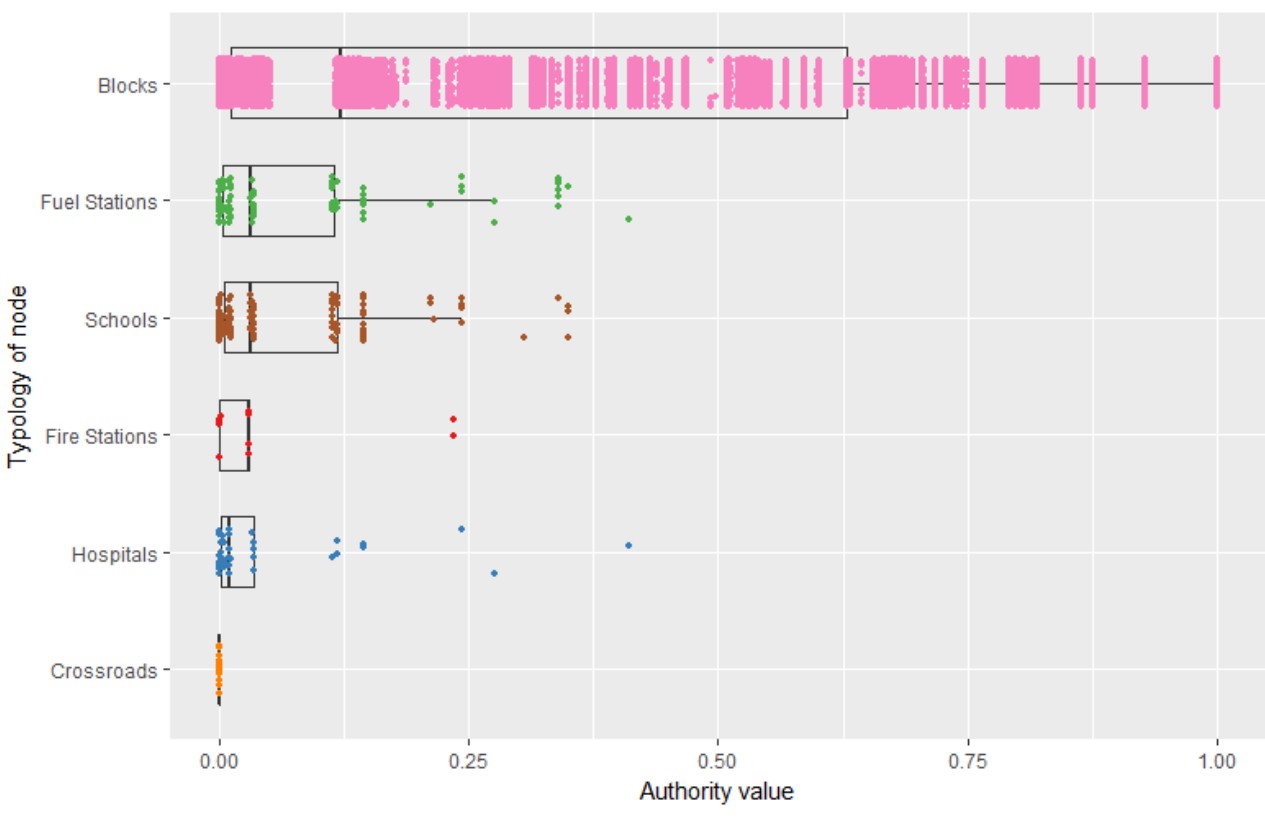

**Figure 12: Boxplot of authority values for different provider services.**

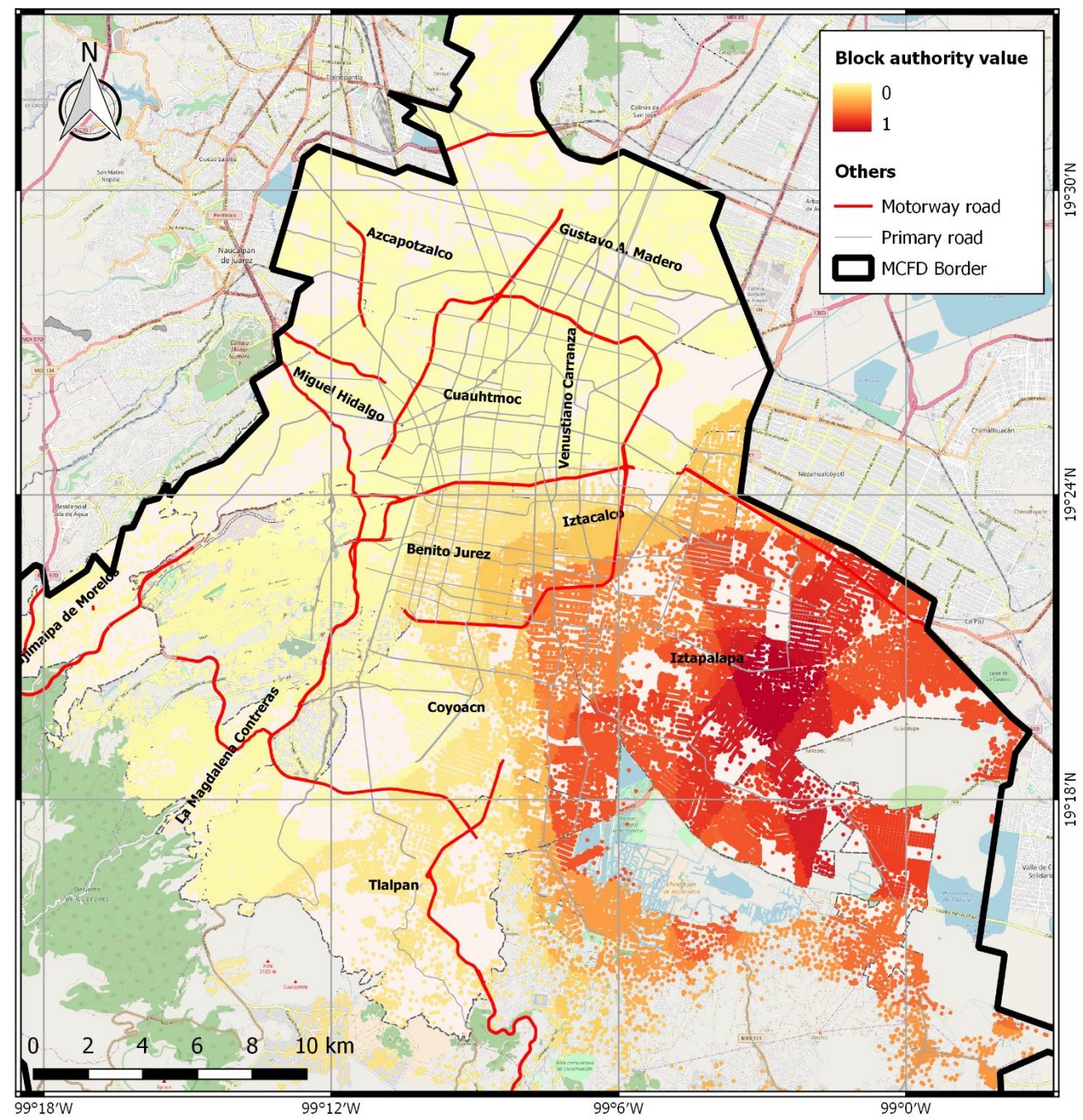

**Figure 13: Geographical distribution of the Block authority value (©** *OpenStreetMap contributors 2019. Distributed under a Creative Commons BY-SA License.***).**


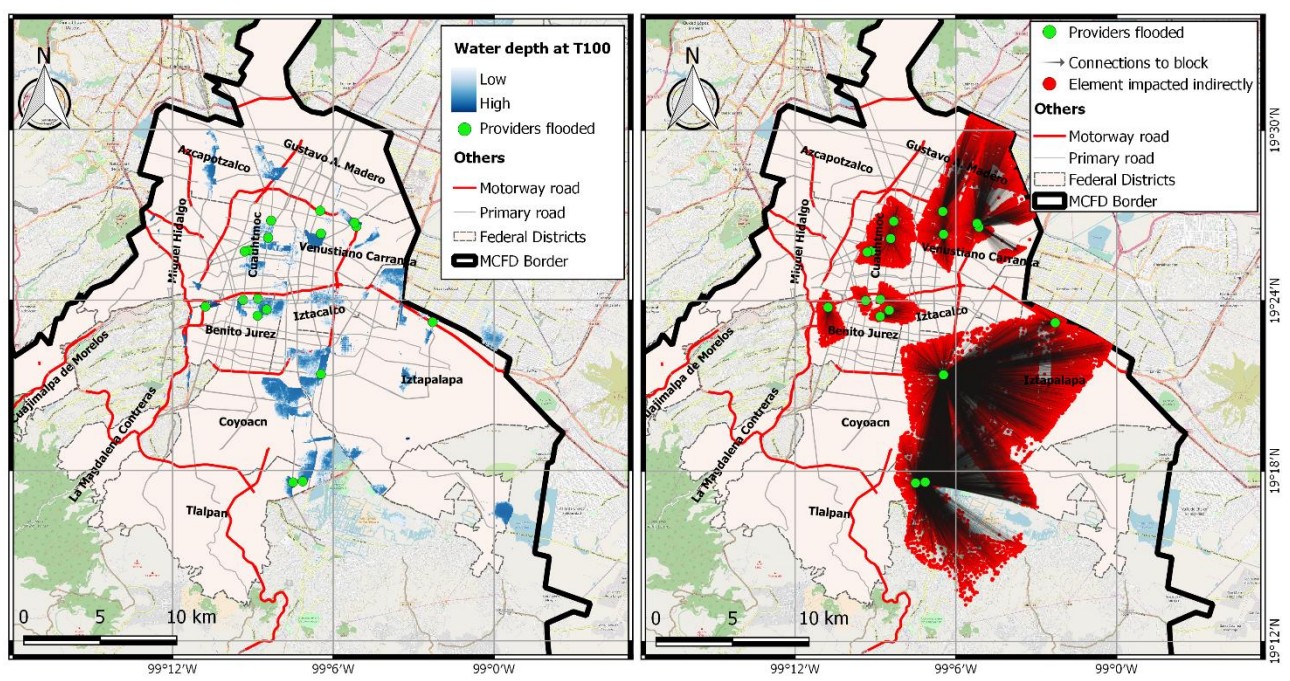

**Figure 14: a) Flooded area for T100 and flooded providers; b) blocks connected to the flooded providers (©** *OpenStreetMap* **contributors 2019. Distributed under a Creative Commons BY-SA License.).**

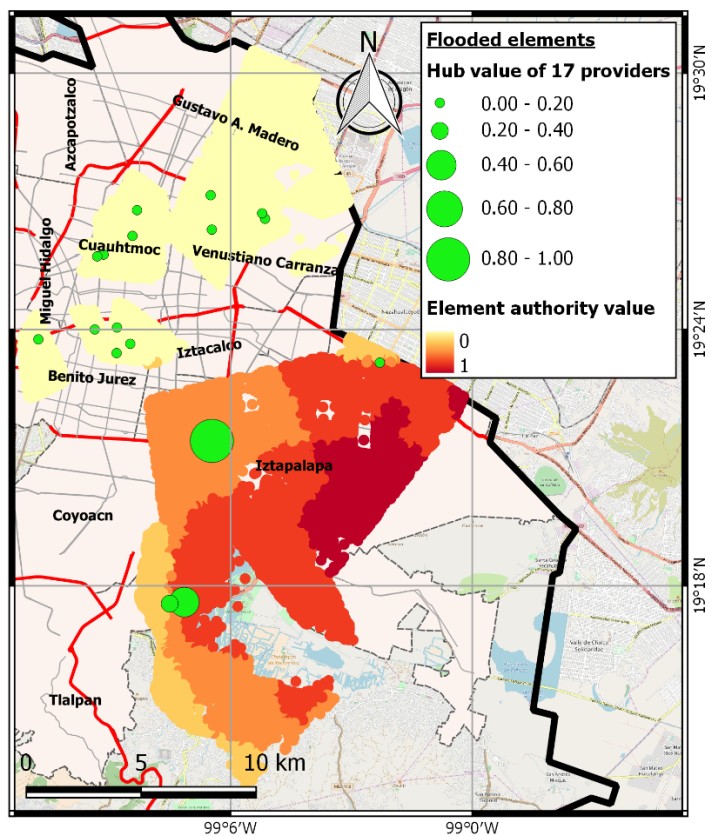

**Figure 15: Hub and authority values of flooded nodes (©** *OpenStreetMap contributors 2019. Distributed under a Creative Commons BY-SA License.***).**

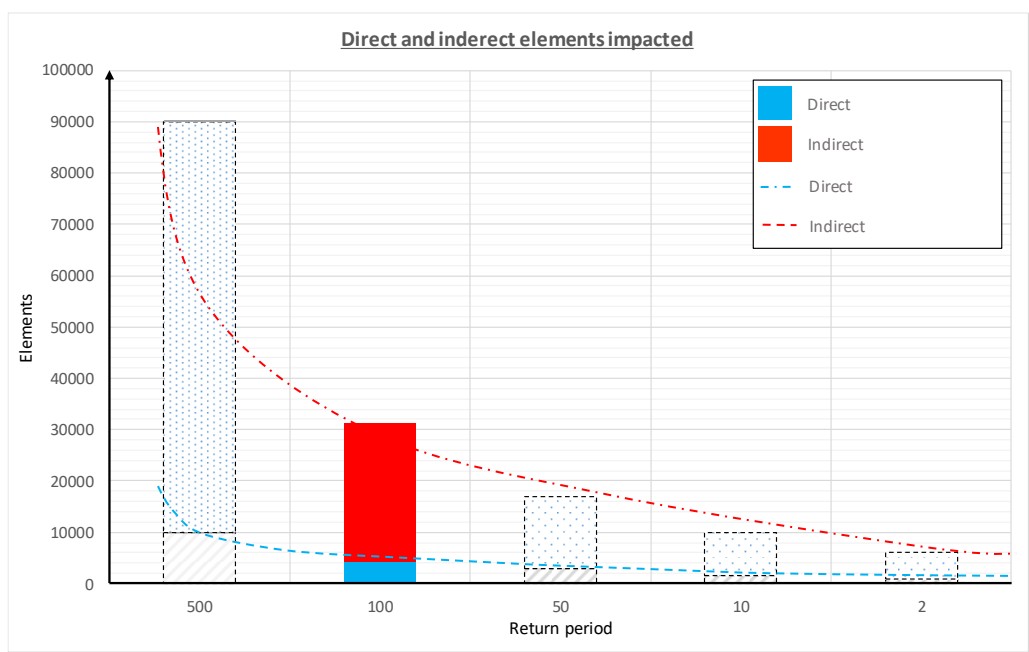

**Figure 16: The full coloured bar reports the computed direct and indirect impacted elements at T=100 years, shadow bars represent conceptually the impacts for other return period to visualize the complete risk curve**


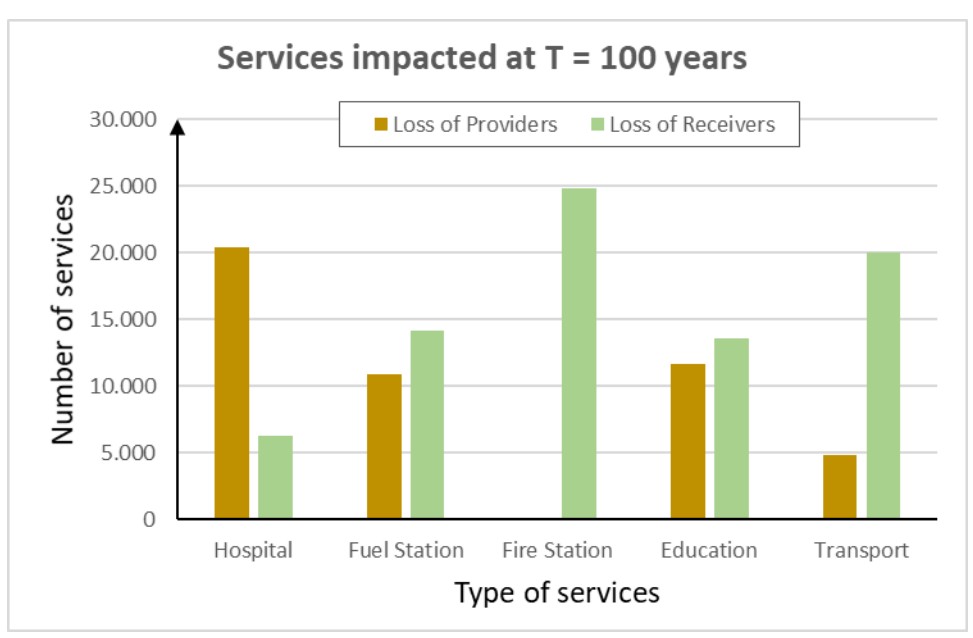

**Figure 17: Services impacted at T = 100 years.**

 **Table 1: Properties of a graph G with N nodes defined by its adjacency matrix $A(G)$ with $N \times N$ elements $a_{ij}$, whose value is $a_{ij}>0$ if nodes $i$ and $j$ are connected, and 0 otherwise**

| Property | Description | Formula |
|---|---|---|
| Degree (k) | The number of edges incident with the node | $$k_i = \sum_j a_{ij}$$ |
| Diameter (D) | The maximum value of all path lengths $d_{ij}$ | $$D = \max_{i,j} d_{ij},$$ where $d_{ij}$ is the geodesic length from node $i$ to node $j$ (*i.e. path length*): |
| Characteristic path length (d) | The average shortest path length | $$d = \frac{1}{N*(N-1)} * \Sigma_{i,j(i \neq j)} d_{ij}$$ |
| Closeness (c) | Shortest path length from a node to every other nodes in the network | $$c_i = \frac{1}{l_i}, \quad where \; l_i = \frac{1}{n-1} * \sum_j d_{i,j}$$ |
| Betweenness (b) | Number of shortest paths between pairs of nodes that pass through a given node | $$b_i = \sum_{j,k} \frac{n.\ of\ shortest\ paths\ connecting\ j,k\ via\ i}{n.\ of\ shortest\ paths\ connecting\ j,k}$$ $$= \sum_{j,k} \frac{n_{jk}(i)}{n_{jk}}$$ |
| Authority (x) | The value proportional to the sum of the node hub values pointing to it | $x_i = \alpha * \sum_j a_{ji} y_j$ -> $A * A^T$, where α is a proportional constant |
| Hub (y) | The value proportional to the sum of authority of nodes pointing to it | $y_i = \beta * \sum_j a_{ij} x_j$ -> $A^T * A$, where β is a proportional constant |
| Percolation threshold (pc) | The minimum value of fraction of remaining nodes (p) that leads to the connectivity phase of the graph | For random graph $p_c = \frac{1}{\bar{k}}$, $\bar{k}$ is the average of degree |

**Table 2: Analogy of risk variables with graph properties.**

| Risk variables | Analogy with graph properties |
|---|---|
| Exposure | The authority represents how the system privileges the nodes, conferring them more or less importance compared with others, according to the connections established in the system. |
| Vulnerability | The propensity of parts of the network to be isolated because of hazard events. The closeness of a node is a measure of the single node vulnerability within the system, while degree distribution, hub, and betweenness are measures of vulnerability of the system as a whole. |
| Resilience | The percolation threshold, together with the network fragmentation analysis, explain the resilience of the network after a perturbation. |