# Peer review of "The whole is greater than the sum of its parts: a holistic graphbased assessment approach for natural hazard risk of complex systems"

_Natural Hazards and Earth System Sciences, 2018_

## Referee Comment (RC1) · Anonymous Referee #1 · 23 Oct 2018

Natural hazard risk of complex systems part I introduces graph theory into risk analysis to promote a paradigm shift from reductive to holistic approaches to risk assessment and assess the risk of complex systems. Through a review of graph theory as it relates to risk, including issues of exposure, vulnerability, and resilience, and the development of an illustrative case, the authors show how network analysis can be employed to assess complex interdependent systems. The authors' main argument is that current risk assessment approaches fail to capture complex interactions between systems as a whole, and that network analysis techniques can be used to capture that complexity.

The authors are correct that current risk assessments are often reductionist and fail to

account for interconnections and the properties of the system as a whole. Readers will also benefit from this topic given the prevalence of risk analysis that take a reductionist perspective. However, there is a significant body of work using graph theory for risk analysis. A large literature builds on Rinaldi et al. (2001) to use graph theory to assess critical infrastructure risks, interdependencies, and cascades (Lewis 2014; Setola et al. 2016), and another focuses on the systemic risks in financial systems (Summer 2013) Instead of focusing primarily on the connections between physical structures of infrastructure, another body of work focuses on the interconnections between hazards or hazards and vulnerabilities showing how risks can propagate and cascade (Clark-Ginsberg 2017; Gill and Malamud 2014).

This literature (and the broader qualitative literature on networks of risk) identifies several challenges with using network analysis for risk. Chief among them is how to account for the multi-level, open-ended nature of systems in graph based approaches. For instance, Schulman and Roe (2016) and other high reliability theorists point out that infrastructure systems are vastly more complex than modelers make them out to be, with substantial coupling across components that is difficult to discern. Clark-Ginsberg et al. (2018) applies these insights to argue that network bases approaches of open-ended systems can never be complete and require careful decisions on how to delimit boundaries and describe networks. The authors allude to the idea of system incompleteness when discussing the nestled nature of power infrastructure, but then purport to offer a complete network (p6), which is not possible given the open-ended nature of risk.

This literature shows how graph theory can be used for representing complex issues of risk in a holistic way and also provides a grounding in some of the challenges associated with the topic. The authors need to clearly state how their work contributes to this literature. Because they do not engage with this literature I do not believe there is enough for a standalone theoretical paper on their topic. Rather than publishing this as a separate piece, I recommend using this article as a basis the literature review/methodology of the empirical paper, which provides a useful contribution to the literature.

---

## Referee Comment (RC2) · Anonymous Referee #2 · 12 Dec 2018

Summary:

This paper only takes us about 60% of the way there. While I do think you have a novel idea of using graph theory to model risk transfer in a way that has not been done, you don't fully show us how to do it conceptually.

e.g. you explain how graphs work. and give some discussion of how these graph properties link to vulnerability, resilience, and exposure. But you need to go much further. What metrics do you propose we use from graph theory that link to which metrics in risk assessment? Maybe this is what you are trying to do with percolation but it is still very unclear. How are you going to get us towards measure cascading

risks with your new approach.

Part of the problem is the disorganized literature review and background. You are missing a lot of the resilience literature on this topic, and it feels like you are describing papers selectively. Please organize this into topics, themes, that lead to the demonstrating the gap in the lit that your new graph theory approach will allow us to fill.

This idea has a lot of promise, but needs work. The conclusion should make me feel like I have a new tool and idea to measure risk. But I am left feeling confused.

Line 45. Vulnerability does consider social conditions. That is a wrong statement

Line 59. See the work on compound flood risk. Eg. Wahl, Thomas, Shaleen Jain, Jens Bender, Steven D. Meyers, and Mark E. Luther. "Increasing risk of compound flooding from storm surge and rainfall for major US cities." Nature Climate Change 5, no. 12 (2015): 1093. Zscheischler, J., Westra, S., Hurk, B.J., Seneviratne, S.I., Ward, P.J., Pitman, A., AghaKouchak, A., Bresch, D.N., Leonard, M., Wahl, T. and Zhang, X., 2018. Future climate risk from compound events. Nature Climate Change, p.1.

Line 80. Great examples. Surprise to see lack of citatations for the large literature on compounding risk and cascading failtures from the resilience field. E.g.

Buldyrev, S. V., Parshani, R., Paul, G., Stanley, H. E., & Havlin, S. (2010). Catastrophic cascade of failures in interdependent networks. Nature, 464(7291), 1025. Chicago Line 84. I have never heard of this rinaldi paper. I doubt it is the most quoted.

In general this literature review feels selective and disorganized. Use subheadings. What is the gap you are filling? Are you really the only/first people to use graph theory to assess risk. I somehow doubt it. A simple google scholar search revealed many articles:

Heckmann, T., Schwanghart, W., & Phillips, J. D. (2015). Graph theory—Recent developments of its application in geomorphology. Geomorphology, 243, 130-146. Holmgren, Åke J. "Using graph models to analyze the vulnerability of electric power

networks." Risk analysis 26, no. 4 (2006): 955-969. Lhomme, S., Serre, D., Diab, Y., & Laganier, R. (2013). Analyzing resilience of urban networks: a preliminary step towards more flood resilient cities. Natural hazards and earth system sciences, 13(2), 221-230.

Also see risk transfer analysis 1. Sapountzaki, K. Social resilience to environmental risks: A mechanism of vulnerability transfer? Manag. Environ. Qual. An Int. J. 18, 274–297 (2007).

Page 165. Its hard to read all your definitions in prose. Made a table or a diagram that shows in a depiction each term. Add more to figure 1.

Line 196. What is pc. What is k.

Line 265. It is not until here that you tell me what graph theory contributes to vulnerability analysis. WHY is current risk analysis lacking and WHAT does graphs uniquely help us understand.

330. I have never heard of this definition of resilience. This need to be motivated by the enourmous literature on the topic to some degree.

---

## Author Comment (AC1) · 23 Jan 2019

**Response to Referee #1**

We wish to thank the Referee for his/her time and effort reviewing the manuscript. We greatly appreciate the constructive comments and suggestions, which we have carefully addressed in this response. Where applicable, changes are proposed to the manuscript accordingly (and marked up for clarity). Following the guidelines of the NHESS Editorial Board, the revised manuscript was not prepared at this point.

*Natural hazard risk of complex systems part I introduces graph theory into risk analysis to promote a paradigm shift from reductive to holistic approaches to risk assessment and assess the risk of complex systems. Through a review of graph theory as it relates to risk, including issues of exposure, vulnerability, and resilience, and the development of an illustrative case, the authors show how network analysis can be employed to assess complex interdependent systems. The authors' main argument is that current risk assessment approaches fail to capture complex interactions between systems as a whole, and that network analysis techniques can be used to capture that complexity.*

*The authors are correct that current risk assessments are often reductionist and fail to account for interconnections and the properties of the system as a whole. Readers will also benefit from this topic given the prevalence of risk analysis that take a reductionist perspective.*

*- However, there is a significant body of work using graph theory for risk analysis. A large literature builds on Rinaldi et al. (2001) to use graph theory to assess critical infrastructure risks, interdependencies, and cascades (Lewis 2014; Setola et al. 2016), and another focuses on the systemic risks in financial systems (Summer 2013). Instead of focusing primarily on the connections between physical structures of infrastructure, another body of work focuses on the interconnections between hazards or hazards and vulnerabilities showing how risks can propagate and cascade (Clark- Ginsberg 2017; Gill and Malamud 2014).*

Graph Theory is a well-established branch of mathematics. As such, it has been used to address a wide number of problems in many different fields, where risk analysis is included. However, risk analysis is, in itself, a very large field. Natural hazard risk, despite falling under the 'risk analysis' umbrella, requires its own specific modelling approaches, which are necessarily different from other types of risk, such as financial contagion in banking systems (as covered by Summer (2013) mentioned by the Referee), or others like car accidents, disease, conflict, to name a few. As this paper focuses of natural hazard risk, we have engaged with literature primarily from this field, where the application of Graph Theory is much sparser. We agree that some of the references mentioned here by the Referee are relevant in this context, and were missing from the original manuscript. Accordingly, in the revised manuscript we will add Clark-Ginsberg *et al.* (2018) at L423, Lewis (2014) and Setola *et al.* (2016) at L91.

*- This literature (and the broader qualitative literature on networks of risk) identifies several challenges with using network analysis for risk. Chief among them is how to account for the multi-level, open-ended nature of systems in graph based approaches. For instance, Schulman and Roe (2016) and other high reliability theorists point out that infrastructure systems are vastly more complex than modelers make them out to be, with substantial coupling across components that is difficult to discern. Clark- Ginsberg et al. (2018) applies these insights to argue that network bases approaches of open-ended systems can never be complete and require careful decisions on how to delimit boundaries and describe networks. The authors allude to the idea of system incompleteness when discussing the nestled nature of power infrastructure, but then purport to offer a complete network (p6), which is not possible given the open-ended nature of risk.*

We agree that the issue of modelling open-ended systems is central in any study of networks, and that this aspect was not sufficiently discussed in the original manuscript. For this reason, following the suggested reference, we propose adding the following paragraph to the Discussion section (L423) in the revised manuscript:

> "Despite the improvements in risk assessment within this systems perspective, Clark-Ginsberg *et al.* (2018) highlights that there are "questions about the validity of such assessment" regarding the ontological foundations of networked risk, the non-linearity and emergent phenomena that characterize system phenomena. The emergence of the risk system demonstrates that the risk will never be completely knowable, and for this reason the "unknown unknowns" are an inseparable part of a risk networks; in fact, the boundary definition of open systems are by nature artificial."

We believe this issue should also be discussed in Section 2.2.1 (Network Conceptualization), and therefore propose to add the following sentence (L237):

> "In defining the topology, it is crucial to define the level of analysis details coherently with the scope and scale, both for the selection of elements and for the relationship between elements that need to be considered. In the case of a very high detail for example, a node of the graph could represent a single person within a population, and in the case of a lower resolution, it could represent a large group of people with a specific common characteristic, such as living

in the same block or having the same hobby. In the case of analyses at a coarser level, an entire network (e.g. electric power system) can be modelled as a single node of another larger network (e.g. national power system). The definition of the topology structure of the graph also identifies immediately the system boundaries (e.g. which hospitals to be considered in the analysis: only the potential flood area, the ones in the district or in the region?). Up to which extent it is necessary to consider elements as nodes of the graph? The topology definition is a necessary step to perform the computational analysis and introduces approximations of the open systems that need to be acknowledged."

*- This literature shows how graph theory can be used for representing complex issues of risk in a holistic way and also provides a grounding in some of the challenges associated with the topic. The authors need to clearly state how their work contributes to this literature.*

As stated in the first comment, it is true that Graph Theory has been used to model risk in different fields, and it is also true that some literature proposes the use of Graph Theory specifically to model natural hazard risk of specific types of systems (most often infrastructures, such as Dueñas-Osorio et al. (2004). However, to the best of our knowledge, the application of Graph Theory for broader disaster risk reduction and collective risk assessment purposes, as proposed in the article, is new.

As we aimed to describe in the Introduction, the common practice in the field of natural hazard risk is to adopt reductionist methods, which focus on exposed elements individually and therefore neglect a very significant parcel of the actual impacts. This is very clearly an under-explored area in catastrophe risk modelling, and one where more research work is warranted. As such, we are firmly convinced that this work makes a relevant contribution to the field.

However, we agree that the original manuscript fails to unequivocally identify this gap through a well-structured literature review, and therefore also fails to clearly position itself among that literature. While the introduction of the original manuscript aimed to achieve this, it probably did so in an insufficiently organized and incomplete manner. We therefore propose to restructure it by splitting it into three subsections and expanding certain parts, as described below:

1. Introduction

**1.1 Collective Disaster Risk Assessment: traditional approaches**

This subsection provides a brief contextualization of current practice and limitations in disaster risk assessments, making use of key references. Here we propose to add a relevant reference related to multi-hazard risk Zscheischler *et al.* (2018), as suggested by Referee #2 (L59).

**1.2 Modelling natural hazard risk in complex systems: state of the art and limitations**

This subsection introduces the need for holistic approaches that are able to handle the complexity of contemporary society. This is in contrast with the reductionist approaches presented in subsection 1.1, which only partially contribute to the assessment of the total impact, because they do not consider the connections between the exposed elements. The literature in this subsection aims to give an overview of the state of art and limitations of existing models to study complex systems. We propose to improve it by adding the suggested references listed here:

L70: Lhomme et al. (2013) showed that the "*city has to be considered as an entity composed by different elements and not merely as a set of concrete buildings.*"

L73: "The reductionist approach, in which the "risks are an additive product of their constituent parts" (Clark-Ginsberg *et al.*, 2018), contrasts with the complex nature of disasters."

We then show how the networks are treated in the infrastructure sectors, one of the sectors that traditionally address the complexity of interdependency. This brings to the concept of systemic vulnerability typical of cascading failures in the network, for which we also propose to add two suggested references Lewis (2014) and Setola *et al.* (2016) at L91. Subsection 1.2 ends with the presentation of the system of systems perspective.

**1.3 Positioning and aims**

The main difference between the original and the revised manuscript will be this subsection, where we wish to clarify:

(1) where our work is positioned in the recent theoretical framework covered by the proposed literature;

(2) the misunderstanding about the use of the network-based analysis: we do not analyse systems already organized as a network (e.g. electric power network), but we instead employ the network to represent a complex system such as an urban environment, and use Graph Theory as a diagnosis tool.

To achieve this, we propose adding the following:

"The aspects of complexity and interdependency have been investigated by various models of critical infrastructure as a single system, or as systems of  systems, which are networks by construction (e.g. drainage system). However, there is still a gap in current practice when it comes to modelling the complexity of interconnections between  individual  elements that do not explicitly constitute a network, which tend to be neglected by traditional reductionist risk assessments. Therefore, in this manuscript we propose an approach to model such interconnections and  develop a more holistic collective risk assessments.

This work proposes an approach to assess the interconnected risk (i.e. complex interaction between human, environment and technological systems) and a potential tool to model cascading risk (i.e. the results of escalation processes) and support more informed DRR decision making (Pescaroli and Alexander, 2018).

In particular, we understand that it is necessary to better analyse the interaction between these elements at risk and their influences on indirect impact assessment. The analyses of interaction and influence are assessed in this work by adopting the framework of Graph Theory, the branch of mathematics for the treatment of networks. Since its birth in 1736, Graph Theory has witnessed many exciting developments, and has been able to provide answers to a wide range of practical questions in many sectors (Boccaletti *et al.*, 2006). Given this context, this paper proposes an insight into collective risk assessment from an innovative holistic perspective.

The aims of this paper are: (…)"

*- Because they do not engage with this literature, I do not believe there is enough for a standalone theoretical paper on their topic. Rather than publishing this as a separate piece, I recommend using this article as a basis the literature review/methodology of the empirical paper, which provides a useful contribution to the literature.*

This article proposes the theoretical framework for a new approach to model collective risk of natural hazards in complex systems, such as urban environments. In order to do so, the introduction aims to engage with literature that is representative of the state of the art in this field, following the logical sequence described in the previous comment. Ultimately, the goal of the introduction is to provide a concise overview (i.e. brief but comprehensive), and then position this work among the existing body of literature.

We recognize that certain key references suggested by both Referees were missing, and following the very useful suggestions provided, we have expanded the introduction and added them. However, we would like to highlight that the goal of this article is *not* to provide an exhaustive literature review of the application of Graph Theory to risk analysis, or on collective risk assessments – these topics would likely require extensive review articles by themselves. As such, we believe that this does not justify the insufficiency of the article as a standalone theoretical paper, as the contents of the article go incontrovertibly beyond the literature review. Moreover, we believe that merging the two papers, even after a significant hypothetical reduction of both of them, would be harmful for their quality, and would still result in an excessively long manuscript. Finally, it is worth noting that the two papers may address different audiences: part I is targeted to a more general audience who may be interested in understanding the foundations of the approach, while part II points to technical experts and researchers who may want to implement this approach for their own practical applications. For these reasons, we believe that keeping the current structure with two companion papers is the optimal solution.

**References**

Boccaletti, Stefano, V. Latora, Y. Moreno, M. Chavez, and D. U. Hwang. 2006. "Complex Networks: Structure and Dynamics." *Physics Reports* 424(4–5):175–308.

Clark-Ginsberg, Aaron, Leili Abolhassani, and Elahe Azam Rahmati. 2018. "Comparing Networked and Linear Risk Assessments: From Theory to Evidence." *International Journal of Disaster Risk Reduction* 30(April):216–24. Retrieved (https://doi.org/10.1016/j.ijdrr.2018.04.031).

Dueñas-Osorio, Leonardo, James I. Craig, and Barry J. Goodno. 2004. "Probabilistic Response of Interdependent Infrastructure Networks." in *2nd annual meeting of the Asian-pacific network of centers for earthquake engineering research (ANCER)*. Honolulu, Hawaii. Retrieved (http://citeseerx.ist.psu.edu/viewdoc/download?doi=10.1.1.123.2036&rep=rep1&type=pdf).

Lewis, Ted G. 2014. *Critical Infrastructure Protection in Homeland Security: Defending a Networked Nation*. John Wiley & Sons.

Lhomme, S., D. Serre, Y. Diab, and R. Laganier. 2013. "Analyzing Resilience of Urban Networks: A Preliminary Step towards More Flood Resilient Cities." *Natural Hazards and Earth System Science* 13(2):221–30.

Pescaroli, Gianluca and David Alexander. 2018. "Understanding Compound, Interconnected, Interacting, and Cascading Risks: A Holistic Framework." *Risk Analysis* 38(11):2245–57.

Setola, Roberto, Vittorio Rosato, Elias Kyriakides, and Erich Rome. 2016. *Managing the Complexity of Critical Infrastructures*. Springer Nature.

Summer, Martin. 2013. "Financial Contagion and Network Analysis." *Annual Review of Financial Economics* 5:277–97.

Zscheischler, Jakob et al. 2018. "Future Climate Risk from Compound Events." *Nature Climate Change* 8(6):469–77. Retrieved (http://dx.doi.org/10.1038/s41558-018-0156-3).

---

## Author Comment (AC2) · 23 Jan 2019

Response to Referee #2

We wish to thank the Referee for his/her time and effort reviewing the manuscript. We greatly appreciate the constructive comments and suggestions, which we have carefully addressed in this response. Where applicable, changes are proposed to the manuscript accordingly (and marked up for clarity). Following the guidelines of the NHESS Editorial Board, the revised manuscript was not prepared at this point.

*- This paper only takes us about 60% of the way there. While I do think you have a novel idea of using graph theory to model risk transfer in a way that has not been done, you don't fully show us how to do it conceptually. e.g. you explain how graphs work. and give some discussion of how these graph properties link to vulnerability, resilience, and exposure. But you need to go much further.*

This article is organized as two companion papers, and our understanding is that this may have been overlooked by the Referee. The article is organized such that part I provides the theoretical framework, and part II demonstrates how it can be applied using a pilot study. We believe that part II (https://www.nat-hazards-earth-syst-sci-discuss.net/nhess-2018-278/) fully covers the issue raised by the Referee.

*- What metrics do you propose we use from graph theory that link to which metrics in risk assessment? Maybe this is what you are trying to do with percolation but it is still very unclear. How are you going to get us towards measure cascading risks with your new approach.*

These two companion papers, from a theoretical point of view in part I and a practical application in part II, propose a list of selected graph properties and discuss how these can be used in the assessment of the traditional components of risk. In particular, authority, closeness and percolation threshold are proposed respectively as metrics for the risk variables: exposure, vulnerability and resilience. Furthermore, these analogies are summarized in Table 1, and finally most of proposed metrics are applied in a case study in part II.

*- Part of the problem is the disorganized literature review and background. You are missing a lot of the resilience literature on this topic, and it feels like you are describing papers selectively. Please organize this into topics, themes, that lead to the demonstrating the gap in the lit that your new graph theory approach will allow us to fill.*

We fully agree with the Reviewer that the structure of the literature review of the manuscript can be improved. For this reason, we propose to restructure the Introduction, introduce the reference relevant for to the aim of the introduction, and write a new subsection (1.3) that help positioning the paper among the existing body of literature. These changes are described in detail below:

1. Introduction

1.1     Collective Disaster Risk Assessment: traditional approaches

This subsection provides a brief contextualization of current practice and limitations in disaster risk assessments, making use of key references. Here we propose to add a relevant reference related to multi-hazard risk Zscheischler *et al.* (2018) and Wahl *et al.* (2015) at L59.

1.2     Modelling natural hazard risk in complex systems: state of the art and limitations

This subsection introduces the need for holistic approaches that are able to handle the complexity of contemporary society. This is in contrast with the reductionist approaches presented in subsection 1.1, which only partially contribute to the assessment of the total impact, because they do not consider the connections between the exposed elements. The literature in this subsection aims to give an overview of the state of art and limitations of existing models to study complex systems. We propose to improve it by adding the suggested references listed here:

L70: "Lhomme et al., [2013] showed that the city has to be considered as an entity composed by different elements and not merely as a set of concrete buildings."

L73: "The reductionist approach, in which the "risks are an additive product of their constituent parts" (Clark-Ginsberg *et al.*, 2018), contrasts with the complex nature of disasters."

We than show how the networks are treated in the infrastructure sectors, one of the sectors that are traditionally able to assess the complexity of interdependency. This brings to the concept of systemic vulnerability typical of cascading failures in the network, for which we also propose to add two suggested references Lewis (2014) and Setola *et al.* (2016) at L91. Subsection 1.2 ends with the presentation of the system of system perspective.

1.3     Positioning and aims

The main difference between the original and the revised manuscript will be this subsection, where we wish to clarify:

(1) where our work is positioned in the recent theoretical framework covered by the proposed literature;

(2) the misunderstanding about the use of the network-based analysis: we do not analyse systems already organized as a network (e.g. electric power network), but we instead employ the network to represent a complex system such as an urban environment, and use Graph Theory as a diagnosis tool.

To achieve this, we propose adding the following:

"The aspects of complexity and interdependency have been investigated by various models of critical infrastructure as a single system, or as systems of  systems, which are networks by construction (e.g. drainage system). However, there is still a gap in current practice when it comes to modelling the complexity of interconnections between  individual  elements that do not explicitly constitute a network, which tend to be neglected by traditional reductionist risk assessments. Therefore, in this manuscript we propose an approach to model such interconnections and  develop a more holistic collective risk assessments.

This work proposes an approach to assess the interconnected risk (i.e. complex interaction between human, environment and technological systems) and a potential tool to model cascading risk (i.e. the results of escalation processes) and support more informed DRR decision making (Pescaroli and Alexander, 2018).

In particular, we understand that it is necessary to better analyse the interaction between these elements at risk and their influences on indirect impact assessment. The analyses of interaction and influence are assessed in this work by adopting the framework of Graph Theory, the branch of mathematics for the treatment of networks. Since its birth in 1736, Graph Theory has witnessed many exciting developments, and has been able to provide answers to a wide range of practical questions in many sectors (Boccaletti *et al.*, 2006). Given this context, this paper proposes an insight into collective risk assessment from an innovative holistic perspective.

The aims of this paper are: (…)"

We believe that this new structure of the Introduction and its proposed improvements address the Referee's comment and provide a much more logical sequence and organization for these topics. It is worth noting that the purpose of this section is not to provide an exhaustive literature review of a specific sector (e.g. resilience or critical infrastructure), but to engage with literature that is representative of the state of the art in this field, providing a concise overview (i.e. brief but comprehensive) to readers, and then position this work among the existing body of literature.

*- This idea has a lot of promise, but needs work. The conclusion should make me feel like I have a new tool and idea to measure risk. But I am left feeling confused.*

Presumably this comment applies to part I of the article. We believe that the Referee's idea, with which we agree, is covered by the full article (i.e. both companion papers together). Nevertheless, note that this topic warrants further research, which is duly acknowledged in the manuscript.

*- Line 45. Vulnerability does consider social conditions. That is a wrong statement*

We agree and do not state otherwise. The contrast presented here is between social and *physical* vulnerability, as explicitly written in the previous line of the manuscript.

*- Line 59. See the work on compound flood risk. Eg. Wahl, Thomas, Shaleen Jain, Jens Bender, Steven D. Meyers, and Mark E. Luther. "Increasing risk of compound flooding from storm surge and rainfall for major US cities." Nature Climate Change 5, no. 12 (2015): 1093. Zscheischler, J., Westra, S., Hurk, B.J., Seneviratne, S.I., Ward, P.J., Pitman, A., AghaKouchak, A., Bresch, D.N., Leonard, M., Wahl, T. and Zhang, X., 2018. Future climate risk from compound events. Nature Climate Change, p.1.*

We agree on the relevance of both references and propose adding them to the article at L59

*- Line 80. Great examples. Surprise to see lack of citatations for the large literature on compo unding risk and cascading failtures from the resilience field. E.g. Buldyrev, S. V., Parshani, R., Paul, G., Stanley, H. E., & Havlin, S. (2010). Catastrophic cascade of failures in interdependent networks. Nature, 464(7291), 1025. Chicago*

We will add the suggested reference in the Introduction.

*- Line 84. I have never heard of this rinaldi paper. I doubt it is the most quoted.*

Rinaldi et al. (2001) is an essential reference in the assessment of critical infrastructure risks, with currently over 2000 citations on Google Scholar for example, as also underlined by Referee #1 "*A large literature builds on Rinaldi et al. (2001) to use graph theory to assess critical infrastructure risks*". We recognize that this article may be not as well known

in other neighbouring fields of science (e.g. other sectors of risk, or applications of graph theory), but the validity of this statement is indisputable and we therefore propose to keep it.

*- In general this literature review feels selective and disorganized. Use subheadings. What is the gap you are filling? Are you really the only/first people to use graph theory to assess risk. I somehow doubt it. A simple google scholar search revealed many articles:*

*Heckmann, T., Schwanghart, W., & Phillips, J. D. (2015). Graph theoryâATRecent developments of its application in geomorphology. Geomorphology, 243, 130-146. Holmgren, Åke J. "Using graph models to analyze the vulnerability of electric power networks." Risk analysis 26, no. 4 (2006): 955-969. Lhomme, S., Serre, D., Diab, Y., & Laganier, R. (2013). Analyzing resilience of urban networks: a preliminary step towards more flood resilient cities. Natural hazards and earth system sciences, 13(2), 221-230.*

*Also see risk transfer analysis 1. Sapountzaki, K. Social resilience to environmental risks: A mechanism of vulnerability transfer? Manag. Environ. Qual. An Int. J. 18, 274–297 (2007).*

We agree with the Reviewer that the structure of the literature review of the manuscript can be improved, and above we proposed a significant number of improvements to address this issue. In the revised manuscript we will also add the following references suggested by the Referee: Åke J. Holmgren (2006); Lhomme et al. (2013) and Sapountzaki (2007).

Note that the manuscript does not claim in any way to "*be the only/first people to use Graph Theory to assess risk*": As we underline above, the literature on the use Graph Theory in risk assessment is large, but also (and more importantly here) extremely diverse. Many (if not most) of the articles that show up on a "*simple google scholar search*" actually have little relevance given the scope of our manuscript. The manuscript includes the references that we believe are most relevant and representative of the state of the art in this field (i.e. natural hazard risk modelling of systems that are not explicitly arranged as a network but whose underlying connections can significantly magnify impacts and risk), and could help the reader to understand the purpose of the proposed approach.

*- Page 165. Its hard to read all your definitions in prose. Made a table or a diagram that shows in a depiction each term. Add more to figure 1.*

We agree, and propose adding the following table at L151 to address this issue:

**Table 1: Graph properties description**

| Graph properties | Description |
| --- | --- |
| Degree (k) | The number of edges incident with the node |
| Path length | The geodesic length from node i to node j |
| Closeness | The distance (number of links) of a node to all others |
| Betweenness | The shortest paths between pairs of nodes that pass through a given node |
| Authority | Value of a node proportional to the sum of the node hubs pointing to it |
| Hub | Value of a node proportional to the sum of authority of nodes pointing to it |
| Percolation threshold (pc) | The minimum value of fraction of remaining nodes (p) that leads to the connectivity phase of the graph |

*- Line 196. What is pc. What is k.*

The definitions of pc and k are presented at L194 and L197, respectively.

*- Line 265. It is not until here that you tell me what graph theory contributes to vulnerability analysis.*

This follows the logic behind the structure of the manuscript, where we first provide context for the research (Introduction), then present some relevant aspects of graph theory, followed by the workflow that we propose in our approach, and finally show the analogy between graph properties and exposure, vulnerability and resilience. We believe this aids overall clarity and organization.

*- WHY is current risk analysis lacking and WHAT does graphs uniquely help us understand.*

We believe that the main shortcoming of current reductionist approaches is the impossibility to consider the connections between exposed elements, as also underlined by the suggested reference Lhomme et al. (2013). Our manuscript proposes

an approach based on Graph Theory that aims to take into consideration these connections, and treat the exposed elements as part of a whole system. The analogy proposed in part I and the application in part II show how the properties of a graph can provide information on the risk variables.

*- 330. I have never heard of this definition of resilience. This need to be motivated by the enourmous literature on the topic to some degree.*

The definition of resilience is proposed at line 48 ("system's capacity to cope with stress and failures and to return to its previous state"). Instead, at L330 we specifically underline the dynamic features of resilience compared to vulnerability. Since this was not clear enough, we are now suggesting a modification that also incorporates two references proposed by the Reviewer L328-L331:

"Resilience differentiates from vulnerability in terms of dynamic features of the system as a whole. The properties and functions used to model vulnerability are static characteristics that do not consider any time evolution, or using the words of Sapountzaki (2007), "*vulnerability is a state, while resilience is a process*";  in fact the definition of resilience implies a time evolution of the characteristics of the whole system. In addition, Lhomme et al. (2013) underline "*the need to move beyond reductionist approaches, trying, instead, to understand the behaviour of a system as a whole*". These two different features, dynamic aspect and whole system  can be expressed by a cinematography analogy: vulnerability is a single frame of the resilience video."

**References:**

Åke J. Holmgren. 2006. "Using Graph Models to Analyze the Vulnerability of Electric Power Networks." *Risk Analysis* 26(4):955–69.

Boccaletti, Stefano, V. Latora, Y. Moreno, M. Chavez, and D. U. Hwang. 2006. "Complex Networks: Structure and Dynamics." *Physics Reports* 424(4–5):175–308.

Clark-Ginsberg, Aaron, Leili Abolhassani, and Elahe Azam Rahmati. 2018. "Comparing Networked and Linear Risk Assessments: From Theory to Evidence." *International Journal of Disaster Risk Reduction* 30(April):216–24. Retrieved (https://doi.org/10.1016/j.ijdrr.2018.04.031).

Lewis, Ted G. 2014. *Critical Infrastructure Protection in Homeland Security: Defending a Networked Nation*. John Wiley & Sons.

Lhomme, S., D. Serre, Y. Diab, and R. Laganier. 2013. "Analyzing Resilience of Urban Networks: A Preliminary Step towards More Flood Resilient Cities." *Natural Hazards and Earth System Science* 13(2):221–30.

Pescaroli, Gianluca and David Alexander. 2018. "Understanding Compound, Interconnected, Interacting, and Cascading Risks: A Holistic Framework." *Risk Analysis* 38(11):2245–57.

Rinaldi, Steven M., James P. Peerenboom, and Terrence K. Kelly. 2001. "Identifying, Understanding, and Analyzing Critical Infrastructure Interdependencies." *IEEE Control Systems Magazine* 21(6):11–25.

Sapountzaki, K. 2007. "Social Resilience to Environmental Risks: A Mechanism of Vulnerability Transfer?" *Management of Environmental Quality: An International Journal* 18(3):274–97.

Setola, Roberto, Vittorio Rosato, Elias Kyriakides, and Erich Rome. 2016. *Managing the Complexity of Critical Infrastructures*. Springer Nature.

Wahl, Thomas, Shaleen Jain, Jens Bender, Steven D. Meyers, and Mark E. Luther. 2015. "Increasing Risk of Compound Flooding from Storm Surge and Rainfall for Major US Cities." *Nature Climate Change* 1093–97.

Zscheischler, Jakob et al. 2018. "Future Climate Risk from Compound Events." *Nature Climate Change* 8(6):469–77. Retrieved (http://dx.doi.org/10.1038/s41558-018-0156-3).

---

## Referee Report (RR1)

**Review Report on**

"Natural hazard risk of complex systems – the whole is more than the sum of its parts: I. A holistic modelling approach based on Graph Theory" and

"Natural hazard risk of complex systems – the whole is more than the sum of its parts: II. A pilot study in Mexico City"

*General summary*

I carefully read the second version of both the manuscripts which is organized in two parts: part I describes the proposed "graph theory" approach, and part II illustrates an application of the graph theory to a pilot study in Mexico City. The author(s) argues that from a more practical perspective, these two companion papers may address different audiences: part I is targeted to a more general audience who may be interested in understanding the foundations of the approach, while part II points to technical experts and researchers who may want to implement this approach for their own practical applications.

I identified various loopholes in the written manuscripts which needs to be sincerely addressed before this manuscript could proceed with publications, which are as follows:

1. *Unjustified strong claim*
   I do not agree with the author(s) claim that this manuscript has proposed a new approach. Graph theory is an existing approach which has invaded almost all branches of science (as clearly mentioned by both the reviewers in their first round of review). This manuscript only extends its application to "Natural hazard risk of a complex system". I would suggest moderating the tone of the paper mentioning that they have used/applied graph theory for X purpose properly giving credit to all seminal papers.
   *For instance, see Part I:* page 6, second aim, *"to propose a new approach……."*.

2. *Confusing between Graph and Network theory, terms and concepts*
   Reading both the manuscripts it seems that the author(s) did not do a detailed (breadth and depth) literature review. For instance, (a) throughout both the manuscripts authors used particular terms associated with network and graph interchangeably which is not scientifically correct and created confusion; (b) many seminal papers, on which graph theory is based, are omitted from the citation list, (c) vertices and edges are particularly associated with graph whereas node and links with network. Either stick to one terminology or highlight the difference between graph and network and make a statement that all terminologies (Vertices and node, edge and link) are same and they are using it without any distinction.
   Further, in a single statement, the study is using Network and graph simultaneously, in my opinion, which is not acceptable. For instance, "*Part I: Section 2.1, In the scientific community, the mathematical properties of a network are studied using Graph theory,*"
   Also, I am not fine with the definition of Graph theory. In part I, the author write that Graph Theory is the branch of mathematics that studies the properties of **networks** (P6/L143) whereas in the

part II author states that Graph Theory is the branch of mathematics that studies the properties of graphs (P2/L37). Does the author think that Graph and Network are exactly the same? Well, in my understanding they are different indeed there is overlapping.

I am struggling to understand why authors decided to call the approach based on Graph theory and not Network theory. What makes it really a graph theory? Does the author agree that "a network is a diagrammatical representation of some physical system or structure whereas a graph, on the other hand, is a mathematical notion that represents only the structure of a network without physical meanings?" If yes, I would prefer to call approach based on Network theory. Further, I also have a notion that Graph theory largely has its root in Mathematics where it has been used to conceptualize the problems into a graph whereas Network theory provides a set of techniques for analyzing such graphs. Further, many concepts such as multi-layer network, dynamic network, coarse-graining our flourishing with network theory only.

At last, I would say that I am not strict with terminology since every researcher has its own notion but indeed I am more inclined to use Network theory instead of graph theory. Further, if the author continues to go with graph theory makes sure manuscripts clearly deal with the terms and concepts of only graph theory.

3. *Structuring and content of part I*

The author(s) argues that part I is targeted to a more general audience who may be interested in understanding the graph theory.

Being a general audience, I sincerely have difficulties to understand many terms theoretically as well as mathematically, presented examples, terminologies, mathematical concepts and more importantly the aim. As both the reviewer mentioned (reviewer 2, specifically) I feel the paper is disorganized and at last, it does not convince me. My very specific observations are as follows:

Abstract: L14, *this paper proposes a new holistic approach to assess the risk in a complex system based on Graph theory.* What is the approach? To identify vertices of a graph, setup edges and analyze the resultant graph? I feel it is a standard way, isn't it? Then, how this manuscript justify the approach? Only based on a hypothetical city example, which is very subjective. I am still struggling to understand what this manuscript contributes to the existing knowledge.

My biggest concern is that the study deals with graph theory but does not provide any mathematical details. Which is unacceptable in such kind of study.

Page 6/Line 158, section 2.1: electrical power grid, the internet, highway and neural network, being general audience I am not able to visualize what is vertices and edges in above-mentioned graphs and importantly directions (if directed network).

Coming back to the aim of the paper, the entire study deals with the directed network and hence since the beginning the author (s) need to put more stress on the directed network. For example, section 2.1 need more words, more examples and mathematical notions about a directed network.

*The author(s) take many different examples to explain the network concept whenever they need, without any coherence structure, For instance, P13/L351: As an example, in a road graph, a bridge node has a higher value of betweenness because all the nodes of a sub-graph (e.g. one side of the river) need to*

*pass through the bridge node in order to connect to the nodes of the other sub-graph (the other side of the river). In the case of bridge failure, the two sides of the river are isolated and the original road graph splits into two sub-graphs.*

*What is the road graph? What are the edges and vertices? What is the bridge node? How to decide? Do we have a river in the road graph? It is all theoretical and very subjective. Not acceptable and convincing.*

*Another example is section 3.1.3 of the earthquake.*

4. *Mathematical details and graph theory measures*

Page 7/line 194: *A node with high hub value points to many other nodes, while a node with high authority value is linked by many different hubs. Mathematically, the authority value of a node is proportional to the sum of the node hubs pointing to it and the hub value of a node is proportional to the sum of the authority of nodes pointing to it.*

Is this definition provided by the authors exclusively? If not, I couldn't find any citation to valid above-written definition.

I am wondering whether or not this problem has a unique solution. Author claim that this article is suitable to a general audience, being an expert in network theory I am unable to understand it. I request to take a dummy graph (very simple) and explain how the author(s) setup direct graph, decide hubs, authority values and hub value of a node.

Page 8/line 201: *Depending on the statistical properties of the degree distribution, there are two broad classes of networks: homogeneous and heterogeneous…….. Again, i*s this definition provided by the authors exclusively? If not, I couldn't find any citations. Is the author sure about the above-mentioned statement? I am struggling to validate this definition.  What do authors mean by the homogeneous network? What are the properties of the homogeneous network? I assume the author might be pointing to a regular network because, by definition, each node in a regular network has the same number of links. If so, this is a very absurd statement.

So as per the claim, all the network having Poisson distributions are homogeneous? This again makes my conviction strong that authors did not check the literature appropriately.

What a random network is, as mentioned in (P8/L224)? Further, $P_c = 1/\overline{K}$, I do not understand this mathematical expression?

Section 3.1.2: P12/L343: *closeness: a shorter path between a node and the network?* Do you mean shorted path between a node and all other nodes? Difficult to understand, please Rewrite it.

*Table 1: closeness:*

Which is closeness author talking about? I did not understand the definition? Is it closeness centrality?  If yes, please rewrite it to "shortest path length from the node to every other node in the network". Again mathematical formulas of all the terms are unavoidable.

Hub? Does the author mean two different hubs in the network? Hub value of a node and hub node itself? Confusing. Make it clear.

I just did not understand the concept of percolation and others in the absence of proper mathematical definitions. Mathematical details are indeed important.

5. *Section 2.2*

   *"We proposed an approach based on the following two major phases".* I appreciate the *author's creativity in designing the text and section however, this is more general and clubbed into one section called graph construction to reduce the redundancy.*

   *In an entire section of based on topology, I couldn't see any seminal paper cited. Not appreciated. Did the author has heard about coarse-graining of network or topological scale in the network? It goes in the same direction what the author has explained (entire network, community and a single node).*

   *P10/L279: it is necessary to define rules? I would be happy to see the rules since the author claim that they are proposing an approach and hence very consolidate approach with the full explanation is needed. Section 2.2 from part II should have been here.*

6. *Section 3.2*

   *I am sorry to the author but I couldn't find anything new here.*

7. *Discussion*

   Too much repetition, again and again, that we have proposed an approach which can solve all the problems

   Page 16/line 443: *"This new approach...."* Is it really a new approach?

   Line 443 to line 454 should go to introduction.

   Line 455 and line 342: *"The proposed approach is suitable for multi-hazard assessment" many times repeated.*

8. *Other minor comments*

   Fig. 1c could be improved by weighting the links.
   Fig.3: there are 9 blocks, not 8. Correct in the text (Section 3.2, line 403).

*Summary*

I am not convinced with the author's claim that this study is suitable for General Audience. The author (s) has tried to oversell the content without giving proper justification and showing any results. Indeed it fails in giving credit to previous studies. Many seminal papers are missing and it clearly seems that vast and in-depth literature review is missing. I found many terse and absurd statements. The study clearly lacks in terms of mathematical justifications. Too many over claim statement such as approach "Could be used for risk mitigation strategies", "Similar analysis could be carried out for betweenness to obtain more insights into the risk assessment". Therefore, I recommend rejection of part I.

Nevertheless, it was a great attempt and I motivate the team to work and explore more to fill the gaps. Due to this, I am afraid that the entire manuscript should be rejected, and the authors should given an opportunity to merge both the manuscripts and resubmit. I am willing to review the contribution, if the authors want to submit a revised work. I hope I am not unduly discouraging, but the problems detailed above are sufficiently severe for the work not to be considered for publication in the current form. However, I do think that the authors have applied an interesting methodology that should be investigated thoroughly, and which may lead to important breakthroughs in the area of interest. Therefore, I encourage the authors to further pursue this method, and I hope my suggestions are useful for this endeavor.

**Few specific comments to part II**

"Natural hazard risk of complex systems – the whole is more than the sum of its parts: II. A pilot study in Mexico City"

I have read both the articles in detail and based on my observation I think it is well clear that part II need major revisions, rewriting and restructuring. I suggest the author to moderate tone in the revised version, specifically focus on the primary goal and go through the seminar paper published in other domains. Please remove all the content which is not relevant to the objective. For example, discussion on the homogeneous and heterogeneous network, unsuitable examples etc. Kindly do not provide off-topic details.

If the author will do that I am sure merging both the paper will not be lengthy at all. The introduction of part II is not convincing and always direct the readers to the part I.

*Minor comments*

*Page 6/line 151: closeness centrality measures........which it is.....remove it.*

*Summary*

As indicated merging both the manuscript will be an ideal option. Therefore, I am not providing content specific comments at this stage since the author(s) needs to majorly restructure the content.

---

## Referee Report (RR2)

**Review Report on**

"Natural hazard risk of complex systems – the whole is more than the sum of its parts: I. A holistic graph-based assessment approach"

and

"Natural hazard risk of complex systems – the whole is more than the sum of its parts: II. A pilot study in Mexico City"

**General summary on the revised manuscript**

I received the third version of both the manuscripts describing the potential advantages of the use of a graph theory particularly in understanding fundamental aspects of complex systems which may have relevant implications to natural hazard risk and among others. In addition, part II illustrates an application of the graph theory to a pilot study in Mexico City. I could see a little refinement in both the manuscripts and I do acknowledge author's efforts, but there are still many open issues. I'm still not completely convinced by the revised version and hence I do emphasize that all my previous major concerns are still remains unanswered.

These concerns need to be sincerely addressed before this manuscript could be thought to proceed with publications, which are as follows:

*Standalone contribution of each individual manuscript*

In my last report, I very clearly emphasized that both the manuscripts do not standalone. I am unhappy to see that authors overlooked my efforts on the manuscript and tried to come up with good answers rather than sincerely working on it. I again point to the aim of the paper (P5/L139-145). Is stating or elaborating on construction of a hypothetical graph (shown in Fig.3), potential advantages of the use of the graph, limitation, potentialities etc. is a worthy contribution when authors already acknowledged that it is not a new approach?

A complete study of all relevant graph properties (P15/L411-415) discussed above and a more realistic hazard scenario are presented in Part II of this manuscript for a selected case study. (this statement again shows the need to merge both the studies)

*Mathematical details and graph theory measures*

Mathematical representation and visualization (Section 2): I could see that authors have provided all the mathematical formulas in the Table 1. However, the idea was to take an example of a simple graph and discuss all these measures and there significance mathematically. With this simple graph, author could have discussed network terminologies used in the manuscript such as hub node, authority etc. (see my previous comments: The author(s) argues that part I is targeted to a more general audience who may be interested in understanding the graph theory. My biggest concern is that the study deals with graph theory but does not provide any mathematical details. Which is unacceptable in such kind of study).

(Please see my previous comments: I am wondering whether or not this problem has a unique solution (NOT ANSWERED!!!). Author claim that this article is suitable to a general audience, being an expert in network theory I am unable to understand it. I request to take a dummy graph (very simple) and explain how the author(s) setup direct graph, decide hubs, authority values and hub value of a node).

**Subjective examples and interpretations**

As pointed previously as well, the example presented in Fig. 3 (section 4) is very subjective and case specific. I am not able to replicate it. How should I use it on another problem? Let's imagine a system of airport with 20 blocks, 3 out of which deal with intercontinental flights (short, mild and long distance respectively), 3 deals with intracontinental flights (short, mild and long distance respectively) and one block is for local flights. Other blocks are fire stations, fuel stations etc. Reading through the manuscript leaves one with many questions than answers. Authors critically need to asses the objectivity of the manuscript in its present form.

This illustrative example shows (P15/L401-405) how the single elements can be considered as part of the whole network and not as single separate entities. This holistic approach adds information to the traditional approach…. SUCH STATEMENTS ARE VERY GENERIC.

**Vague statements**

(P16/L438-444): First of all, there is a mature theory of mathematics, the Graph Theory, that already studies the properties of a graph (REPETITIVE). Are these graph properties telling us something useful to assess the risk of natural hazards affecting these complex systems? We showed that some of the graph properties can disclose some relevant characteristics of the system related to the risk 440 assessment. What is the vulnerability and exposure of the system? There are some interesting analogies between graph properties such as hub, betweenness and degree-out values and the "systemic" vulnerability. WHAT ARE SOME?

I couldn't find any limitations which is also one aim of the study (P6/L144: to discuss the limitations, potentialities and future developments of this approach compared to other more traditional approaches).

In my last review I pointed that the author(s) took many different examples to explain the network concept whenever they need, without any explanation (what is node, what is link, how setup link etc.). In the revised revision, author escaped by deleting many such examples still without explaining the retained ones.

(P12/L315-323): As an example, in a road network where road segments are represented by links, whereas crossroads and bridges are represented by nodes. In this case, a bridge would likely be the node with a higher value of betweenness because all the nodes of a sub-network (e.g. all the nodes that are in one side of the river) need to pass through the bridge node in order to connect to the nodes of the other sub-network 320 (all the nodes on the other side of the river). In the case of a bridge failure, the two sides network, separated geographically by the river are isolated and the original road network splits into two sub-networks. THIS IS NOT TRUE!!!

[Figure]

I find the bridge node (node number 7) having less betweenness than crossroads.

**Summary**

I strongly retain my decision and recommend rejection of both the manuscripts in the present format. A fresh submission could be the way forward.

---

## Referee Report (RR3)

The authors continue to improve the article, yet questions remain about the article's specific rationale and focus. The authors need to develop a clearer rationale and carry it through the entire article. The literature review primarily focuses infrastructure literature, arguing that it focuses mostly on "the analysis of a single infrastructure typology" and identifies SoS as a potential solution. I would expect the rest of the article to highlight this gap and then develop a solution. Yet the positioning and aims does not discuss this but instead focuses on general systems, their interconnections, and their need for a holistic assessment. Whether the article is about SoS and infrastructure interdependencies or a broader question of general systems the authors should revise these sections to build a more coherent argument for the article.

The authors then need to carry this into the methodology, either referencing or building on previous research to justify their approach. The methodology section overviews literature on graph theory before discussing concepts related to exposure, vulnerability, and resilience and how they might be represented in a graph. While this conceptual discussing is important for the remainder of the article, it is not referenced or imbedded in previous studies. Are the operationalizations outlined in this section new or do they build on work of others? Why are they worth looking at? Be explicit about why it's important to understand issues of exposure, vulnerability, and resilience, and if the concepts are new to this type of analysis say so. This discussion should reference literature and occur in both the literature review and the methodology section.

Finally, the discussion and final considerations should discuss how this study built on, challenged, and/or complemented previous research. What new does this study say? How is it similar or dissimilar to other studies? How does it support more informed DRR decision making (one of the aims identified by the authors). Again, the authors need to bring in and reference other studies, comparing their results to the results of others.

A few other minor comments:
1. tone down the opening statement of 'we live in an increasingly complex world'. Very difficult statement to provide and tangential to the study. Instead, say something like 'the world is complex' and then describe how it's complex. Easier to defend and aligns with the argument.
2. define resilience (paragraph starting on line 330). Resilience is a fraught and contested topic with a multitude of meanings (see e.g. Manyena 2006 'the concept of resilience revisited' as an example of the multitude of meanings). What do you mean by resilience? Is it about recovery and 'return to normal', ability to absorb shocks, the ability to transform in the face of adversity, or something else?
3. avoid the term 'natural disasters'. Hazards may be natural but disasters are not. Instead you can say 'disaster induced by natural hazard'. See https://www.undp.org/content/undp/en/home/blog/2017/5/18/Natural-disasters-don-t-exist-but-natural-hazards-do.html, and https://www.unisdr.org/we/inform/publications/65974 and https://www.nonaturaldisasters.com

---

## Author Response (AR2)

**Report#1 - accepted subject to minor revisions.**

We wish to thank the Referee for his/her time and effort reviewing the manuscript. We are grateful for the helpful comments and suggestions, which we have carefully addressed in the following responses and in the revised manuscript. Please note that all references to line numbers in this response refer to the marked-up version of the manuscript.

*-Overall, much improved from the previous version, more clear, and I liked the figures and table- really helped. The only revision I would request is to be really careful conceptually. A lot of people using these terms in their own disciplines! So its not totally new to do holistic risk assessment-but perhaps new to the authors. Maybe sure to cite and recognize other disciplines/folks working on the topic (some of which are not Italian!).*

*-Line 43- I still don't think there is even consensus among disciplines! Geography and the social sciences approaches vulnerability really differently and measures it differently to this day. Rephrase.*

We acknowledge that the calculation of risk is in continuous discussion in the scientific community, particularly among disciplines. We have rephrased this sentence accordingly:

> L49: "The research community concerned with Disaster Risk Reduction (DRR), particularly in the fields of physical risk, has generally agreed on a common approach for the calculation of risk ($R$) as a function of hazard ($H$), exposure ($E$), and vulnerability ($V$): $R = f ( H, E, V )$ (e.g.  Balbi et al., 2010; David, 1999; IPCC, 2012; Schneiderbauer and Ehrlich, 2004)."

We have also included additional references to support the provided definition:

- David, C. (1999). *The Risk Triangle*. London, UK: Jon Ingleton.
- IPCC, 2012: *Managing the Risks of Extreme Events and Disasters to Advance Climate Change Adaptation*. A Special Report of Working Groups I and II of the Intergovernmental Panel on Climate Change [Field, C.B., V. Barros, T.F. Stocker, D. Qin, D.J. Dokken, K.L. Ebi, M.D. Mastrandrea, K.J. Mach, G.-K. Plattner, S.K. Allen, M. Tignor, and P.M. Midgley (eds.)]. Cambridge University Press, Cambridge, UK, and New York, NY, USA, 582 pp.
- Schneiderbauer, S. and Ehrlich, D.: *Risk, hazard and people's vulnerability to natural hazards: A review of definitions, concepts and data*, Eur. Comm. Jt. Res. Centre. EUR, 21410, 40(January), doi:10.1007/978-3-540-75162-5_7, 2004.

*-Line 64… could add Eakin et al 2017 as a reference (to include resilience cities approaches, ub habitat etc also demanding holistic approaches) Eakin, H., Bojórquez-Tapia, L. A., Janssen, M. A., Georgescu, M., Manuel-Navarrete, D., Vivoni, E. R., … Lerner, A. M. (2017). Opinion: Urban resilience efforts must consider social and political forces. Proceedings of the National Academy of Sciences, 114(2), 186–189.* http://doi.org/10.1073/pnas.1620081114 *Also there is the SETS literature e.g.: Markolf, S. A., Chester, M. V, Eisenberg, D. A., Iwaniec, D. M., Davidson, C. I., Zimmerman, R., … Chang, H. (2018). Interdependent Infrastructure as Linked Social, Ecological, and Technological Systems (SETSs) to Address Lock-in and Enhance Resilience. Earth's Future, 6(12), 1638–1659.*

We agree on the relevance of both references and have added them to the revised manuscript at L72.

*-LINE 121. Two periods? Remove one?*

We propose the followings modifications:

> L132: "Tsuruta and Kataoka (2008) use matrices to determine damage propagation within infrastructure networks (e.g. electric power, waterworks, sewerage, telecommunication, road, and social functions like finance, medical treatment and administration) due to interdependency based on earthquake data and expert judgment considering infrastructure networks ."

*-Line 270…more thorough assessments? More thorough than what? I am still not totally clear as to what graphs and graph concepts tell us about risk in a systems that we could not get through another method. What is its unique contribution?*

This manuscript contributes to explore some fundamental aspects of complex systems exposed to natural hazard risk leveraging well known graph properties. Furthermore, the graph is proposed as tool to model the propagation of impacts of a natural hazard in order to analyse second-order consequences and quantitatively estimate risk. Both aspects analysed by the graph, properties of the systems and cascading effects, allow a more thorough assessment of risk compared to traditional approaches. We agree that this may not have been fully clear in the previous versions. To address this, we have completely rewritten sub-section "*1.3 Positioning and aims*", modified Figure 2 in order to show the two major advantages of the proposed approach, and considerably expanded sections 3 and 4.

*-Line 329. I agree that resilience does include more time dimensions but I don't think the analogy of cinematography is right. Resilience is about the ability to preserve system structure and function with a perturbation. Vulnerability is not a subcomponent of resilience. In fact, some disciplines use the work resilience in opposition to vulnerability. (See Anderies's work on robustness-vulnerability tradeoffs) or this article by Turner Turner, B. L. (2010). Vulnerability and resilience: Coalescing or paralleling approaches for sustainability science? Global Environmental Change, 20(4), 570–576. http://doi.org/10.1016/j.gloenvcha.2010.07.003 I do agree however that the percolation threshold is a new and interesting way to measure resilience.*

We propose rephrasing this sentence as described below, removing the cinematography analogy as suggested:

> L394: "These two features, dynamic aspect and whole system, make vulnerability different from resilience and further clarify the need to develop an approach that it is able to consider the dynamic of the system as a whole. ."

*-Line 455. Consider a stronger conclusion, that really gets us excited about why using graph theory can help us unlock large theoretical questions in vulnerability/resilience science (and then state what those questions are!)*

We have restructured the original sections "*4. Discussion*" and "*5. Conclusion*" into a new section "*5. Discussion and final considerations*". Following the Reviewer's much appreciated comment, we have extensively rewritten this new section.

**Report#2 - reconsidered after major revisions: I am willing to review the revised paper.**
We wish to thank the Referee for his/her time and effort reviewing the manuscript. We are grateful for the helpful comments and suggestions, which we have carefully addressed in the following responses and in the revised manuscript. Please note that all references to line numbers in this response refer to the marked-up version of the manuscript.

*- The authors have improved this article by describing how their model is situated in the literature. Their review of the state of the art (section 1.2) is an improvement from their previous draft, and now touches on some of the relevant issues in holistic approaches to risk and the application of network analysis. However, the specific contribution of this model can be further clarified. The authors make a convincing argument that most approaches to risk assessment utilize reductionist approaches, and note that their model is designed to contribute to the holistic approaches to assessing risk.*

*- They argue that they provide a new way to model holistic risk, specifically of interconnected infrastructures, and provide a short review of the interconnected infrastructure/SoS literature. Their treatment of the interconnected infrastructure/SoS literature could be improved. In their critique they state that "this well developed branch of research is mostly focused on the analysis of a single infrastructure typology and the aim is usually to assess the efficiency of the infrastructure rather than the impact that its failure may have on society." However, in the preceding paragraph they provide several references (such as Lewis, Rinaldi) that focus specifically on interconnected infrastructure and failures therein. These authors represent part of a very large literature that has been developing since the early 2000s (see for instance some of the references to Rinaldi (link) including network - centric approaches (e.g. work of Lewis). The authors*

*should clarify specifically how their model relates to this work and what specifically it provides. What is the current state of literature, what are the gaps, and what does this model provide that is different?*

We have expanded the introduction section regarding the current state of the literature ("*1.2 Modelling natural hazard risk in complex systems: state of the art and limitations*"), particularly from L115. A recent key reference (Pant et al., 2018) that helps clarify what the gaps area has been added, as described below. We believe that this addition, together with the extensive changes and new content added to sub-section 1.3 and section 3, now allow for a much better understanding of what this approach can bring to the field of natural hazard risk.

L113: "Trucco et al. (2012) propose a functional model aimed at i) propagating impacts, within and between infrastructures, in terms of disservice due to a wide set of threats and ii) applying it to a pilot study in the metropolitan area of Milan. Pant et al. (2018) proposed a spatial network model to quantify  flood impacts on infrastructures in terms of disrupted customer services both directly and indirectly linked to flooded assets  These analyses could inform flood risk management practitioners to identify and compare critical infrastructure risks on flooded and non-flooded land, for prioritising flood protection investments and improve resilience of cities.

However, this well-developed branch of research is mostly focused on the analysis of a single infrastructure typology, and the aim is usually to assess the efficiency of the infrastructure itself rather than the impact that its failure may have on society. In particular, "*representations of infrastructure network interdependencies in existing flood risk assessment frameworks are mostly non-existent*" (Pant et al., 2018). These interdependencies are crucial for understanding how the impacts of natural hazards propagate across infrastructures and towards society."

*- On page 5 they state that "there is still a gap in current practice when it comes to modelling the complexity of interconnections between individual elements that do not explicitly constitute a network". It seems like this could be one potential contribution. Further articulating this contribution (or others that are identified following a more engaged literature review) will help understand the added value of this model.*

We agree, and have modified sub-section 1.3 accordingly (see below). We have also considerably expanded Section 3 and made various other changes throughout the document which should help clarify the added value of the approach.

L145: "In fact, although several authors have shown how to model risk in systems which are already networks by construction (Havlin et al., 2010; Reed, Kapur, & Christie, 2009; Rinaldi, 2004; Zio, 2016), fewer have addressed the topic of risk modelling in systems where that is not the case, i.e. systems are not immediately and manifestly depicted as a network (Hammond et al., 2013; Zimmerman et al., 2019). These include cities, regions or countries, which are complex systems made of different elements (e.g. people, services, factories) connected in different ways among each other in order to carry out their own activities. Therefore, in this manuscript we would like to promote an approach, which has previously deserved the attention of other authors, to model  the interconnections between the elements that constitute those systems and assess collective risk in a holistic manner. The approach involves the translation of the complex system into a graph, i.e. a mathematical structure used to model relations between elements."

*- The discussion on the contributions, once articulated in the literature review and positioning and aims, should also be used to improve the discussion. What are the specific implications that this model provides that are beyond what is already known in the literature? Based on the model, where should the field of practice be going? As with the literature review it would be important to reference relevant articles in describing where the field is and what your findings suggest.*

We agree with the Reviewer that the discussion should be improved in accordance with the new clarifications of literature review and positioning and aims. For this reason, we have made considerable modifications to sub-sections "*1.2. Modelling natural hazard risk in complex systems: state of the art and limitations*" and "*1.3. Positioning and aims*". Section 3, which describes the methodology, has also been expanded, which should now better highlight the novelty of

the model within the field of natural hazard risk. Finally, we have merged the original sections "*4. Discussion*" and "*5. Conclusion*" into a new section "*5. Discussion and final considerations*", which has been rewritten extensively.

*This article still does not seem substantial enough to stand on its own. It offers a short literature review and a detailed methodology, which is then used in the second article on the case study. Perhaps the value of this article will become more apparent once the contribution is better articulated. Better linking the model to the various bodies of literature (risk assessment, infrastructure interdependencies, and network analysis) will also help identify which elements of both articles are critical and which should be streamlined and reduced. This could help to determine whether this should be one article or two. Without such clarity I still suggest integrating both articles into a single article. Again, these articles could be a nice addition to network analysis approaches of interdependent infrastructure, but it is not yet clear what they add to the literature.*

We believe that the extensive modifications introduced to the article have resulted in a much-improved articulation between what its contribution and the existing body of literature. In particular, we have expanded the literature review and integrated additional relevant references, significantly improved the "*Positioning and aims*" sub-section, and expanded the workflow in Section 2.2 (including Figure 2) to highlight the main contributions of the methodology, among other changes aimed at improving overall clarity. We are convinced that this new version of the article has sufficient substance to stand its own, which should now be more easily evaluable.

**Report#3**
We wish to thank the Referee for his/her time and effort reviewing the manuscript. We are grateful for the helpful comments and suggestions, which we have carefully addressed in the following responses and in the revised manuscript. Please note that all references to line numbers in this response refer to the marked-up version of the manuscript.

*General summary*

*I carefully read the second version of both the manuscripts which is organized in two parts: part I describes the proposed "graph theory" approach, and part II illustrates an application of the graph theory to a pilot study in Mexico City. The author(s) argues that from a more practical perspective, these two companion papers may address different audiences: part I is targeted to a more general audience who may be interested in understanding the foundations of the approach, while part II points to technical experts and researchers who may want to implement this approach for their own practical applications. I identified various loopholes in the written manuscripts which needs to be sincerely addressed before this manuscript could proceed with publications, which are as follows:*

*- 1. Unjustified strong claim*
*I do not agree with the author(s) claim that this manuscript has proposed a new approach. Graph theory is an existing approach which has invaded almost all branches of science (as clearly mentioned by both the reviewers in their first round of review). This manuscript only extends its application to "Natural hazard risk of a complex system". I would suggest moderating the tone of the paper mentioning that they have used/applied graph theory for X purpose properly giving credit to all seminal papers.*
*For instance, see Part I: page 6, second aim, "to propose a new approach……..".*

We agree, and have moderated the tone throughout the document. In the revised manuscript, the approach is not presented as "new" anymore. We have also completely rewritten the "*1.3 Positioning and aims*" sub-section, including the main aims:

> L164: "The aims of this paper  can be summarized as follows:
> - to call for a paradigm shift from a reductionist to a holistic approach to assess natural hazard risk, supported by the construction of a graph;
> - to show the potential advantages of the use of a graph: (1) understanding fundamental aspects of complex systems which may have relevant implications to natural hazard risk, leveraging well known graph properties, (2) using

the graph as a tool to model the propagation of impacts of a natural hazard and, eventually, assess risk in complex systems;
- to discuss the limitations, potentialities and future developments of this approach compared to other more traditional approaches.
-
-
-
- "

*- 2. Confusing between Graph and Network theory, terms and concepts*
*Reading both the manuscripts it seems that the author(s) did not do a detailed (breadth and depth) literature review. For instance,*
*(a) throughout both the manuscripts authors used particular terms associated with network and graph interchangeably which is not scientifically correct and created confusion;*
We recognize that the use of these terms interchangeably throughout the document is not scientifically correct and might create confusion among readers. Where appropriate, we have replaced the "network" term with "graph": L182, L186, L190, L191, L213, L214, L220, L221, L227, L280, L309 and L380.

*(b) many seminal papers, on which graph theory is based, are omitted from the citation list,*

We have added references to several seminal papers regarding graph theory: Barabasi, 2016; Biggs, Lloyd, & Wilson, 1976; Börner, Soma, & Vespignani, 2007; Euler, 1736; Luce & Perry, 1949; Wilson, 1996. We have also included additional references to support the provided definition:
- Barabasi, A. L. (2016). Network Science (Cambridge University Press, ed.). Retrieved from http://barabasi.com/networksciencebook/%3E
- Biggs, N. L., Lloyd, E. K., & Wilson, R. J. (1976). Graph Theory 1736-1936 (Clarendon Press, ed.).
- Börner, K., Soma, S., & Vespignani, A. (2007). Network Science. In Medford (Ed.), Annual Review of Information & Technology (Vol. 41, pp. 537–607). New Jersey.
- Euler, L. (1736). Solutio problematis ad geometrian situs pertinentis. Comentarii Academiae Scientarum Petropolitanae, 8(1741), 128–140. https://doi.org/002433.d/232323
- Luce, R. D., & Perry, A. D. (1949). A method of matrix analysis of group structure. Psychometrika, 14(2), 95–116. https://doi.org/10.1007/BF02289146
- Wilson, R. J. (1996). Introduct to graph theory. Oliver & Boyd.

*(c) vertices and edges are particularly associated with graph whereas node and links with network. Either stick to one terminology or highlight the difference between graph and network and make a statement that all terminologies (Vertices and node, edge and link) are same and they are using it without any distinction.*
We have addressed this by adding the following clarification when the definition of graph is presented Section 2.1:

L199: "Formally, a complex network can be represented by a graph *G* which consists of a finite set of elements *V(G)* called vertices (or nodes, in network terminology), and a set *E(G)* of pairs of elements of *V(G)* called edges (or links, in network terminology (Boccaletti, Latora, Moreno, Chavez, & Hwang, 2006)."

*- Further, in a single statement, the study is using Network and graph simultaneously, in my opinion, which is not acceptable. For instance, "Part I: Section 2.1, In the scientific community, the mathematical properties of a network are studied using Graph theory,"*

We agree, and have modified the sentence accordingly:
L181: " Tn the scientific  the mathematical properties of a  graph  can be studied using Graph Theory (Biggs et al.,

1976), which as mentioned above could provide a framework  to assess risk from a holistic and systemic viewpoint.  This section summarizes some of the main  concepts of Graph Theory on which the proposed methodology, presented in Section 3, is based."

*- Also, I am not fine with the definition of Graph theory. In part I, the author write that Graph Theory is the branch of mathematics that studies the properties of networks (P6/L143) whereas in the part II author states that Graph Theory is the branch of mathematics that studies the properties of graphs (P2/L37). Does the author think that Graph and Network are exactly the same? Well, in my understanding they are different indeed there is overlapping.*

We are aware of the difference between Graph and Network and agree with the Reviewer's comment. We recognize that clearer explanations were needed regarding this aspect and for this reason we added new references and modified the network and graph terminology in section 2.1 as illustrated in the previous point. Regarding to the definition of Graph Theory, we have kept the definition of Part II and modified the following sentence in Part I accordingly:

L190: "Graph Theory is the branch of mathematics that studies the properties of  graphs (Barabasi, 2016).  Graphs can represent  networks of physical elements in the Euclidean space (e.g. electric power grids and highways) or of entities defined in an intangible space (e.g. collaborations between individuals) (Wilson, 1996).

*- I am struggling to understand why authors decided to call the approach based on Graph theory and not Network theory. What makes it really a graph theory? Does the author agree that "a network is a diagrammatical representation of some physical system or structure whereas a graph, on the other hand, is a mathematical notion that represents only the structure of a network without physical meanings?" If yes, I would prefer to call approach based on Network theory.*

As underlined above, we are aware of the difference between Graph and Network. However, we are also aware that these two terms are often synonyms, as underlined by Barabasi (2016)[1]. "*In the scientific literature the terms network and graph are used interchangeably [...] Yet, there is a subtle distinction between the two terminologies [...] Yet, this distinction is rarely made, so these two terminologies are often synonyms of each other.*" (Barabasi, 2016). Nevertheless, we recognize that this could generate confusion, and for this reason we have adjusted the name of proposed approach to "graph-based" instead using "network". Furthermore, we prefer to use the term graph because we are suggesting to use it not only for networks with physical meaning (e.g. electric or sewage networks) but also interconnections between individual elements that do not explicitly constitute a network (e.g. population, houses, schools, industry). Finally, since in the proposed approach we analysed the analogy between the risk variable and graph properties, with the adoption of "graph-based" terminology we would like to put more emphasis on the study of mathematical properties ("*we use the terms {graph, vertex, edge} when we discuss the mathematical representation of these networks*" (Barabasi, 2016)).

*- Further, I also have a notion that Graph theory largely has its root in Mathematics where it has been used to conceptualize the problems into a graph whereas Network theory provides a set of techniques for analyzing such graphs. Further, many concepts such as multi-layer network, dynamic network, coarse-graining our flourishing with network theory only. At last, I would say that I am not strict with terminology since every researcher has its own notion but indeed I am more inclined to use Network theory instead of graph theory. Further, if the author continues to go with graph theory makes sure manuscripts clearly deal with the terms and concepts of only graph theory.*

As illustrated above, we have adjusted the name of the proposed approach to "graph-based" instead using "network", and we have made modifications accordingly throughout the manuscripts. We still mention the main mathematical properties of Graph Theory in sub-section 2.1, but we have removed Graph Theory from the designation of proposed approach, as can be recognized also from the proposed change to the title:
* * *
[1] Barabasi has H-index 139

L1: "Natural hazard risk of complex systems – the whole is more than the sum of its parts: I. A holistic graph-based assessment approach "

*3. Structuring and content of part I*

*The author(s) argues that part I is targeted to a more general audience who may be interested in understanding the graph theory. Being a general audience, I sincerely have difficulties to understand many terms theoretically as well as mathematically, presented examples, terminologies, mathematical concepts and more importantly the aim. As both the reviewer mentioned (reviewer 2, specifically) I feel the paper is disorganized and at last, it does not convince me. My very specific observations are as follows:*

*- Abstract: L14, this paper proposes a new holistic approach to assess the risk in a complex system based on Graph theory. What is the approach? To identify vertices of a graph, setup edges and analyze the resultant graph? I feel it is a standard way, isn't it?*

As requested, we have reduced the tone also in this specific part of the text. We clarify that the aim of the paper is to call for a paradigm shift from a reduction to holistic approach supported by the construction of a graph. This is supported by several changes throughout the document, for example in Section 2.2. and Figure 2.

*- Then, how this manuscript justify the approach? Only based on a hypothetical city example, which is very subjective. I am still struggling to understand what this manuscript contributes to the existing knowledge.*

The manuscript shows that adopting an approach based on a graph could contribute to solve some of the challenges that existing reductionist risk assessment approaches have. We restructured and amply the section "3 Methodology, the new workflow in Figure 2 shows the two major advantages of the proposed approach. We improved the section "5 Discussion and final considerations" of the manuscript in accordance with the new clarification applied in sub-section "*1.2 Modelling natural hazard risk in complex systems: state of the art and limitations*" and the new restructured sub-section "*1.3 Positioning and aims*". In particular, we clarify that this manuscript proposes a graph-based approach to assess the risk of a system that are not immediately depicted as a network: we have rewritten this at L136:

L145: "In fact, although several authors have shown how to model risk in systems which are already networks by construction (Havlin et al., 2010; Reed et al., 2009; Rinaldi, 2004; Zio, 2016), fewer have addressed the topic of risk modelling in systems where that is not the case, i.e. systems are not immediately and manifestly depicted as a network (Hammond et al., 2013; Zimmerman et al., 2019). These include cities, regions or countries, which are complex systems made of different elements (e.g. people, services, factories) connected in different ways among each other in order to carry out their own activities. Therefore, in this manuscript we would like to promote an approach, which has previously deserved the attention of other authors, to model  the interconnections between the elements that constitute those systems and assess collective risk in a holistic manner."

We are aware that this paper is only the first attempt in this direction, the illustrative example of this manuscript is then expanded in real case study application in the second part of the companion paper.

*My biggest concern is that the study deals with graph theory but does not provide any mathematical details. Which is unacceptable in such kind of study.*

We agree, and have included all relevant mathematical formulas in Table 1 (L211).

**Table 1:**  Properties of a graph **G** with **N** nodes defined by its adjacency matrix **$A(G)$** with **$N \times N$** elements $a_{ij}$, whose value is $a_{ij}>0$ if nodes *i* and *j* are connected, and 0 otherwise

| Property | Description | Formula |
|---|---|---|
| Degree (k) | The number of edges incident with the node | $$k_i = \sum_j a_{ij}$$ |
| Diameter (D) | The maximum value of all path lengths $d_{ij}$ | $$D = \max_{i,j} d_{ij},$$ where $d_{ij}$ is the geodesic length from node *i* to node *j* *(i.e. path length)*: |
| Characteristic path length (d) | The average shortest path length | $$d = \frac{1}{N*(N-1)} * \sum_{i,j(i\neq j)} d_{ij}$$ |
| Closeness (c) | Shortest path length from the node to every other node in the network | $$c_i = \frac{1}{l_i}, \qquad where\ l_i = \frac{1}{n-1} * \sum_j d_{i,j}$$ |
| Betweenness (b) | Number of shortest paths between pairs of nodes that pass through a given node | $$b_i = \sum_{j,k} \frac{n.\ of\ shortest\ paths\ connecting\ j,k\ via\ i}{n.\ of\ shortest\ paths\ connecting\ j,k}$$ $$= \sum_{j,k} \frac{n_{jk}(i)}{n_{jk}}$$ |
| Authority (x) | The value proportional to the sum of the node hub values pointing to it | $x_i = \alpha * \sum_j a_{ji} y_j$ -> $A * A^T$, where α is a proportional constant |
| Hub (y) | The value proportional to the sum of authority of nodes pointing to it | $y_i = \beta * \sum_j a_{ij} x_j$ -> $A^T * A$, where β is a proportional constant |
| Percolation threshold (pc) | The minimum value of fraction of remaining nodes (p) that leads to the connectivity phase of the graph | For random graph $p_C = \frac{1}{\bar{k}}$ , $\bar{k}$ is the average of degree |

*Page 6/Line 158, section 2.1: electrical power grid, the internet, highway and neural network, being general audience I am not able to visualize what is vertices and edges in above-mentioned graphs and importantly directions (if directed network).*

We agree that it might not be obvious what vertices and edges are, particularly on the internet and on a neural network, and that these examples are unnecessary for the purpose at hand in any case. For this reason, we have made the following modification:

L191:  Graphs can represent  networks of physical elements in the Euclidean space (e.g. electric power grids  highways) or of entities defined in an intangible space (e.g. collaborations between individuals) (Wilson, 1996).

*- Coming back to the aim of the paper, the entire study deals with the directed network and hence since the beginning the author (s) need to put more stress on the directed network. For example, section 2.1 need more words, more examples and mathematical notions about a directed network.*

After careful consideration, we consider that the tone taken throughout the article to describe networks is suitable for the purpose at hand. We believe that all the concepts of directed networks that are necessary to grasp the approach are presented in a clear manner. We agree with the Reviewer that section 2.1 lacked mathematical formulations, and have expanded Table 1 in order to address this.

*- The author(s) take many different examples to explain the network concept whenever they need, without any coherence structure, For instance, P13/L351: As an example, in a road graph, a bridge node has a higher value of betweenness because all the nodes of a sub-graph (e.g. one side of the river) need to pass through the bridge node in order to connect to the nodes of the other sub-graph (the other side of the river). In the case of bridge failure, the two sides of the river are isolated and the original road graph splits into two sub-graphs. What is the road graph? What are the edges and vertices? What is the bridge node? How to decide? Do we have a river in the road graph? It is all theoretical and very subjective. Not acceptable and convincing.*

We agree. We propose to improve this by introducing the following modifications:

L379: "As an example, consider a road network where road segments are represented by links, whereas crossroads and bridges are represented by nodes. In this case, a bridge would likely be the node  with a higher value of betweenness, because all the nodes of a sub-network (e.g. all the nodes that are in one side of the river) need to pass through the bridge node in order to connect to the nodes of the other sub-network (all the nodes on the other side of the river). In the case of a bridge failure, the two sides of the network, separated geographically by the river, are isolated and the original road network splits into two sub-networks."

*Another example is section 3.1.3 of the earthquake.*
We have removed this example from the manuscript.

*4. Mathematical details and graph theory measures*
*Page 7/line 194: A node with high hub value points to many other nodes, while a node with high authority value is linked by many different hubs. Mathematically, the authority value of a node is proportional to the sum of the node hubs pointing to it and the hub value of a node is proportional to the sum of the authority of nodes pointing to it. Is this definition provided by the authors exclusively? If not, I couldn't find any citation to valid above-written definition.*

This definition is not provided by the authors. As suggested, we have added the following citations to support the definition; (Newman, 2010) and (Nepusz & Csard, 2018).

*I am wondering whether or not this problem has a unique solution. Author claim that this article is suitable to a general audience, being an expert in network theory I am unable to understand it. I request to take a dummy graph (very simple) and explain how the author(s) setup direct graph, decide hubs, authority values and hub value of a node.*

The graph is setup following the workflow presented in section 2.2, which has been improved in the revised manuscript. We do not decide a-priori which nodes are hubs, but we use the equations presented in Table 1 that are proposed in (Newman, 2010) and implemented in R software (Nepusz & Csard, 2018).

*Page 8/line 201: Depending on the statistical properties of the degree distribution, there are two broad classes of networks: homogeneous and heterogeneous…….. Again, is this definition provided by the authors exclusively? If not, I couldn't find any citations. Is the author sure about the above-mentioned statement? I am struggling to validate this definition. What do authors mean by the homogeneous network? What are the properties of the homogeneous network? I assume the author might be pointing to a regular network because, by definition, each node in a regular network has the same number of links. If so, this is a very absurd statement. So as per the claim, all the network having Poisson*

*distributions are homogeneous? This again makes my conviction strong that authors did not check the literature appropriately.*

The definition is not provided by the authors but by Boccaletti et al. (2006). This reference, which has received thousands of citations (e.g. 6006 citations on Scopus; 9221 citations on Google Scholar), at paragraph 2.2.2 "Scale-free degree distributions" reports: *"The usual case in Science until a few years ago was that of homogeneous networks. Homogeneity in the interaction structure means that almost all nodes are topologically equivalent, like in regular lattices or in random graphs. In these distribution is binomial or Poisson in the limit of large graph size. It is not startling then that, when the scientists approached the study of real networks from the available databases, it was considered reasonable to find degree distributions localized around an average value, with a well-defined average of quadratic fluctuations. In contrast with all the expectancies, it was found that most of the real networks display power law shaped degree distribution. [...] Such networks have been named scale-free networks [2,93], because power-laws have the property of having the same functional form at all scales. [...] These networks, having a highly inhomogeneous degree distribution, result in the simultaneous presence of a few nodes (the hubs) linked to many other nodes, and a large number"*.

*What a random network is, as mentioned in (P8/L224)? Further, $Pc=1/\overline{K}$, I do not understand this mathematical expression?*

We propose to add the definition of random network:

L257: "In a random network (i.e. network with N nodes where each node pair is connected with probability p), for example, pc=$1/\overline{k}$, where $\overline{k}$ is the mean of degree $k$ (Bunde & Havlin, 1991)."

*Section 3.1.2: P12/L343: closeness: a shorter path between a node and the network? Do you mean shorted path between a node and all other nodes? Difficult to understand, please Rewrite it.*

We propose to modify as requested:

L370: "A lower value of closeness, i.e. a shortest path length from  a node  to every other nodes in the network, means a higher probability of a node of being impacted by a hazard event. On the other hand, high value of closeness, i.e. a longer path length from  a node  to every other nodes in the network, means a low probability of being impacted."

*Table 1: closeness: Which is closeness author talking about? I did not understand the definition? Is it closeness centrality? If yes, please rewrite it to "shortest path length from the node to every other node in the network". Again mathematical formulas of all the terms are unavoidable.*

Yes, we refer to closeness centrality and we propose to modify it as requested and provided the mathematical formulas in the table.

*Hub? Does the author mean two different hubs in the network? Hub value of a node and hub node itself? Confusing. Make it clear.*

We agree that it might not be clear and we modify accordingly the definition of authority in the table:

"The value proportional to the sum of the node hub values pointing to it"

*I just did not understand the concept of percolation and others in the absence of proper mathematical definitions. Mathematical details are indeed important.*

We have inserted the relevant mathematical formulas in Table 1.

*5. Section 2.2*

*"We proposed an approach based on the following two major phases". I appreciate the author's creativity in designing the text and section however, this is more general and clubbed into one section called graph construction to reduce the redundancy. In an entire section of based on topology, I couldn't see any seminal paper cited. Not appreciated. Did the*

*author has heard about coarse-graining of network or topological scale in the network? It goes in the same direction what the author has explained (entire network, community and a single node).*

Section 2 is divided in two sub-sections: the first illustrates the theoretical background and now cites a number of seminal papers (e.g Barabasi, 2016; Biggs et al., 1976; Boccaletti et al., 2006; Luce & Perry, 1949), while the second sub-section illustrates the proposed workflow. As suggested, we clubbed into one section all the original "*2.2 Proposed workflow*" and restructure the section to emphasise the two main contributes of this manuscripts: to understand the fundamental aspects of complex systems leveraging well known graph properties and invite to use the graph as a tool to model the propagation of impacts of a natural hazard.

*P10/L279: it is necessary to define rules? I would be happy to see the rules since the author claim that they are proposing an approach and hence very consolidate approach with the full explanation is needed. Section 2.2 from part II should have been here.*

At L314 we presented the example of "student go to school" where we adopt the geographical proximity rule, the same rule that is adopted in the second part of the companion paper. In the first part we provide the conceptual phases of the approach and an illustrative example, instead in the second part we apply the approach to a real data in Mexico City.

*6. Section 3.2*
*I am sorry to the author but I couldn't find anything new here.*

The aim of this paper is to introduce a new perspective in the traditional risk assessment based on a graph and not add new discovery in the network theory. Sub-section 3.2, Section 4 in the revised manuscript, shows the feasibility of the proposed approach and illustrates with a simple example the two main advantages of using the proposed graph-based approach: "*(1) understanding fundamental aspects of complex systems which may have relevant implications to natural hazard risk leveraging well known graph properties, (2) using the graph as a tool to model the propagation of impacts of a natural hazard and, eventually, assess risk in complex system*". Regarding to the second point, we had a new section in the methodology "*3.3. Hazard impact propagation via the graph*" and the Section 3.2 has been also expanded with an explanation of the potential impact propagation into the graph of an external hazard event, as describe below:

L373: "We assume that these elements are located in a flood-prone area and Bridge 3 and Block 6 are directly flooded (Figure 3.d). Since those elements are directly damaged it is possible to follow the cascading effect following the direction of the service into the graph from providers to receivers. In this artificial example, the service of transportation provided from the Bridge is lost and this has an indirect consequence to the Hospital 16 which is not directly damaged but cannot provide humanitarian services since people cannot reach the hospital any more. The graph allows to extend the impact not only to the elements directly hit by the hazard but also to all elements that receive service from element directly or indirectly affected by the hazard."

*7. Discussion*
Too much repetition, again and again, that we have proposed an approach which can solve all the problems
Page 16/line 443: "*This new approach….*" Is it really a new approach?
As mentioned previously, we have moderated the tone throughout the manuscript. The approach is no longer presented as new.

Line 443 to line 454 should go to introduction.

As requested by the reviewers, we propose to restructure the original sections "*4 Discussion*" and "*5 Conclusion*" into a new section "*5 Discussion and final consideration*", as described below. We kept the suggested sentences (L443 and L 454) because we think are useful to critically discuss and underline the limits of the adopted approach.

Line 455 and line 342: "*The proposed approach is suitable for multi-hazard assessment*" *many times repeated.*
As previous comments, we completely modify the conclusion and discussion and the new version is more synthetic and avoid repetition.

*8. Other minor comments*
Fig. 1c could be improved by weighting the links.
We modified the Figure as requested.

[Figure]

Fig.3: there are 9 blocks, not 8. Correct in the text (Section 3.2, line 403).
We correct the error.

L449: "In specific, our example includes 20 elements: 8 9 Blocks of residential buildings, 1 Hospital, 2 Fire Stations, 3 Schools, 3 Fuel Stations and 2 Bridges."

*Summary*
I am not convinced with the author's claim that this study is suitable for General Audience. The author (s) has tried to oversell the content without giving proper justification and showing any results. Indeed it fails in giving credit to previous studies. Many seminal papers are missing and it clearly seems that vast and in-depth literature review is missing. I found many terse and absurd statements. The study clearly lacks in terms of mathematical justifications. Too many over claim statement such as approach "Could be used for risk mitigation strategies", "Similar analysis could be carried out for betweenness to obtain more insights into the risk assessment". Therefore, I recommend rejection of part I.
Nevertheless, it was a great attempt and I motivate the team to work and explore more to fill the gaps. Due to this, I am afraid that the entire manuscript should be rejected, and the authors should given an opportunity to merge both the manuscripts and resubmit. I am willing to review the contribution, if the authors want to submit a revised work. I hope I am not unduly discouraging, but the problems detailed above are sufficiently severe for the work not to be considered for publication in the current form. However, I do think that the authors have applied an interesting methodology that should be investigated thoroughly, and which may lead to important breakthroughs in the area of interest. Therefore, I encourage the authors to further pursue this method, and I hope my suggestions are useful for this endeavor.

We appreciate the Reviewer's candid comments. We recognize that the previous version of the article had a considerable number of flaws, many of which are well summarized in this comment. In the revised version that we are now submitting, we believe to have addressed most of the issues raised, as described in the responses above. This includes a moderation of the overall tone regarding the novelty of a graph-based approach. Having said this, we must note that we found the Reviewer's criticism on the overall scientific contribution of the manuscripts excessive. We remain convinced that the novelty and potential of the application of graphs to model complex systems of elements within the field of natural hazard risk is in itself a significant contribution (as confirmed also by our expansion of the literature review in the new version), and the reviewer may not be able to fully appreciate being an expert in the field of network theory. Nevertheless, we fully recognize that the shortcomings present in the previous version made this evaluation more difficult. We believe that the extensive modifications introduced to both articles represent a vast improvement, and are thankful to the Reviewer for his/her willingness to review this new version, despite his/her concerns regarding the previous version.

[revised manuscript text omitted]

It is probably necessary before going into the methodology, to provide a short summary on the general concepts and the main properties of the graph that will be used in the proposed graph-based approach.

As discussed above, a network graph could allow portraying the complexity of a risk system. TIn the scientific community, the mathematical properties of a network graph are can be studied using Graph Theory (Biggs et al., 1976), which as mentioned above could alsocould provide a framework provide a better angle to assess risk from a holistic and systemic viewpoint. The following paragraphs review This section summarizes some of the main aspects concepts of Graph Theory, on which our approachthe 
[revised manuscript text omitted]

| | Steps | Description | Conceptual representation |
|---|---|---|---|
| **Network conceptualization** | **1. Typologies** | Identify the relevant typologies of exposed elements (e.g. populations, fire stations, schools, bridges) | |
| | **2. Connections** | Define the connections between typologies (e.g. fire stations provide recovery service to households) | |
| **Graph construction** | **3. Rules** | Define the rules between elements (e.g. associate each household to the closest fire station) | $F_i$ $B_i$ $F_j$ $B_j$ |
| | **4. Links** | Build the graph: all exposed elements are linked and establish a unique network | |

[Figure]

**Figure 2: Workflow.**

[Figure]

780

[Figure]

**Figure 3: (a) Map of the various elements of a hypothetical municipality in a flood-prone area; (b) Same, with node sizes proportional to authority values; (c) Same, with node sizes proportional to hub values; d) Same, with flood area and nodes directly impacted highlighted with in red cross; (e) Same, with also the nodes indirectly impacted highlighted with black cross.**

785

**Table 1:  Properties of a graph G with N nodes defined by its adjacency matrix $A(G)$ with $N \times N$ elements $a_{ij}$, whose value is $a_{ij}>0$ if nodes $i$ and $j$ are connected, and 0 otherwise**

|  |  |
|---|---|
|  |  |
|  |  |
|  |  |
|  |  |
|  |  |
|  |  |
|  |  |

| Property | Description | Formula |
|---|---|---|
| Degree (k) | The number of edges incident with the node | $$k_i = \sum_j a_{ij}$$ |
| Diameter (D) | The maximum value of all path lengths $d_{ij}$ | $D = \max_{i,j} d_{ij}$, where $d_{ij}$ is the geodesic length from node $i$ to node $j$ *(i.e. path length)*: |
| Characteristic path length (d) | The average shortest path length | $$d = \frac{1}{N*(N-1)} * \sum_{i,j(i \neq j)} d_{ij}$$ |
| Closeness (c) | Shortest path length from a node to every other nodes in the network | $$c_i = \frac{1}{l_i}, \quad where \; l_i = \frac{1}{n-1} * \sum_j d_{i,j}$$ |
| Betweenness (b) | Number of shortest paths between pairs of nodes that pass through a given node | $$b_i = \sum_{j,k} \frac{n. \; of \; shortest \; paths \; connecting \; j,k \; via \; i}{n. \; of \; shortest \; paths \; connecting \; j,k} = \sum_{j,k} \frac{n_{jk}(i)}{n_{jk}}$$ |
| Authority (x) | The value proportional to the sum of the node hub values pointing to it | $x_i = \alpha * \sum_j a_{ji} y_j$ -> $A * A^T$, where $\alpha$ is a proportional constant |
| Hub (y) | The value proportional to the sum of authority of nodes pointing to it | $y_i = \beta * \sum_j a_{ij} x_j$ -> $A^T * A$, where $\beta$ is a proportional constant |
| Percolation threshold (pc) | The minimum value of fraction of remaining nodes (p) that leads to the connectivity phase of the graph | For random graph $p_C = \frac{1}{\bar{k}}$, $\bar{k}$ is the average of degree |

790 **Table 2: Analogy of risk variables with graph properties.**

| Risk variables | Analogy with graph properties |
|---|---|
| Exposure | The authority represents how the system privileges the nodes, conferring them more or less importance compared with others, according to the connections established in the system. |
| Vulnerability | The propensity of parts of the network to be isolated because of hazard events. The closeness of a node is a measure of the single node vulnerability within the system, while degree distribution, hub, and betweenness are measures of vulnerability of the system as a whole. |
| Resilience | The percolation threshold, together with the network fragmentation analysis, explain the resilience of the network after a perturbation. |

---

## Author Response (AR3)

**Response to Referee #1**

We wish to thank the Referee for his/her time and effort reviewing the manuscript.

*The authors have made substantial improvements to the two papers by engaging further with the literature, more clearly specifying their contribution, and toning down their claims of novelty. However, my major concern—which other reviewers and I have expressed in previous reviews—that the research is not sufficiently unique to require two papers, remains. The three summary aims identified in the first article have been addressed by other articles. The two gaps identified in the first article (that modelling the complexity of interconnections between individual elements that do not explicitly constitute a network, and that there are few representations of network interdependencies in flood risk assessments) are not well elaborated and are not carried through throughout the two papers. As a result, the first paper reads like an extended literature review, with the discussion and final considerations providing little of note, and there are still questions on what the second paper contributes.*

*The changes the authors have made have been moving the articles in the right direction, but are not sufficient enough to provide a gap for both articles. Because this is the third time offering this critique, I again (strongly) recommend rejecting the theory based article and using it as an introduction/literature review/methodology for the second article.*

Based on the Reviewer's suggestions, we have merged the two manuscripts into a new single article. This new article follows a more conventional structure, where the contents from the theory-based article have been used as the introduction, literature review and description of the methodology, as proposed by the Reviewer. In addition, some redundant content has been removed, and several changes have been made in order to improve the readability and quality of the manuscript.

**Response to Referee #3**

We wish to thank the Referee for his/her time and effort reviewing the manuscript.

**General summary on the revised manuscript**

*I received the third version of both the manuscripts describing the potential advantages of the use of a graph theory particularly in understanding fundamental aspects of complex systems which may have relevant implications to natural hazard risk and among others. In addition, part II illustrates an application of the graph theory to a pilot study in Mexico City. I could see a little refinement in both the manuscripts and I do acknowledge author's efforts, but there are still many open issues. I'm still not completely convinced by the revised version and hence I do emphasize that all my previous major concerns are still remains unanswered. These concerns need to be sincerely addressed before this manuscript could be thought to proceed with publications, which are as follows:*

*Standalone contribution of each individual manuscript*

*In my last report, I very clearly emphasized that both the manuscripts do not standalone. I am unhappy to see that authors overlooked my efforts on the manuscript and tried to come up with good answers rather than sincerely working on it. I again point to the aim of the paper (P5/L139-145). Is stating or elaborating on construction of a hypothetical graph (shown in Fig.3), potential advantages of the use of the graph, limitation, potentialities etc. is a worthy contribution when authors already acknowledged that it is not a new approach?*

We have merged the two manuscripts into a new single article. We have acknowledged very clearly that our approach is not a new approach, but the contribution to an ongoing research that we believe is promising for the risk assessment of natural hazard in complex systems.

*A complete study of all relevant graph properties (P15/L411-415) discussed above and a more realistic hazard scenario are presented in Part II of this manuscript for a selected case study. (this statement again shows the need to merge both the studies)*

In the new manuscript we have followed the reviewer's suggestion.

*Mathematical details and graph theory measures*

*Mathematical representation and visualization (Section 2): I could see that authors have provided all the mathematical formulas in the Table 1. However, the idea was to take an example of a simple graph and discuss all these measures and there significance mathematically. With this simple graph, author could have discussed network terminologies used in the manuscript such as hub node, authority etc. (see my previous comments: The author(s) argues that part I is targeted to a more general audience who may be interested in understanding the graph theory. My biggest concern is that the study deals with graph theory but does not provide any mathematical details. Which is unacceptable in such kind of study). (Please see my previous comments: I am wondering whether or not this problem has a unique solution (NOT*

*ANSWERED!!!). Author claim that this article is suitable to a general audience, being an expert in network theory I am unable to understand it. I request to take a dummy graph (very simple) and explain how the author(s) setup direct graph, decide hubs, authority values and hub value of a node).*

In the previous review step, we implicitly answered the question of whether there is a unique solution by providing unambiguous definitions for the graph metrics, supported by different references. We have also included the mathematical expressions of the main properties of the Graph as requested by the Reviewer. However, we believe that, being a consolidated theory, the purpose of the paper is not to report all of its details (which are in any case reported in the cited references), but to summarize the main concepts that we use in our approach. We acknowledged that our approach is taking advantage of an existing and consolidated theory, which we do not think justifies a full coverage in this paper, even more so in this new version where there is no longer a specific paper dealing with theoretical aspects. Nevertheless, the illustrative example is included to address the Reviewer's request: presenting a simple, dummy graph and explaining the basics of the graph construction before moving to the more complex pilot study.

*Subjective examples and interpretations*

*As pointed previously as well, the example presented in Fig. 3 (section 4) is very subjective and case specific. I am not able to replicate it. How should I use it on another problem? Let's imagine a system of airport with 20 blocks, 3 out of which deal with intercontinental flights (short, mild and long distance respectively), 3 deals with intracontinental flights (short, mild and long distance respectively) and one block is for local flights. Other blocks are fire stations, fuel stations etc. Reading through the manuscript leaves one with many questions than answers. Authors critically need to asses the objectivity of the manuscript in its present form.*

The illustrative example of Fig. 4 (in the new version) aims at exemplifying a specific case based on arbitrary assumptions not referred to a real case. Therefore, (1) there is a subjective decision at the origin of it, (2) it does not cover the general case. On the other hand, in Fig. 1 we present the workflow of the methodology and in paragraph 2.1 we present the procedure for the general case. Moreover, we believe also that the pilot study provides the concrete implementation of the theoretical procedure on a real case and can thus contribute to the understanding of the methodology. Nevertheless, in Section 4 we discussed that in any case there will be a subjective interpretation of the relationships among the elements of a network and on the construction of the graph mainly due to the lack of data. However, we believe that new database could in the future provide more information and reduce the subjectivity embedded in the analysis.

*This illustrative example shows (P15/L401-405) how the single elements can be considered as part of the whole network and not as single separate entities. This holistic approach adds information to the traditional approach…. SUCH STATEMENTS ARE VERY GENERIC.*

In the revised manuscript we removed the generic statements of this section.

*Vague statements*

*(P16/L438-444): First of all, there is a mature theory of mathematics, the Graph Theory, that already studies the properties of a graph (REPETITIVE). Are these graph properties telling us something useful to assess the risk of natural hazards affecting these complex systems? We showed that some of the graph properties can disclose some relevant characteristics of the system related to the risk 440 assessment. What is the vulnerability and exposure of the system? There are some interesting analogies between graph properties such as hub, betweenness and degree-out values and the "systemic" vulnerability. WHAT ARE SOME?*

In the paragraph 2.2.2 we listed and discussed which are the analogies.

*I couldn't find any limitations which is also one aim of the study (P6/L144: to discuss the limitations, potentialities and future developments of this approach compared to other more traditional approaches).*

In the discussion and final considerations, we present the limitations of the methodology and of the study case.

*In my last review I pointed that the author(s) took many different examples to explain the network concept whenever they need, without any explanation (what is node, what is link, how setup link etc.). In the revised revision, author escaped by deleting many such examples still without explaining the retained ones.*

*(P12/L315-323): As an example, in a road network where road segments are represented by links, whereas crossroads and bridges are represented by nodes. In this case, a bridge would likely be the node with a higher value of betweenness because all the nodes of a sub-network (e.g. all the nodes that are in one side of the river) need to pass through the bridge node in order to connect to the nodes of the other sub-network 320 (all the nodes on the other side of the river). In the case of a bridge failure, the two sides network, separated geographically by the river are isolated and the original road network splits into two sub-networks. THIS IS NOT TRUE!!!*

[Figure]

*I find the bridge node (node number 7) having less betweenness than crossroads.*

The reviewer is right. In the general case, it is not necessarily true that a bridge has a betweenness value greater than the other nodes. However, our aim is to link the betweenness properties to the vulnerability of a network. The presence of nodes with high betweenness values in a network could more likely produce fragmented subnetworks regardless of the type of node (bridge, crossroads, etc.). For this reason, we took out the example and we left the general consideration on this issue.

[revised manuscript text omitted]
 City is situated in a high mountain valley (approximately 2,200 m a.s.l) surrounded by mountains of volcanic origin in the southern part of the Basin of Mexico. Mexico is one of the most seismological active regions on earth (Santos-Reyes et al., 2014), floods and storms are recorded in indigenous documents, and the Popocatépetl volcano has erupted intermittently for at least 500,000 years. At present, people settle in hazardous areas such as scarps, steep slopes, ravines and next to stream channels.

The Mexico City Metropolitan Area (MCMA) is one of the largest urban agglomerations in the world. Located in a closed basin of 9,600 km$^2$, MCMA spreads over a surface of 4,250 km$^2$. The MCMA has a metropolitan population estimated at 21.2 million, concentrating 18% of country's population, and generates 35% of Mexico's gross domestic product on a surface equivalent to less than 0.3% of the national territory (
[revised manuscript text omitted]
 network can assess the nodes from two main points of views: 1) providers, elements that provide services and 2) receivers, elements that receive services. Regarding the providers, it is relevant to explore how some providers compare to others in terms of relevance to the system, according to their connections with the receivers. In particular, we propose a comparison between providers through the analysis of two properties: hub analysis of all nodes that provide service to the population, and betweenness analysis of the Crossroads.

**Providers: role of hubs**

In directed graphs, it is important to explore if some nodes are more important for network function than others. The importance of a node, 
[revised manuscript text omitted]

---

## Author Response (AR4)

**Response to Referee #1**

We wish to thank the Referee for his/her time and effort reviewing the manuscript.

*- The authors continue to improve the article, yet questions remain about the article's specific rationale and focus. The authors need to develop a clearer rationale and carry it through the entire article. The literature review primarily focuses infrastructure literature, arguing that it focuses mostly on "the analysis of a single infrastructure typology" and identifies SoS as a potential solution. I would expect the rest of the article to highlight this gap and then develop a solution. Yet the positioning and aims does not discuss this but instead focuses on general systems, their interconnections, and their need for a holistic assessment. Whether the article is about SoS and infrastructure interdependencies or a broader question of general systems the authors should revise these sections to build a more coherent argument for the article.*

We propose some modifications in the manuscript in order to build a more coherent argument for the article, to better highlight the existing gaps and the solutions provided by this work. These changes are described in detail below:

L123: "The aspects of complexity and interdependency have been investigated by various models of critical infrastructure as a single system, or as systems of systems, which are networks by construction (e.g. drainage system or electric power network, Holmgren, 2006; Navin, 2016). However,  the current practice related to both single system and SoS needs further research, in particular when it comes to modelling the complexity of interconnections between individual elements that do not explicitly constitute a network, which tends to be neglected by traditional reductionist risk assessments."

L160: "The construction of a graph for systems already in a form of a network is well developed and consolidated in the literature (e.g. Rinaldi, 2004; Setola et al., 2016). Instead, the use of the graph-theory – and the exploitation of its diagnosis tools - for systems non already structurally in the form of a network is relatively new. At this regard, in this section we propose a procedure to build a graph for a complex system such as a city by linking the individual elements constituting it."

L365: "While the literature of the impact propagation or cascading effects for critical infrastructures is large (e.g. Pant et al., 2018; Trucco et al., 2012), applications on the risk quantification of natural hazard including the cascading effects are scarce."

*- The authors then need to carry this into the methodology, either referencing or building on previous research to justify their approach. The methodology section overviews literature on graph theory before discussing concepts related to exposure, vulnerability, and resilience and how they might be represented in a graph. While this conceptual discussing is important for the remainder of the article, it is not referenced or imbedded in previous studies. Are the*

30   *operationalizations outlined in this section new or do they build on work of others? Why are they worth looking at? Be explicit about why it's important to understand issues of exposure, vulnerability, and resilience, and if the concepts are new to this type of analysis say so. This discussion should reference literature and occur in both the literature review and the methodology section.*

Since we assumed that exposure/vulnerability/resilience are well known concepts for the risk community, we briefly
35   recall their definition in the introduction without discussing in depth the issues related to risk assessment until the presentation of the approach based on graph theory. We'd like to keep this order.

The suggestion to the take the advantage of the graph properties to gain an insight to the risk properties of a complex system, as far as we know, is new. We specified this better now at:

L303: "For the three other variables, namely, exposure, vulnerability, and resilience, below we propose and provide an
40   innovative and original  discussion on their analogies with the graph properties presented in previous Sub-section. The analogies are summarized in Table 2."

The modification provided above (L123, L160 and L365) clarify better what are the novelty of this approach. We specified better now the importance to look even just at the risk components before quantifying the risk in order to understand which features of the system can affect more its risk propensity/adversity.

45   L700: "In section 2.2 and 3.2 we highlight the importance, before to quantify the risk, to look the single risk components from the systemic lens provided by the graph properties. This information could support more informed DRR decision making by strategically suggest how to prioritize intervention in order to minimize exposure and vulnerability from a system point of view"

*Finally, the discussion and final considerations should discuss how this study built on, challenged, and/or complemented
50   previous research. What new does this study say? How is it similar or dissimilar to other studies? How does it support more informed DRR decision making (one of the aims identified by the authors). Again, the authors need to bring in and reference other studies, comparing their results to the results of others.*

In the final conclusion, as we did for the previous sections, we explicitly remark the specific aspects of this research that are novelty and how this approach can provide support for DRR decision making. In particular at:

55   L696: "What is the vulnerability and exposure of the system? We proposed  new analogies between some graph properties such as authority, hub, betweenness and degree-out values and the "systemic" exposure and vulnerability."

L700: "In section 2.2 and 3.2 we highlight the importance, before to quantify the risk, to look the single risk components from the systemic lens provided by the graph properties. This information could support more informed DRR decision making by strategically suggest how to prioritize intervention in order to minimize exposure and vulnerability from a system point of view."

L717: "The application to the case of urban flooding in Mexico City it is a first attempt to demonstrate the feasibility of the proposed approach and it is also the first example in literature that try to quantitatively analyse the propagation of impact into a network of individual elements that do not explicitly constitute a network."

L739: "Furthermore, the proposed approach could introduce a common base for future research on both multi-hazard and integrated risk assessment."

*A few other minor comments:*

*1. tone down the opening statement of 'we live in an increasingly complex world'. Very difficult statement to provide and tangential to the study. Instead, say something like 'the world is complex' and then describe how it's complex. Easier to defend and aligns with the argument.*

We modified opening statement as suggested.

*2. define resilience (paragraph starting on line 330). Resilience is a fraught and contested topic with a multitude of meanings (see e.g. Manyena 2006 'the concept of resilience revisited' as an example of the multitude of meanings). What do you mean by resilience? Is it about recovery and 'return to normal', ability to absorb shocks, the ability to transform in the face of adversity, or something else?*

The resilience definition is provided at L47 and we also underline that the concept of resilience is still being debated.

*3. avoid the term 'natural disasters'. Hazards may be natural but disasters are not. Instead you can say 'disaster induced by natural hazard'.*

We modified accordingly at L350.

[revised manuscript text omitted]